

**Description and evaluation of the UKCA stratosphere-troposphere chemistry scheme**
**(StratTrop vn 1.0) implemented in UKESM1.**
Archibald, Alexander T. [1,2,*], O'Connor, Fiona M. [3],  Abraham, N. Luke[1,2], Archer-Nicholls,
Scott[1], Chipperfield, Martyn P.[4,5], Dalvi, Mohit [3], Folberth, Gerd A. [3], Dennison, Fraser [4],
Dhomse, Sandip S.[4,5], Griffiths, Paul T. [1,2], Hardacre, Catherine[3], Hewitt, Alan J. [3], Hill,
Richard[3], Johnson, Colin E.[3], Keeble, James[1,2], Köhler, Marcus O.[1,7,†], Morgenstern, Olaf[6],
Mulcahy, Jane P.[3], Ordóñez, Carlos[3,‡], Pope, Richard J.[4,5], Rumbold, Steven[3,8], Russo,
Maria R. [1,2], Savage, Nicholas[3], Sellar, Alistair[3], Stringer, Marc[3], Turnock, Steven[3], Wild,
Oliver[9] and Zeng, Guang[6].
1) Department of Chemistry, University of Cambridge, Cambridge, UK, CB2 1EW
2) NCAS-Climate, University of Cambridge, UK, CB2 1EW
3) Met Office Hadley Centre, FitzRoy Road, Exeter, UK, EX1 3PB
4) School of Earth and Environment, University of Leeds, Leeds, UK, LS2 9JT.
5) National Centre for Earth Observation (NCEO), University of Leeds, Leeds, U.K.
6) National Institute of Water & Atmospheric Research Ltd (NIWA), 301 Evans Bay
Parade, Greta Point, Wellington, New Zealand.
7) Centre for Ocean and Atmospheric Sciences, School of Environmental Sciences,
University of East Anglia, Norwich, U.K.
8) NCAS-Climate, Department of Meteorology, University of Reading, Reading, UK,
RG6 6BB.
9) Lancaster Environment Centre, Lancaster University, Lancaster, UK, LA1 4YQ
(†) Now at ECMWF, Reading, UK.
(‡) Now at Departamento de Física de la Tierra y Astrofísica, Facultad de Ciencias
Físicas, Universidad Complutense de Madrid, Madrid 28040, Spain
*Email: ata27@cam.ac.uk
**Abstract**
Here we present a description of the UKCA StratTrop chemical mechanism which is used in
the UKESM1 Earth System Model for CMIP6. The StratTrop chemical mechanism is a merger
of previously well evaluated tropospheric and stratospheric mechanisms and we provide
results from a series of bespoke integrations to assess the overall performance of the model.
We find that the StratTrop scheme performs well when compared to a wide array of
observations. The analysis we present here focuses on key components of atmospheric
composition, namely the performance of the model to simulate ozone in the stratosphere and
troposphere and constituents that are important for ozone in these regions. We find that the
results obtained from the use of the StratTrop mechanism are sensitive to the host model;
simulations with the same chemical mechanism run in an earlier version of the MetUM host
model show a range of sensitivity to emissions that the current model does not fall within.
Whilst the general model performance is suitable for use in the UKESM1 CMIP6 integrations,
we note some shortcomings in the scheme that future targeted studies will address.



**1.0 Introduction**

The ability to model the composition of the atmosphere is vital for a wide range of applications relevant to society at large. Atmospheric composition modelling can broadly be subdivided into two sub disciplines: (1) aerosol processes and microphysics and (2) atmospheric chemistry. Coupling these processes in climate models is paramount for being able to simulate atmospheric composition at the global scale. The most societally important questions revolve around understanding how the composition of the atmosphere has changed over the past, attributing this change, understanding how this system is likely to change into the future and what the impacts of these changes are on the Earth system and on human health. It is these pressing issues that have led to the development of the new UK Earth System Model, UKESM1 (Sellar et al., 2019a), which uses the UK Chemistry and Aerosol model (UKCA) (O'Connor et al., 2014; Morgenstern et al., 2009; Mulcahy et al., 2018) as its key component to simulate atmospheric composition in the Earth system. The key challenge UKCA is applied to is understanding and predicting how the concentrations of a range of trace gases, especially the greenhouse gases methane ($CH_4$) ozone ($O_3$) and nitrous oxide ($N_2O$), and aerosol species will evolve in the Earth system under a range of different forcings. UKCA simulates the processes that control the formation and destruction of these species. Here we describe and document the performance of the version of UKCA used in UKESM1, which includes a representation of combined stratospheric and tropospheric chemistry that enhances the capability of UKCA beyond that used in the Atmospheric Chemistry and Climate Model Intercomparison Project (ACCMIP; Young et al., 2013; O'Connor et al., 2014) and the recent Chemistry-Climate Model Initiative (CCMI) intercomparison (Bednarz et al., 2018; Hardiman et al., 2017; Morgenstern et al., 2017). There have been a number of versions of UKCA with defined scopes but we denote the version used in UKESM1 and described here as UKCA StratTrop, to signify its purpose of holistic treatment of composition processes in the troposphere and stratosphere.

As a result of the Chemistry-Climate Model Validation Activity (CCMVal), it was recommended that models which are aimed at simulating the coupled ozone-climate problem should include processes to enable interactive ozone in the troposphere and stratosphere (Morgenstern et al., 2010). In Chemistry-Climate Models (CCMs), the chemistry schemes are used to describe the reactions that chemical compounds undergo. These chemistry schemes can be constructed to explicitly model a specific chemical reaction system (Aumont et al., 2005) but in most applications the chemistry schemes are heavily simplified. Until recently, models of atmospheric chemistry tended to focus on chemistry schemes formulated for limited regions of the atmosphere; detailed schemes have been constructed to examine phenomena in the stratosphere, such as ozone depletion, whereas other schemes that focus on the troposphere have been developed to examine phenomena such as air pollution. An example of this using the UKCA model framework are two studies of the effects of the eruption of Mt. Pinatubo, where Telford et al. (2009) used the stratospheric scheme of Morgenstern et al. (2009) to study the effects of the eruption on stratospheric ozone, whereas Telford et al. (2010) used the tropospheric scheme of O'Connor et al. (2014) to examine the effects on tropospheric oxidising capacity. Whilst the chemical schemes described in O'Connor et al. (2014) (hereafter



OC14) and Morgenstern et al. (2009) (hereafter MO09) have some overlap (for example the
use of some common reactions) the schemes were developed with specific applications in
scope. The reason for partitioning chemical complexity up like this is to reduce the
computational resources required. Moreover, simulations with these process limitations were
found to be able to capture the phenomena of interest.
However, increases in computational power and a drive to answer a greater number of
questions from model simulations have allowed models that simulate both the stratosphere
and troposphere to be developed and which are now widely used (e.g. Pitari et al., 2002;
Jockel et al., 2006; Lamarque et al., 2008; Morgenstern et al., 2012). The removal of the need
for prescribed upper boundary conditions (for the stratosphere), and a more comprehensive
chemistry scheme, make their increased cost worth bearing. In this work, we describe the
implementation of a combined chemistry scheme suitable for simulating the stratosphere and
the troposphere within the UKCA model as used in UKESM1 (Sellar et al., 2019a). This
scheme, UKCA StratTrop, builds on and combines the existing stratospheric (MO09) and
tropospheric schemes (OC14). In various configurations of UKCA (under the names
HadGEM3-ES, UMUKCA-UCAM, NIWA-UKCA, ACCESS), this combined chemical scheme
has already been used to study: stratospheric ozone and its sensitivity to changes in bromine
(Yang et al., 2014), subsequent circulation changes (Braesicke et al., 2013), and how it may
be impacted by certain forms of geoengineering (Tang et al., 2014); the role of ozone radiative
feedback on temperature and humidity biases at the tropical tropopause layer (TTL)
(Hardiman et al., 2015); the effects on tropospheric and stratospheric ozone changes under
climate and emissions changes following the Representative Concentration Pathways (RCPs)
(Banerjee et al., 2015; Dhomse et al., 2018); climate induced changes in lightning (Banerjee
et al., 2014) and changes in methane chemistry between the present day and the last
interglacial (Quiquet et al., 2015). The scheme has been included in model simulations as part
of the CCMI project (Eyring et al., 2013; Hardiman et al., 2017; Morgenstern et al., 2017;
Dhomse et al., 2018) as well as all future Earth System modelling studies using the UKESM1
model (Sellar et al., 2019a).
This paper is organised in the following sections: In Section 2, we present a thorough
description of UKCA StratTrop, including the physical model and details of the chemistry
scheme, followed by a detailed description of the emissions used in the StratTrop scheme and
some notes on the historical development of the scheme. In Section 3, we describe the two
15-year simulations we have performed with UKCA StratTrop in an atmosphere-only
configuration of UKESM1. In Section 4, we use these simulations to review the performance
of UKCA StratTrop, focusing on the model's ability to simulate key features of tropospheric
and stratospheric chemistry as simulated by other models or observed using in situ and remote
sensing measurements. Finally, in Section 5, we discuss the performance of the model and
make some recommendations for further targeted studies.
**2.0 Model Description**



In this section, we present a thorough description of UKCA StratTrop, from the host physical
model to the detailed process representation of the StratTrop chemistry scheme.
**2.1 Physical Model**
The physical model to which the UKCA StratTrop chemistry scheme has been coupled is the
Global Atmosphere 7.1/Global Land 7.0 (GA7.1/GL7.0; Walters et al., 2019) configuration of
the Hadley Centre Global Environment Model version 3 (HadGEM3; Hewitt et al., 2011).
The coupling between the UKCA StratTrop chemistry scheme and the GA7.1/GL7.0
configuration of HadGEM3 is based on the Met Office's Unified Model (MetUM; Brown et al.,
2012). As a result, UKCA uses aspects of MetUM for the large-scale advection, convective
transport, and boundary layer mixing of its tracers. The large-scale advection makes use of
the semi-implicit semi-Lagrangian formulation of the ENDGame dynamical core (Wood et al.,
2014) to solve the non-hydrostatic, fully compressible deep-atmosphere equations of motion.
These are discretized on to a regular latitude-longitude grid, with Arakawa C-grid staggering
(Arakawa and Lamb, 1977). The discretization in the vertical uses Charney–Phillips staggering
(Charney and Phillips, 1953) with terrain-following hybrid height coordinates. Although
GA7.1/GL7.0 can be run at a variety of resolutions, as detailed in Walters et al. (2019), the
resolution here is N96L85 (1.875° x 1.25° longitude-latitude) i.e. approximately 135 km
resolution in the horizontal and with 85 levels covering the altitude range from the surface to
85 km. Of the 85 terrain-following model levels, 50 lie below 18 km and 35 levels are above
18 km (Walters et al., 2019). Mass conservation of UKCA tracers is achieved with the
optimised conservative filter (OCF) scheme (Zerroukat and Allen, 2015); use of this scheme
for virtual dry potential temperature resulted in reducing the warm bias at the tropical
tropopause layer (TTL) (Hardiman et al., 2015; Walters et al., 2019). This conservation
scheme is also used for moist prognostics (e.g. water vapour mass mixing ratio and prognostic
cloud fields). Although it makes the conservation scheme for moist prognostics consistent with
the treatment of UKCA tracers and virtual dry potential temperature, Walters et al. (2019)
found that it had little impact on moisture biases in the lower stratosphere.
The convective transport of UKCA tracers is treated within the MetUM convection scheme. It
is essentially the mass flux scheme of Gregory and Rowntree (1990) but with updates for
downdrafts (Gregory and Allen, 1991), convective momentum transport (Gregory et al., 1997),
and Convective Available Potential Energy closure. The scheme involves diagnosis of
possible convection from the boundary layer, followed by a call to shallow or deep convection
on selected grid points based on the diagnosis from step one, and then a call to the mid-level
convection scheme at all points. One key difference between the convective treatment of
UKCA chemical and aerosol tracers is that convective scavenging of aerosols (simulated with
GLOMAP-mode) is coupled with the convective transport following Kipling et al. (2013),
whereas for chemical tracers, convective transport and scavenging are treated independently.
Further details on the convection scheme in GA7.1 can be found in Walters et al. (2019).
Finally, mixing over the full depth of the troposphere is carried out by the so-called "boundary-
layer" scheme in GA7.1; this scheme is that of Lock et al. (2000), but with updates from Lock
(2001) and Brown et al. (2008).
The GA7.1/GL7.0 configuration described in Walters et al. (2019) already includes the two-
moment GLOMAP-mode aerosol scheme from UKCA (Mann et al., 2010; Mulcahy et al., 2018;





Mulcahy et al., 2019), in which sulphate and secondary organic aerosol (SOA) formation is
driven by prescribed oxidant fields. In the UKCA-StratTrop configuration described here, the
oxidants driving secondary aerosol formation are fully interactive; this coupling between UKCA
chemistry and GLOMAP-mode is fully described in Mulcahy et al. (2019). Together with
dynamic vegetation and a terrestrial carbon/nitrogen scheme (Sellar et al., 2019a),
GA7.1/GL7.0 and UKCA StratTrop make up the atmospheric and land components of the UK
Earth System Model, UKESM1 (Sellar et al., 2019a) which will be used as part of the UK
contribution to the 6th Coupled Model Intercomparison Project (CMIP6; Eyring et al., 2016).
**2.2 Chemistry scheme**
The UKCA StratTrop scheme is based on a merger between the stratospheric scheme of
MO09 and the tropospheric "TropIsop" scheme of OC14. StratTrop simulates the $O_x$, $HO_x$
and $NO_x$ chemical cycles and the oxidation of carbon monoxide, ethane, propane, and
isoprene in addition to Cl and Br chemistry, including heterogeneous processes on polar
stratospheric clouds (PSCs) and liquid sulphate aerosols (SAs). The level of detail of the
VOC oxidation is far from the complexity of explicit representations (Aumont et al., 2005) but
the VOCs simulated are treated as discrete species.
Wet deposition is parameterised using the approach of Giannakopoulos et al. (1999). Dry
deposition is parameterised employing a resistance type model (Wesely, 1989) using the
implementation described in OC14, updated to account for advancements in the Joint UK Land
Environment Simulator (JULES; Best et al., 2011), in particular a significant increase in land
surface types (an increase from 9 to 27; see below for more details). Interactive photolysis is
represented with the Fast-JX scheme (Neu et al., 2007), as implemented in Telford et al.
(2013). Fast-JX covers the wavelength range of 177 to 750 nm. For shorter wavelengths,
effective above 60 km of altitude, a correction is applied to the photolysis rates following the
formulation of Lary and Pyle (1991).
The StratTrop scheme includes emissions of 12 chemical species: nitrogen oxide (NO),
carbon monoxide (CO), formaldehyde (HCHO), ethane ($C_2H_6$), propane($C_3H_8$), acetaldehyde
($CH_3CHO$), acetone (($CH_3$)$_2CO$), methanol ($CH_3OH$) and isoprene ($C_5H_8$) in addition to trace
gas aerosol-precursor emissions (dimethyl sulphide (DMS), sulphur dioxide ($SO_2$), and
monoterpenes). For the implementation used in UKESM1, emissions may be prescribed or
interactive and are described in more detail in Sections 2.6.1 to 2.6.3. A further 7 species
($N_2O$, $CF_2Cl_2$, $CFCl_3$, $CH_3Br$, COS, $H_2$, and $CH_4$) are constrained by lower boundary conditions;
for more details see Section 2.6.4.
UKCA StratTrop was developed by starting with the stratospheric chemistry scheme (MO09)
and adding aspects of chemistry unique to the tropospheric scheme (OC14). In most cases
the formulation and reaction coefficients are taken from reference evaluations (JPL and
IUPAC) or the Master Chemical Mechanism, as detailed in OC14. Table 1 provides a list of
the chemical tracers included in the StratTrop configuration used in UKESM1. Tables 2-5
include lists of the bi-molecular, ter-molecular, photolysis and heterogeneous reactions
included in the model configuration. In total the model employs 84 tracers and represents the
chemistry of 81 of these (Table 1). $O_2$, $N_2$ and $CO_2$ are not treated as chemically active species.



This chemistry scheme accounts for 199 bimolecular reactions (Table 2), 25 uni- and
termolecular reactions (Table 3), 59 photolytic reactions (Table 4), 5 heterogeneous reactions
(Table 5) and 3 aqueous phase reactions for the Sulfur cycle (Table 6). Hence, UKCA-
StratTrop describes the oxidation of the organic compounds – methane, ethane, propane and
isoprene and their oxidation products – coupled to the inorganic chemistry of $O_x$, $NO_x$, $HO_x$,
$ClO_x$ and $BrO_x$, using a continuous set of equations with no artificial boundaries imposed on
where to stop performing chemistry. The top model levels, where dynamics isn't fully resolved,
act as a dynamical sponge where species' concentrations are overwritten. The time dependent
chemical reactions are integrated forward in time using the iterative Newton Raphson solver
described in Essenturk et al. (2018), with a time step of 60 minutes throughout the atmosphere.
**Table 1.** List of chemical species in UKCA StratTrop. Species in italics are not advected tracers but
are calculated using a steady state approximation. Species in bold are set as constant values
throughout the atmosphere. †The molecular mass of Sec_Org is set to 150 g/mol.

| Name | Formula | Dry Deposited | Wet Deposited | Emitted or LBC |
|---|---|---|---|---|
| O(3P) | $O(^3P)$ | No | No | No |
| *O(1D)* | *$O(^1D)$* | No | No | No |
| O3 | $O_3$ | Yes | No | No |
| N | N | No | No | No |
| NO | NO | Yes | No | Emitted |
| NO3 | $NO_3$ | Yes | Yes | No |
| NO2 | $NO_2$ | Yes | No | No |
| N2O5 | $N_2O_5$ | Yes | Yes | No |
| HO2NO2 | $HO_2NO_2$ | Yes | Yes | No |
| HONO2 | $HONO_2$ | Yes | Yes | No |
| H2O2 | $H_2O_2$ | Yes | Yes | No |
| CH4 | $CH_4$ | No | No | LBC |
| CO | CO | Yes | No | Emitted |
| HCHO | HCHO | Yes | Yes | Emitted |
| MeOO | $CH_3OO$ | No | Yes | No |
| MeOOH | $CH_3OOH$ | Yes | Yes | No |
| H | H | No | No | No |
| H2O | $H_2O$ | No | No | No |
| OH | OH | No | No | No |
| HO2 | $HO_2$ | No | Yes | No |
| Cl | Cl | No | No | No |
| Cl2O2 | $Cl_2O_2$ | No | No | No |
| ClO | ClO | No | No | No |
| OClO | OClO | No | No | No |
| Br | Br | No | No | No |
| BrO | BrO | No | No | No |
| BrCl | BrCl | No | No | No |





| | | | | |
|---|---|---|---|---|
| BrONO2 | BrONO$_2$ | No | Yes | No |
| N2O | N$_2$O | No | No | LBC |
| HCl | HCl | Yes | Yes | No |
| HOCl | HOCl | Yes | Yes | No |
| HBr | HBr | Yes | Yes | No |
| HOBr | HOBr | Yes | Yes | No |
| ClONO2 | ClONO$_2$ | No | Yes | No |
| CFCl3 | CFCl$_3$ | No | No | LBC |
| CF2Cl2 | CF$_2$Cl$_2$ | No | No | LBC |
| MeBr | CH$_3$Br | No | No | LBC |
| HONO | HONO | Yes | Yes | No |
| C2H6 | C$_2$H$_6$ | No | No | Emitted |
| EtOO | C$_2$H$_5$OO | No | No | No |
| EtOOH | C$_2$H$_5$OOH | Yes | Yes | No |
| MeCHO | CH$_3$CHO | Yes | No | Emitted |
| MeCO3 | CH$_3$C(O)OO | No | No | No |
| PAN | PAN | Yes | No | No |
| C3H8 | C$_3$H$_8$ | No | No | Emitted |
| n-PrOO | C$_3$H$_7$OO | No | No | No |
| i-PrOO | CH$_3$CH(OO)CH$_3$ | No | No | No |
| n-PrOOH | C$_3$H$_7$OOH | Yes | Yes | No |
| i-PrOOH | CH$_3$CH(OOH)CH$_3$ | Yes | Yes | No |
| EtCHO | C$_2$H$_5$CHO | Yes | No | No |
| EtCO3 | C$_2$H$_5$C(O)OO | No | No | No |
| Me2CO | CH$_3$C(O)CH$_3$ | No | No | Emitted |
| MeCOCH2OO | CH$_3$C(O)CH$_2$OO | No | No | No |
| MeCOCH2OOH | CH$_3$C(O)CH$_2$OOH | Yes | Yes | No |
| PPAN | PPAN | Yes | No | No |
| MeONO2 | MeONO$_2$ | No | No | No |
| C5H8 | C$_5$H$_8$ | No | No | Emitted |
| ISO2 | HOC$_5$H$_8$OO | No | No | No |
| ISOOH | HOC$_5$H$_8$OOH | Yes | Yes | No |
| ISON | ISON | Yes | Yes | No |
| MACR | C$_4$H$_6$O | Yes | No | No |
| MACRO2 | C$_4$H$_6$O(OO) | No | No | No |
| MACROOH | C$_4$H$_6$O(OOH) | Yes | Yes | No |
| MPAN | MPAN | Yes | No | No |
| HACET | CH$_3$C(O)CH$_2$OH | Yes | Yes | No |
| MGLY | CH$_3$COCHHO | Yes | Yes | No |
| NALD | NALD | Yes | No | No |
| HCOOH | HC(O)OH | Yes | Yes | No |
| MeCO3H | CH$_3$C(O)OOH | Yes | Yes | No |
| MeCO2H | CH$_3$C(O)OH | Yes | Yes | No |
| H2 | H$_2$ | No | No | LBC |
| MeOH | CH$_3$OH | Yes | Yes | Emitted |



| | | | | |
|---|---|---|---|---|
| **CO2** | **$CO_2$** | No | No | No |
| **O2** | **$O_2$** | No | No | No |
| **N2** | **$N_2$** | No | No | No |
| DMS | $CH_3SCH_3$ | No | No | Emitted |
| SO2 | $SO_2$ | Yes | Yes | Emitted |
| H2SO4 | $H_2SO_4$ | Yes | No | No |
| MSA | MSA | No | No | No |
| DMSO | DMSO | Yes | Yes | No |
| COS | COS | No | No | Emitted |
| SO3 | $SO_3$ | No | No | No |
| Monoterp | $C_{10}H_{16}$ | Yes | No | Emitted |
| Sec_Org | †Sec_Org | Yes | Yes | No |

**Table 2.** List of bi-molecular reactions in UKCA-StratTrop. Reactions with a † next to them have additional code to account for branching of reaction as, for example, a function of pressure (CO+OH) or $H_2O$ ($HO_2 + HO_2$). The temperature dependent rate coefficient can be calculated for each grid box at each chemistry timestep through: $k_{(T)} = A \left(\frac{T}{300}\right)^\alpha exp\left(\frac{-Ea}{RT}\right)$ where T refers to the grid box temperature (K).

| Reactants | Products | A | α | -Ea/R |
|---|---|---|---|---|
| $Br + Cl_2O_2$ | $BrCl + Cl + O_2$ | 5.90E-12 | 0 | 170 |
| Br + HCHO | $HBr + CO + HO_2$ | 1.70E-11 | 0 | 800 |
| $Br + HO_2$ | $HBr + O_2$ | 4.80E-12 | 0 | 310 |
| $Br + O_3$ | $BrO + O_2$ | 1.60E-11 | 0 | 780 |
| Br + OClO | BrO + ClO | 2.60E-11 | 0 | 1300 |
| BrO + BrO | $Br + Br + O_2$ | 2.40E-12 | 0 | -40 |
| BrO + ClO | $Br + Cl + O_2$ | 2.30E-12 | 0 | -260 |
| BrO + ClO | Br + OClO | 9.50E-13 | 0 | -550 |
| BrO + ClO | $BrCl + O_2$ | 4.10E-13 | 0 | -290 |
| $BrO + HO_2$ | $HOBr + O_2$ | 4.50E-12 | 0 | -460 |
| BrO + NO | $Br + NO_2$ | 8.80E-12 | 0 | -260 |
| BrO + OH | $Br + HO_2$ | 1.70E-11 | 0 | -250 |
| $CF_2Cl_2 + O(^1D)$ | Cl + ClO | 1.40E-10 | 0 | 0 |
| $CFCl_3 + O(^1D)$ | Cl + Cl + ClO | 2.30E-10 | 0 | 0 |
| $Cl + CH_4$ | HCl + MeOO | 7.30E-12 | 0 | 1280 |
| $Cl + Cl_2O_2$ | Cl + Cl + Cl | 7.60E-11 | 0 | -65 |
| $Cl + ClONO_2$ | $Cl + Cl + NO_3$ | 6.50E-12 | 0 | -135 |
| $Cl + H_2$ | HCl + H | 3.05E-11 | 0 | 2270 |
| $Cl + H_2O_2$ | $HCl + HO_2$ | 1.10E-11 | 0 | 980 |
| Cl + HCHO | $HCl + CO + HO_2$ | 8.10E-11 | 0 | 30 |
| $Cl + HO_2$ | ClO + OH | 3.65E-11 | 0 | 375 |
| $Cl + HO_2$ | $HCl + O_2$ | 1.40E-11 | 0 | -270 |
| Cl + HOCl | Cl + Cl + OH | 3.40E-12 | 0 | 130 |
| $Cl + NO_3$ | $ClO + NO_2$ | 2.40E-11 | 0 | 0 |
| $Cl + O_3$ | $ClO + O_2$ | 2.30E-11 | 0 | 200 |
| Cl + OClO | ClO + ClO | 3.40E-11 | 0 | -160 |



| | | | | |
|---|---|---|---|---|
| Cl + MeOOH | HCl + MeOO | 5.70E-11 | 0 | 0 |
| ClO + ClO | Cl + Cl + $O_2$ | 1.00E-12 | 0 | 1590 |
| ClO + ClO | Cl + Cl + $O_2$ | 3.00E-11 | 0 | 2450 |
| ClO + ClO | Cl + OClO | 3.50E-13 | 0 | 1370 |
| ClO + $HO_2$ | HOCl + $O_2$ | 2.60E-12 | 0 | -290 |
| ClO + MeOO | Cl + HCHO + $HO_2$ | 3.30E-12 | 0 | 115 |
| ClO + NO | Cl + $NO_2$ | 6.40E-12 | 0 | -290 |
| ClO + $NO_3$ | Cl + $O_2$ + $NO_2$ | 4.60E-13 | 0 | 0 |
| EtCO3 + NO | EtOO + $CO_2$ + $NO_2$ | 6.70E-12 | 0 | -340 |
| EtCO3 + $NO_3$ | EtOO + $CO_2$ + $NO_2$ | 4.00E-12 | 0 | 0 |
| EtOO + MeCO3 | MeCHO + $HO_2$ + MeOO | 4.40E-13 | 0 | -1070 |
| EtOO + NO | MeCHO + $HO_2$ + $NO_2$ | 2.55E-12 | 0 | -380 |
| EtOO + $NO_3$ | MeCHO + $HO_2$ + $NO_2$ | 2.30E-12 | 0 | 0 |
| H + $HO_2$ | $H_2$ + $O_2$ | 6.90E-12 | 0 | 0 |
| H + $HO_2$ | $O(^3P)$ + $H_2O$ | 1.62E-12 | 0 | 0 |
| H + $HO_2$ | OH + OH | 7.20E-11 | 0 | 0 |
| H + $NO_2$ | OH + NO | 4.00E-10 | 0 | 340 |
| H + $O_3$ | OH + $O_2$ | 1.40E-10 | 0 | 470 |
| $HO_2$ + $HO_2$† | $H_2O_2$ | 3.00E-13 | 0 | -460 |
| $HO_2$ + MeOO† | MeOOH | 3.80E-13 | 0 | -780 |
| $HO_2$ + NO | OH + $NO_2$ | 3.30E-12 | 0 | -270 |
| $HO_2$ + $NO_3$ | OH + $NO_2$ + $O_2$ | 3.50E-12 | 0 | 0 |
| $HO_2$ + $O_3$ | OH + $O_2$ + $O_2$ | 2.03E-16 | 4.57 | -693 |
| $HO_2$ + EtCO3 | $O_2$ + EtCO3H | 4.40E-13 | 0 | -980 |
| $HO_2$ + EtCO3 | $O_3$ + EtCO2H | 7.80E-14 | 0 | -980 |
| $HO_2$ + EtOO | EtOOH | 6.40E-13 | 0 | -710 |
| $HO_2$ + ISO2 | ISOOH | 2.05E-13 | 0 | -1300 |
| $HO_2$ + MACRO2 | MACROOH | 1.82E-13 | 0 | -1300 |
| $HO_2$ + MeCO3 | MeCO2H + $O_3$ | 7.80E-14 | 0 | -980 |
| $HO_2$ + MeCO3 | MeCO3H | 2.13E-13 | 0 | -980 |
| $HO_2$ + MeCO3 | OH + MeOO | 2.29E-13 | 0 | -980 |
| $HO_2$ + $MeCOCH_2OO$ | $MeCOCH_2OOH$ | 9.00E-12 | 0 | 0 |
| $HO_2$ + MeOO† | HCHO | 3.80E-13 | 0 | -780 |
| $HO_2$ + i-PrOO | i-PrOOH | 1.51E-13 | 0 | -1300 |
| $HO_2$ + n-PrOO | n-PrOOH | 1.51E-13 | 0 | -1300 |
| i-PrOO + NO | Me2CO + $HO_2$ + $NO_2$ | 2.70E-12 | 0 | -360 |
| i-PrOO + $NO_3$ | Me2CO + $HO_2$ + $NO_2$ | 2.70E-12 | 0 | -360 |
| ISO2 + ISO2 | MACR + MACR + HCHO + $HO_2$ | 2.00E-12 | 0 | 0 |
| MACRO2 + MACRO2 | HACET + MGLY + 0.5*HCHO + 0.5*CO + $HO_2$ | 2.00E-12 | 0 | 0 |
| MeBr + Cl | Br + HCl | 1.40E-11 | 0 | 1030 |
| MeBr + $O(^1D)$ | Br + OH | 1.80E-10 | 0 | 0 |
| MeBr + OH | Br + $H_2O$ | 2.35E-12 | 0 | 1300 |
| MeCO3 + NO | MeOO + $CO_2$ + $NO_2$ | 7.50E-12 | 0 | -290 |
| MeCO3 + $NO_3$ | MeOO + $CO_2$ + $NO_2$ | 4.00E-12 | 0 | 0 |



| | | | | |
|---|---|---|---|---|
| MeCOCH$_2$OO + NO | MeCO3 + HCHO + NO$_2$ | 2.70E-12 | 0 | -360 |
| MeCOCH$_2$OO + NO$_3$ | MeCO3 + HCHO + NO$_2$ | 2.30E-12 | 0 | 0 |
| MeOO + NO | HO$_2$ + HCHO + NO$_2$ | 2.30E-12 | 0 | -360 |
| MeOO + MeOO† | HO$_2$ + HO$_2$ + HCHO + HCHO | 1.03E-13 | 0 | -365 |
| MeOO + MeCO3 | HO$_2$ + HCHO + MeOO | 1.80E-12 | 0 | -500 |
| MeOO + MeCO3 | MeCO2H + HCHO | 2.00E-13 | 0 | -500 |
| MeOO + MeOO† | MeOH + HCHO | 1.03E-13 | 0 | -365 |
| MeOO + NO | MeONO$_2$ | 2.30E-15 | 0 | -360 |
| MeOO + NO$_3$ | HO$_2$ + HCHO + NO$_2$ | 1.20E-12 | 0 | 0 |
| N + NO | N$_2$ + O($^3$P) | 2.10E-11 | 0 | -100 |
| N + NO$_2$ | N$_2$O + O($^3$P) | 5.80E-12 | 0 | -220 |
| N + O$_2$ | NO + O($^3$P) | 1.50E-11 | 0 | 3600 |
| n-PrOO + NO | EtCHO + HO$_2$ + NO$_2$ | 2.90E-12 | 0 | -350 |
| n-PrOO + NO$_3$ | EtCHO + HO$_2$ + NO$_2$ | 2.70E-12 | 0 | -360 |
| N$_2$O$_5$ + H$_2$O | HONO$_2$ + HONO$_2$ | 2.50E-22 | 0 | 0 |
| NO + NO$_3$ | NO$_2$ + NO$_2$ | 1.50E-11 | 0 | -170 |
| NO + O$_3$ | NO$_2$ | 3.00E-12 | 0 | 1500 |
| NO + ISO2 | ISON | 1.12E-13 | 0 | -360 |
| NO + ISO2 | NO$_2$ + MACR + HCHO + HO$_2$ | 2.43E-12 | 0 | -360 |
| NO + MACRO2 | NO$_2$ + 0.25*MeCO3 + 0.25*HACET + 0.25*CO + 0.5*MGLY + 0.75*HCHO + 0.75*HO$_2$ | 2.54E-12 | 0 | -360 |
| NO$_2$ + NO$_3$ | NO + NO$_2$ + O$_2$ | 4.50E-14 | 0 | 1260 |
| NO$_2$ + O$_3$ | NO$_3$ | 1.20E-13 | 0 | 2450 |
| NO$_3$ + Br | BrO + NO$_2$ | 1.60E-11 | 0 | 0 |
| NO$_3$ + HCHO | HONO$_2$ + HO$_2$ + CO | 2.00E-12 | 0 | 2440 |
| NO$_3$ + C$_5$H$_8$ | ISON | 3.15E-12 | 0 | 450 |
| NO$_3$ + EtCHO | HONO$_2$ + EtCO3 | 6.30E-15 | 0 | 0 |
| NO$_3$ + MGLY | MeCO3 + CO + HONO$_2$ | 3.36E-12 | 0 | 1860 |
| NO$_3$ + Me2CO | HONO$_2$ + MeCOCH$_2$OO | 3.00E-17 | 0 | 0 |
| NO$_3$ + MeCHO | HONO$_2$ + MeCO3 | 1.40E-12 | 0 | 1860 |
| O($^1$D) + CH$_4$ | HCHO + H$_2$ | 9.00E-12 | 0 | 0 |
| O($^1$D) + CH$_4$ | OH + MeOO | 1.31E-10 | 0 | 0 |
| O($^1$D) + CO$_2$ | O($^3$P) + CO$_2$ | 7.50E-11 | 0 | -115 |
| O($^1$D) + H$_2$ | OH + H | 1.20E-10 | 0 | 0 |
| O($^1$D) + H$_2$O | OH + OH | 1.63E-10 | 0 | -60 |
| O($^1$D) + HBr | HBr + O($^3$P) | 3.00E-11 | 0 | 0 |
| O($^1$D) + HBr | OH + Br | 1.20E-10 | 0 | 0 |
| O($^1$D) + HCl | H + ClO | 3.60E-11 | 0 | 0 |
| O($^1$D) + HCl | O($^3$P) + HCl | 1.35E-11 | 0 | 0 |
| O($^1$D) + HCl | OH + Cl | 1.01E-10 | 0 | 0 |
| O($^1$D) + N$_2$ | O($^3$P) + N$_2$ | 2.15E-11 | 0 | -110 |
| O($^1$D) + N$_2$O | N$_2$ + O$_2$ | 4.60E-11 | 0 | -20 |
| O($^1$D) + N$_2$O | NO + NO | 7.30E-11 | 0 | -20 |





| | | | | |
|---|---|---|---|---|
| $O(^1D) + O_2$ | $O(^3P) + O_2$ | 3.30E-11 | 0 | -55 |
| $O(^1D) + O_3$ | $O_2 + O(^3P) + O(^3P)$ | 1.20E-10 | 0 | 0 |
| $O(^1D) + O_3$ | $O_2 + O_2$ | 1.20E-10 | 0 | 0 |
| $O(^1D) + CH_4$ | $HCHO + HO_2 + HO_2$ | 3.45E-11 | 0 | 0 |
| $O(^3P) + BrO$ | $O_2 + Br$ | 1.90E-11 | 0 | -230 |
| $O(^3P) + ClO$ | $Cl + O_2$ | 2.80E-11 | 0 | -85 |
| $O(^3P) + ClONO_2$ | $ClO + NO_3$ | 3.60E-12 | 0 | 840 |
| $O(^3P) + H_2$ | $OH + H$ | 9.00E-18 | 0 | 0 |
| $O(^3P) + H_2O_2$ | $OH + HO_2$ | 1.40E-12 | 0 | 2000 |
| $O(^3P) + HBr$ | $OH + Br$ | 5.80E-12 | 0 | 1500 |
| $O(^3P) + HCHO$ | $OH + CO + HO_2$ | 3.40E-11 | 0 | 1600 |
| $O(^3P) + HCl$ | $OH + Cl$ | 1.00E-11 | 0 | 3300 |
| $O(^3P) + HO_2$ | $OH + O_2$ | 2.70E-11 | 0 | -224 |
| $O(^3P) + HOCl$ | $OH + ClO$ | 1.70E-13 | 0 | 0 |
| $O(^3P) + NO_2$ | $NO + O_2$ | 5.10E-12 | 0 | -210 |
| $O(^3P) + NO_3$ | $O_2 + NO_2$ | 1.70E-11 | 0 | 0 |
| $O(^3P) + O_3$ | $O_2 + O_2$ | 8.00E-12 | 0 | 2060 |
| $O(^3P) + OClO$ | $O_2 + ClO$ | 2.40E-12 | 0 | 960 |
| $O(^3P) + OH$ | $O_2 + H$ | 1.80E-11 | 0 | -180 |
| $O_3 + C_5H_8$ | 0.25*$HO_2$ + 0.25*OH + 0.65*MACR + 0.58*HCHO + 0.1*MACRO2 + 0.1*MeCO3 + 0.08*MeOO + 0.28*HCOOH + 0.14*CO + 0.09*$H_2O_2$ | 9.99E-15 | 0 | 1995 |
| $O_3 + MACR$ | 0.9*MGLY + 0.45*HCOOH + 0.32*$HO_2$ + 0.22*CO + 0.19*OH + 0.1*MeCO3 | 4.26E-16 | 0 | 1520 |
| $O_3 + MACR$ | 0.9*MGLY + 0.45*HCOOH + 0.32*$HO_2$ + 0.22*CO + 0.19*OH + 0.1*MeCO3 | 7.00E-16 | 0 | 2100 |
| $OClO + NO$ | $NO_2 + ClO$ | 2.50E-12 | 0 | 600 |
| $OH + CH_4$ | $H_2O + MeOO$ | 2.45E-12 | 0 | 1775 |
| $OH + CO†$ | $H + CO_2$ | 1.44E-13 | 0 | 0 |
| $OH + ClO$ | $HCl + O_2$ | 6.00E-13 | 0 | -230 |
| $OH + ClO$ | $HO_2 + Cl$ | 7.40E-12 | 0 | -270 |
| $OH + ClONO_2$ | $HOCl + NO_3$ | 1.20E-12 | 0 | 330 |
| $OH + H_2$ | $H_2O + H$ | 2.80E-12 | 0 | 1800 |
| $OH + HBr$ | $H_2O + Br$ | 5.50E-12 | 0 | -200 |
| $OH + HCHO$ | $H_2O + HO_2 + CO$ | 5.40E-12 | 0 | -135 |
| $OH + HCl$ | $H_2O + Cl$ | 1.80E-12 | 0 | 250 |
| $OH + HO_2$ | $H_2O + O_2$ | 4.80E-11 | 0 | -250 |
| $OH + H_2O_2$ | $HO_2 + H_2O$ | 2.90E-12 | 0 | 160 |
| $OH + HO_2NO_2$ | $H_2O + NO_2 + O_2$ | 3.20E-13 | 0 | -690 |
| $OH + HOCl$ | $ClO + H_2O$ | 3.00E-12 | 0 | 500 |
| $OH + HONO_2†$ | $H_2O + NO_3$ | 2.40E-14 | 0 | -460 |
| $OH + MeOOH$ | $H_2O + MeOO$ | 1.89E-12 | 0 | -190 |
| $OH + NO_3$ | $HO_2 + NO_2$ | 2.20E-11 | 0 | 0 |
| $OH + O_3$ | $HO_2 + O_2$ | 1.70E-12 | 0 | 940 |



| | | | | |
|---|---|---|---|---|
| OH + OClO | HOCl + $O_2$ | 1.40E-12 | 0 | -600 |
| OH + OH | $H_2O$ + $O(^3P)$ | 6.31E-14 | 2.6 | -945 |
| OH + $C_2H_6$ | $H_2O$ + EtOO | 6.90E-12 | 0 | 1000 |
| OH + $C_3H_8$† | i-PrOO + $H_2O$ | 7.60E-12 | 0 | 585 |
| OH + $C_3H_8$† | n-PrOO + $H_2O$ | 7.60E-12 | 0 | 585 |
| OH + $C_5H_8$ | ISO2 | 2.70E-11 | 0 | -390 |
| OH + EtCHO | $H_2O$ + EtCO3 | 4.90E-12 | 0 | -405 |
| OH + EtOOH | $H_2O$ + EtOO | 1.90E-12 | 0 | -190 |
| OH + EtOOH | $H_2O$ + MeCHO + OH | 8.01E-12 | 0 | 0 |
| OH + HACET | MGLY + $HO_2$ | 1.60E-12 | 0 | -305 |
| OH + HCOOH | $HO_2$ | 4.50E-13 | 0 | 0 |
| OH + HONO | $H_2O$ + $NO_2$ | 2.50E-12 | 0 | -260 |
| OH + ISON | HACET + NALD | 1.30E-11 | 0 | 0 |
| OH + ISOOH | MACR + OH | 1.00E-10 | 0 | 0 |
| OH + MACR | MACRO2 | 1.30E-12 | 0 | -610 |
| OH + MACR | MACRO2 | 4.00E-12 | 0 | -380 |
| OH + MACROOH | MACRO2 | 3.77E-11 | 0 | 0 |
| OH + MGLY | MeCO3 + CO | 1.90E-12 | 0 | -575 |
| OH + MPAN | HACET + $NO_2$ | 2.90E-11 | 0 | 0 |
| OH + Me2CO | $H_2O$ + $MeCOCH_2OO$ | 1.70E-14 | 0 | -423 |
| OH + Me2CO | $H_2O$ + $MeCOCH_2OO$ | 8.80E-12 | 0 | 1320 |
| OH + MeCHO | $H_2O$ + MeCO3 | 4.70E-12 | 0 | -345 |
| OH + MeCO2H | MeOO | 8.00E-13 | 0 | 0 |
| OH + MeCO3H | MeCO3 | 3.70E-12 | 0 | 0 |
| OH + MeCOCH2OOH | $H_2O$ + $MeCOCH_2OO$ | 1.90E-12 | 0 | -190 |
| OH + MeCOCH2OOH | OH + MGLY | 8.39E-12 | 0 | 0 |
| OH + MeOH | $HO_2$ + HCHO | 2.85E-12 | 0 | 345 |
| OH + $MeONO_2$ | HCHO + $NO_2$ + $H_2O$ | 4.00E-13 | 0 | 845 |
| OH + MeOOH | $H_2O$ + HCHO + OH | 2.12E-12 | 0 | -190 |
| OH + NALD | HCHO + CO + $NO_2$ | 4.70E-12 | 0 | -345 |
| OH + PAN | HCHO + $NO_2$ + $H_2O$ | 3.00E-14 | 0 | 0 |
| OH + PPAN | MeCHO + $NO_2$ + $H_2O$ | 1.27E-12 | 0 | 0 |
| OH + i-PrOOH | Me2CO + OH | 1.66E-11 | 0 | 0 |
| OH + i-PrOOH | i-PrOO + $H_2O$ | 1.90E-12 | 0 | -190 |
| OH + n-PrOOH | EtCHO + $H_2O$ + OH | 1.10E-11 | 0 | 0 |
| OH + n-PrOOH | n-PrOO + $H_2O$ | 1.90E-12 | 0 | -190 |
| DMS + OH | $SO_2$ | 1.20E-11 | 0 | 260 |
| DMS + OH | MSA + $SO_2$ | 3.04E-12 | 0 | -350 |
| DMS + $NO_3$ | $SO_2$ | 1.90E-13 | 0 | -500 |
| DMS + $O(^3P)$ | $SO_2$ | 1.30E-11 | 0 | -410 |
| COS + $O(^3P)$ | CO + $SO_2$ | 2.10E-11 | 0 | 2200 |
| COS + OH | $CO_2$ + $SO_2$ | 1.10E-13 | 0 | 1200 |
| $SO_2$ + $O_3$ | $SO_3$ | 3.00E-12 | 0 | 7000 |
| $SO_3$ + $H_2O$ | $H_2SO_4$ + $H_2O$ | 8.50E-41 | 0 | -6540 |
| Monoterp + OH | 0.13*Sec_Org | 1.20E-11 | 0 | -444 |



| | | | | | |
|---|---|---|---|---|---|
| Monoterp + $O_3$ | 0.13*Sec_Org | 1.01E-15 | 0 | 732 | |
| Monoterp + $NO_3$ | 0.13*Sec_Org | 1.19E-12 | 0 | -925 | |

2 **Table 3:** Termolecular reactions used in UKCA StratTrop as implemented in UKESM1.

| Reactants | Products | Fc | $k_1$ | $\alpha_1$ | $\beta_1$ | $k_2$ | $\alpha_2$ | $\beta_2$ |
|---|---|---|---|---|---|---|---|---|
| $O(^3P) + O_2$ | $O_3$ | 0.00 | 6.00E-34 | -2.5 | 0 | 0.00E+00 | 0 | 0 |
| $O(^3P) + NO$ | $NO_2$ | 0.60 | 9.00E-32 | -1.5 | 0 | 3.00E-11 | 0 | 0 |
| $O(^3P) + NO_2$ | $NO_3$ | 0.60 | 2.50E-31 | -1.8 | 0 | 2.20E-11 | -0.7 | 0 |
| $O(^1D) + N_2$ | $N_2O$ | 0.00 | 2.80E-36 | -0.9 | 0 | 0.00E+00 | 0 | 0 |
| $BrO + NO_2$ | $BrONO_2$ | 0.60 | 5.20E-31 | -3.2 | 0 | 6.90E-12 | 0 | 0 |
| $ClO + ClO$ | $Cl_2O_2$ | 0.60 | 1.60E-32 | -4.5 | 0 | 3.00E-12 | -2 | 0 |
| $Cl_2O_2$ | $ClO + ClO$ | 0.45 | 3.70E-07 | 0.0 | 7690 | 1.80E+14 | 0 | 7690 |
| $ClO + NO_2$ | $ClONO_2$ | 0.60 | 1.80E-31 | -3.4 | 0 | 1.50E-11 | 0 | 0 |
| $H + O_2$ | $HO_2$ | 0.60 | 4.40E-32 | -1.3 | 0 | 7.50E-11 | 0 | 0 |
| $HO_2 + HO_2$[†] | $H_2O_2 + O_2$ | 0.00 | 2.10E-33 | 0.0 | -920 | 0.00E+00 | 0 | 0 |
| $HO_2 + NO_2$ | $HO_2NO_2$ | 0.60 | 2.00E-31 | -3.4 | 0 | 2.90E-12 | 0 | 0 |
| $HO_2NO_2$ | $HO_2 + NO_2$ | 0.50 | 4.10E-05 | 0.0 | 10650 | 4.80E+15 | 0 | 11170 |
| $OH + NO$ | $HONO$ | 0.60 | 7.00E-31 | -2.6 | 0 | 3.60E-11 | -0.1 | 0 |
| $OH + NO_2$ | $HONO_2$ | 0.60 | 1.80E-30 | -3.0 | 0 | 2.80E-11 | 0 | 0 |
| $OH + OH$ | $H_2O_2$ | 0.60 | 6.90E-31 | -1.0 | 0 | 2.60E-11 | 0 | 0 |
| $MeCO_3 + NO_2$ | PAN | 0.30 | 2.70E-28 | -7.1 | 0 | 1.20E-11 | -0.9 | 0 |
| PAN | $MeCO3 + NO_2$ | 0.30 | 4.90E-03 | 0.0 | 12100 | 5.40E+16 | 0 | 13830 |
| $EtCO3 + NO_2$ | PPAN | 0.30 | 2.70E-28 | -7.1 | 0 | 1.20E-11 | -0.9 | 0 |
| PPAN | $EtCO3 + NO_2$ | 0.30 | 4.90E-03 | 0.0 | 12100 | 5.40E+16 | 0 | 13830 |
| $MACRO2 + NO_2$ | MPAN | 0.30 | 2.70E-28 | -7.1 | 0 | 1.20E-11 | -0.9 | 0 |
| MPAN | $MACRO2 + NO_2$ | 0.30 | 4.90E-03 | 0.0 | 12100 | 5.40E+16 | 0 | 13830 |
| $NO_2 + NO_3$ | $N_2O_5$ | 0.35 | 3.60E-30 | -4.1 | 0 | 1.90E-12 | 0.2 | 0 |
| $N_2O_5 + M$ | $NO_2 + NO_3$ | 0.35 | 1.30E-03 | -3.5 | 11000 | 9.70E+14 | 0.1 | 11080 |
| $NO + NO$ | $NO_2 + NO_2$ | 0.00 | 3.30E-39 | 0.0 | -530 | 0.00E+00 | 0 | 0 |
| $SO_2 + OH$ | $SO_3 + HO_2$ | 0.60 | 3.00E-31 | -3.3 | 0 | 1.50E-12 | 0 | 0 |

$k_0 = k_1 \times (T/300)^{\alpha_1} \times \exp(-\beta_1/T)$
$k_\infty = k_2 \times (T/300)^{\alpha_2} \times \exp(-\beta_2/T)$

$$k_{([M],T)} = \left( \frac{k_0[M]}{1 + \frac{k_0[M]}{k_\infty}} \right) F_c^{\left\{ 1 + \left[ log_{10}\left(\frac{k_0[M]}{k_\infty}\right)\right]^2 \right\}^{-1}}$$

[†]Indicates that extra code is used to account for water dependence of this reaction.
M is used to represent a third body (calculated from the grid box pressure and temperature).



**Table 4.** Photodissociation reactions used in UKCA-StratTrop. For details of the cross-section and
quantum yield information please see Telford et al. (2013).

| Reactants | Products |
| --- | --- |
| $BrCl + hv$ | $Br + Cl$ |
| $BrO + hv$ | $Br + O(^3P)$ |
| $BrONO_2 + hv$ | $Br + NO_3$ |
| $BrONO_2 + hv$ | $BrO + NO_2$ |
| $CF_2Cl_2 + hv$ | $Cl + Cl$ |
| $CFCl_3 + hv$ | $Cl + Cl + Cl$ |
| $CH_4 + hv$ | $MeOO + H$ |
| $Cl_2O_2 + hv$ | $Cl + Cl + O_2$ |
| $ClONO_2 + hv$ | $Cl + NO_3$ |
| $ClONO_2 + hv$ | $ClO + NO_2$ |
| $CO_2 + hv$ | $CO + O(^3P)$ |
| $COS + hv$ | $CO + SO_2$ |
| $EtCHO + hv$ | $EtOO + HO_2 + CO$ |
| $EtOOH + hv$ | $MeCHO + HO_2 + OH$ |
| $H_2O + hv$ | $OH + H$ |
| $H_2O_2 + hv$ | $OH + OH$ |
| $H_2SO_4 + hv$ | $SO_3 + OH$ |
| $HACET + hv$ | $MeCO_3 + HCHO + HO_2$ |
| $HCHO + hv$ | $HO_2 + HO_2 + CO$ |
| $HCHO + hv$ | $H_2 + CO$ |
| $HCl + hv$ | $H + Cl$ |
| $HO_2NO_2 + hv$ | $HO_2 + NO_2$ |
| $HO_2NO_2 + hv$ | $OH + NO_3$ |
| $HOBr + hv$ | $OH + Br$ |
| $HOCl + hv$ | $OH + Cl$ |
| $HONO + hv$ | $OH + NO$ |
| $HONO_2 + hv$ | $OH + NO_2$ |
| $i\text{-}PrOOH + hv$ | $Me2CO + HO_2 + OH$ |
| $ISON + hv$ | $NO_2 + MACR + HCHO + HO_2$ |
| $ISOOH + hv$ | $OH + MACR + HCHO + HO_2$ |
| $MACR + hv$ | $MeCO3 + HCHO + CO + HO_2$ |
| $MACROOH + hv$ | $OH + HO_2 + OH + HO_2$ |
| $MACROOH + hv$ | $HACET + CO + MGLY + HCHO$ |
| $Me2CO + hv$ | $MeCO3 + MeOO$ |
| $MeBr + hv$ | $Br + H$ |
| $MeCHO + hv$ | $MeOO + HO_2 + CO$ |
| $MeCHO + hv$ | $CH_4 + CO$ |
| $MeCO3H + hv$ | $MeOO + OH$ |
| $MeCOCH_2OOH + hv$ | $MeCO3 + HCHO + OH$ |




| Reactants | Products |
|---|---|
| $MeONO_2$ + hv | $HO_2$ + HCHO + $NO_2$ |
| MeOOH + hv | $HO_2$ + HCHO + OH |
| MGLY + hv | MeCO3 + CO + $HO_2$ |
| MPAN + hv | MACRO2 + $NO_2$ |
| $N_2O$ + hv | $N_2$ + O($^1$D) |
| $N_2O_5$ + hv | $NO_2$ + $NO_3$ |
| NALD + hv | HCHO + CO + $NO_2$ + $HO_2$ |
| NO + hv | N + O($^3$P) |
| $NO_2$ + hv | NO + O($^3$P) |
| $NO_3$ + hv | NO + $O_2$ |
| $NO_3$ + hv | $NO_2$ + O($^3$P) |
| n-PrOOH + hv | EtCHO + $HO_2$ + OH |
| $O_2$ + hv | O($^3$P) + O($^3$P) |
| $O_2$ + hv | O($^3$P) + O($^1$D) |
| $O_3$ + hv | $O_2$ + O($^1$D) |
| $O_3$ + hv | $O_2$ + O($^3$P) |
| OClO + hv | O($^3$P) + ClO |
| PAN + hv | MeCO3 + $NO_2$ |
| PPAN + hv | $EtCO_3$ + $NO_2$ |
| $SO_3$ + hv | $SO_2$ + O($^3$P) |

**Table 5.** Heterogeneous reaction list used in UKCA StratTrop in UKESM1. Uptake coefficients are
denoted *f* when not constant; see Denison et al. (2018) for full references and formulation.

| Reactants | Products | Uptake coefficient (γ) | | |
|---|---|---|---|---|
| | | Liquid aerosol | Nat | Ice |
| $ClONO_2$ + HCl | Cl + Cl + $HONO_2$ | *f* | 0.3 | 0.3 |
| $ClONO_2$ + H2O | HOCl + $HONO_2$ | | 0.006 | 0.3 |
| $N_2O_5$ + $H_2O$ | $HONO_2$ + $HONO_2$ | 0.1 | 0.0006 | 0.03 |
| $N_2O_5$ + HCl | Cl + $NO_2$ + $HONO_2$ | | 0.003 | 0.03 |
| HOCl + HCl | Cl + Cl + $H_2O$ | *f* | 0.3 | 0.3 |



**Table 6.** Aqueous phase Sulfur cycle reactions used in UKCA StratTrop in UKESM1 (after
Kreidenweis et al (2003)).

| Reactants | Products | Rate expression /cm$^3$ molecule$^{-1}$ s$^{-1}$ |
|---|---|---|
| $HSO_3^-{}_{(aq)} + H_2O_{2(aq)}$ | $SO_4^{2-}{}_{(aq)}$ | 2.1295E+14*exp(-4430.0/T)*([H$^+$]/(1.0 + 13.0*[H$^+$])) |
| $HSO_3^-{}_{(aq)} + O_{3(aq)}$ | $SO_4^{2-}{}_{(aq)}$ | 4.0113E+13*exp(-5530.0/T) |
| $SO_3^{2-}{}_{(aq)} + O_{3(aq)}$ | $SO_4^{2-}{}_{(aq)}$ | 7.43E+16*exp(-5280.0/T) |

[H$^+$] is prescribed in UKCA StratTrop in UKESM1 at 1E-5 molecules cm$^{-3}$.
The stratospheric sulfate aerosol optical depth, used in the radiation scheme of MetUM, is
modified to be consistent with the aerosols seen in the heterogeneous chemistry which, by
default, are taken from a surface aerosol density climatology prepared for the CMIP6 model
intercomparison (Luo, personal communication). The surface aerosol density is converted to
mass mixing ratio, using a climatology of particle size (Thomason and Peter, 2006) and
assuming a density of 1700 kg/m3.
**2.3 Photolysis**
The most significant new development relative to MO09 and OC14 in the UKCA-StratTrop
scheme used in UKESM1 is the interactive Fast-JX photolysis scheme which is applied to
derive photolysis rates between 177 and 750 nm (Neu et al., 2007) as described in Telford et
al. (2013). This is an important new addition as it enables interactive treatment of photolysis
rates (key drivers for the photochemistry of the atmosphere) under changing climate and
atmospheric composition. For shorter wavelengths, relevant above 60 km, a correction is
added, to account for photolysis occurring between 112 and 177 nm, following Lary and Pyle
22 (1991).

In older versions of UKCA (i.e. MO09 and OC14) prealculated photolysis frequencies were
applied in the model by default. Sellar et al (2019) shows a comparison of these and we note
here that the switch from pre-calculated to on-line interactive photolysis calculations has had
a significant effect on shortening the model simulated methane lifetime and increasing the
tropospheric mean [OH] (Telford et al., 2013; O'Connor et al., 2014; Voulgarakis et al., 2009),
as shown in Figure 4.
**2.4 Dry deposition**
In UKCA the representation of dry deposition follows the resistance-in-series model as
described by Wesley (1989) in which the removal of material at the surface is described by
three resistances, $r_a$, $r_b$, and $r_c$. The deposition velocity $v_d$ (m s$^{-1}$) is then a function of these
three resistance terms according to:

$$v_d = \frac{1}{r_a + r_b + r_c},$$

where $r_a$ denotes the aerodynamic resistance to dry deposition, $r_b$ is the quasi-laminar
resistance term, and $r_c$ represents the resistance to uptake at the surface. Of these three terms
$r_c$ tends to be the most complex because it encompasses a variety of exchange fluxes, such
as stomatal and cuticular uptake, assimilation by soil microbes, etc. The uptake at the surface
also depends strongly on the presence of dew, rain, or snow which can interrupt the deposition
process altogether.
**2.4.1 Dry deposition of gas-phase species**
Surface dry deposition is calculated interactive at every time step for a number of atmospheric
gas-phase species (c.f., Table 1 for a list of deposited species). The aerodynamic resistance
$r_a$ is given by:
$r_a = \frac{ln(\frac{z}{z_0}) - \Psi}{k \times u^*},$
where $z_0$ is the roughness length, $\Psi$ denotes the Businger dimensionless stability function, $k$
is the von Karman constant, and $u^*$ is the friction velocity. $r_a$ represents the resistance to
turbulent mixing in the boundary layer and therefore depends crucially on the stability of the
boundary layer. It is independent of the chemical species that is deposited.
The quasi-laminar resistance $r_b$, on the other hand, depends on the chemical and physical
properties of the deposited species. It describes the transport through the thin, laminar layer
of air closest to the surface. Transport through this layer is diffusive due to the absence of
turbulent mixing.
The third resistance term $r_c$ depends on both the physico-chemical properties of the deposited
species and the properties and condition of the respective surface to which deposition occurs.
The surface can be anything from bare soil or rock to vegetation and even urban environments.
Surface uptake varies with season, time of day and current meteorological conditions. The
largest individual surface type is water in the form of the world's oceans. In this latter case
solubility obviously plays the key role (Hardacre et al., 2015; Luhar et al., 2017).
A particularly important surface uptake process is the deposition flux to the terrestrial
vegetation. In this case a number of pathways exist which are commonly integrated into the
so-called "Big-leaf" model  (Smith et al., 2000; Seinfeld and Pandis, 2006). Of all the deposition
pathways manifesting in vegetated regions, for most species the most important is uptake
through the stomata. Through these tiny pores in the leaf surface plants take up carbon dioxide
from the atmosphere and exchange water vapour and oxygen with it. This exchange also
includes all other species that make up the ambient air, including pollutants such as for
instance ozone. For this, the specific type of vegetation is crucial. Ozone deposition fluxes, for
instance, vary widely between forests and grasslands.
The calculation of the surface resistance term and land surface type information provided by
the dynamic vegetation model JULES (Best et al., 2011; Clark et al., 2011) is utilised in UKCA.
JULES forms part of the UKESM1 Earth system model and is thus coupled with UKCA. Within
JULES, various land surface type configurations may be selected. In the most simple
configuration, which was also used in the UKESM1 predecessor model HadGEM2-ES, any
land-based grid box at the surface can be subdivided into variable-sized fractions assigned to
any of 9 different surface types: broadleaf trees, needleleaf trees, C3 grasses, C4 grasses,



shrubs, bare spoil, rivers and lakes, urban environments and ice. Non-land grid boxes are
treated separately.
Since then, the number of land surface types in JULES has increased substantially (c.f. Harper
et al., 2018). Apart from the original 9-tile version (5 vegetation and 4 non-vegetation types),
13, 17, and also 27-tile configurations are now included. The upgrade to the 13-tile
configuration increases the number of vegetation types by introducing 3 broadleaf plant
functional types (PFTs), 2 needleleaf PFTS, and 2 shrub PFTS; the number of grass-related
PFTs as well as the number of non-vegetation type remains the same.in this configuration.
The 17-tile configuration further extends the number of PFTs by introducing 4 cropland types,
two C3-grass related and two C4-grass related PFTs; again, the number of non-vegetation
types remains the same. Finally, the 27-tile land surface configuration, corresponding to the
UKESM1 release configurations and the configurations used for this manuscript, introduces a
substantial number of additional land ice tiles. Each of these land surface and PFT tiles offers
a specific resistance to dry deposition of atmospheric gas-phase species.
For dry deposition of aerosols a slightly different treatment is taken to that described above
and we direct the reader to Mulchay et al. (2019) and references therein for more details.
**2.5 Wet deposition**
The wet deposition scheme employed in UKCA for the removal of tropospheric gas-phase
species through convective and stratiform precipitation is the same as that described in
O'Connor et al., 2014. The original scheme was implemented from the TOMCAT chemistry
transport model (CTM) where it previously had been validated by Giannakopoulos (1998) and
Giannakopoulos et al. (1999). In this paper we provide a brief description of the scheme but
will not present an evaluation because there have been no changes since the last published
version. For an in-depth performance evaluation in UKCA we refer to section 3.4 in O'Connor
et al. (2014).
Following a scheme originally developed by Walton et al. (1988) wet deposition is
parameterized as a first-order loss process which is calculated as a function of the three-
dimensional convective and stratiform precipitation. The climate model provides the required
precipitation activity to UKCA. The wet scavenging rate $r$ is calculated at every grid box and
time step according to:
$$r = S_j \times p_j(l)$$
where $S_j$ is the wet scavenging coefficient for precipitation type $j$ (cm$^{-1}$) and $p_j(l)$ is the
precipitation rate for type $j$ (convective or stratiform), provided by the climate model at model
level $l$ (cm h$^{-1}$).
Scavenging coefficients for nitric acid (HNO$_3$) of 2.4 cm$^{-1}$ and 4.7 cm$^{-1}$ for stratiform and
convective precipitation, respectively, are applied (c.f., Penner et al., 1991). These parameters
are scaled down for individual species using the fraction of each species in the aqueous phase,
$f_{aq}$, calculated by:



$$f_{aq} = \frac{L \times H_{eff} \times R \times T}{1 + L \times H_{eff} \times R \times T}$$

where $L$ represents the liquid water content, $R$ the universal gas constant, $T$ denotes ambient
temperature, and $H_{eff}$ is the effective Henry's Law constant for each species. $H_{eff}$ includes
the effects of solubility, dissociation, and complex formation. Tables 7, 8 and 9 summarise the
parameters used in the UKCA wet deposition scheme for each soluble species included in the
StratTrop chemical mechanism.
Furthermore, in the scheme precipitation only occurs over a fraction of the grid box. This
fraction is assumed to be 1.0 and 0.3 for stratiform and convective precipitation, respectively.
These fractions are applied in the calculation of the grid box mean wet scavenging rate for
both precipitation types after which point the two rates are added together.
**Table 7.** Values required to calculate the effective Henry's Law coefficient for the soluble tropospheric
species included in the UKCA strat-trop scheme, where Me=CH$_3$, Et=C$_2$H$_5$, Pr=C$_3$H$_7$.

| Species | Henry's Law Data | | Dissociation Data | |
|---------|------------------|--------|-------------------|--------|
|         | $K_H$(298 K)     | -Δ$H/R$ | $K_a$(298 K)     | -Δ$H/R$ |
|         | M atm$^{-1}$     | K$^{-1}$ | M               | K$^{-1}$ |
| NO$_3$        | 2.0E+00 | 2000.0 | 0.0E+00 | 0.0 |
| N$_2$O$_5$    | 2.1E+05 | 8700.0 | 2.0E+01 | 0.0 |
| HO$_2$NO$_2$  | 1.3E+04 | 6900.0 | 1.0E-05 | 0.0 |
| HONO$_2$      | 2.1E+05 | 8700.0 | 2.0E+01 | 0.0 |
| HO$_2$        | 4.0E+03 | 5900.0 | 2.0E-05 | 0.0 |
| H$_2$O$_2$    | 8.3E+04 | 7400.0 | 2.4E-12 | -3730.0 |
| HCHO          | 3.3E+03 | 6500.0 | 0.0E+00 | 0.0 |
| MeOO          | 2.0E+03 | 6600.0 | 0.0E+00 | 0.0 |
| MeOOH         | 3.1E+02 | 5000.0 | 0.0E+00 | 0.0 |
| HONO          | 5.0E+01 | 4900.0 | 5.6E-04 | -1260.0 |
| EtOOH         | 3.4E+02 | 5700.0 | 0.0E+00 | 0.0 |
| n-PrOOH       | 3.4E+02 | 5700.0 | 0.0E+00 | 0.0 |
| i-PrOOH       | 3.4E+02 | 5700.0 | 0.0E+00 | 0.0 |
| MeCOCH$_2$OOH | 3.4E+02 | 5700.0 | 0.0E+00 | 0.0 |
| ISOOH         | 1.7E+06 | 9700.0 | 0.0E+00 | 0.0 |





| | | | |
|---|---|---|---|
| ISON | 3.0E+03 | 7400.0 | 0.0E+00 | 0.0 |
| MACROOH | 1.7E+06 | 9700.0 | 0.0E+00 | 0.0 |
| HACET | 1.4E+02 | 7200.0 | 0.0E+00 | 0.0 |
| MGLY | 3.5E+03 | 7200.0 | 0.0E+00 | 0.0 |
| HCOOH | 6.9E+03 | 5600.0 | 1.8E-04 | -1510.0 |
| $MeCO_3H$ | 7.5E+02 | 5300.0 | 6.3E-09 | 0.0 |
| $MeCO_2H$ | 4.7E+03 | 6000.0 | 1.8E-05 | 0.0 |
| MeOH | 2.3E+02 | 4900.0 | 0.0E+00 | 0.0 |

**Table 8.** Values required to calculate the effective Henry's Law coefficient for the soluble stratospheric species included in the UKCA strat-trop scheme, where $Me=CH_3$, $Et=C_2H_5$, $Pr=C_3H_7$.

| Species | Henry's Law Data | | Dissociation Data | |
|---|---|---|---|---|
| | $K_H$(298 K) | $-\Delta H/R$ | $K_a$(298 K) | $-\Delta H/R$ |
| | M atm$^{-1}$ | K$^{-1}$ | M | K$^{-1}$ |
| $BrONO_2$ | 2.1E+05 | 8700.0 | 1.57E+02 | 0.0 |
| HCl | 1.9E+01 | 600.0 | 1.0E+04 | 0.0 |
| HOCl | 9.2E+02 | 5900.0 | 3.2E+06 | 0.0 |
| HBr | 1.3E+00 | 10,200.0 | 1.0E+09 | 0.0 |
| HOBr | 6.1E+04 | 0.0 | 0.0E+00 | 0.0 |
| $ClONO_2$ | 2.1E+05 | 8700.0 | 1.57E+01 | 0.0 |

**Table 9.** Values required to calculate the effective Henry's Law coefficient for the soluble aerosol precursor species included in the UKCA strat-trop scheme, where $Me=CH_3$, $Et=C_2H_5$, $Pr=C_3H_7$.

| Species | Henry's Law Data | | Dissociation Data | |
|---|---|---|---|---|
| | $K_H$(298 K) | $-\Delta H/R$ | $K_a$(298 K) | $-\Delta H/R$ |
| | M atm$^{-1}$ | K$^{-1}$ | M | K$^{-1}$ |
| $O_3$ | 1.13E-02 | 2300.0 | 0.0E+00 | 0.0 |
| $SO_2$ | 1.23E+00 | 3020.0 | 1.23E-02 | 2010.00 |
| DMSO | 5.0E+04 | 6425.0 | 0.0E+00 | 0.0 |

**2.6 Emissions**



In this section, the implementation of tropospheric ozone precursor emissions used in the
UKCA StratTrop scheme are described in detail. The StratTrop scheme includes the
emissions of nine chemical species: nitrogen oxide (NO), carbon monoxide (CO),
formaldehyde (HCHO), ethane ($C_2H_6$), propane ($C_3H_8$), acetaldehyde (MeCHO), acetone
($Me_2CO$), isoprene ($C_5H_8$), and methanol (MeOH). Emissions to UKCA can be broadly
classified into two categories: *offline,* where pre-computed fluxes are read from input files; and
*online,* where fluxes are computed in real-time during the simulation making use of online
meteorological variables from the MetUM. The implementation of *offline* emissions will be
described in Section 2.6.1. Examples of *online* emissions currently in UKCA StratTrop are
biogenic volatile organic compound (BVOC) emissions (Section 2.6.2) and lightning NOx
(Section 2.6.3).
When UKCA StratTrop is coupled to the UKCA aerosol scheme, GLOMAP-mode (Mann et al.,
2010) as here, there are additional trace gas aerosol-precursor emissions for dimethyl
sulphide (DMS), sulphur dioxide ($SO_2$), and monoterpenes ($C_{10}H_{16}$). These emissions will be
discussed in the context of the UKESM1 aerosol performance in Mulcahy et al. (2019); the
focus here will solely be on the tropospheric ozone precursor emissions. Table 11 and Figures
5-8 summarise the mean global annual emissions totals for the time period considered here
(2005-2014) and their global and seasonal distributions.
**2.6.1 *Offline* Anthropogenic and Natural Emissions**
*Offline* tropospheric ozone precursor emissions are either injected into the model's lowest
layer or, in the case of aircraft emissions and some biomass burning emissions, injected into
a number of model levels. The emissions are added to the appropriate UKCA tracers (see
Table 1) and mixed simultaneously by the boundary-layer mixing scheme (Section 2.1). Boreal
and temperate forest and deforestation emissions are considered 'high-level' and are spread
uniformly up to level 20 (~3 km in L85).
For anthropogenic emissions, we make use of historical (1750–2014) annual emissions of
reactive gases from the Community Emissions Data System (CEDS; Hoesly et al., 2018) that
were prepared for use in CMIP6. The CEDS emissions are generally greater  that those of
other emission datasets (e.g. Lamarque et al., 2010) for the years that are used in the
simulations evaluated here (i.e. 2005-2014). Biomass burning emissions are taken from van
Marle et al. (2017). They combined satellite observations from 1997 with various proxies and
output from six fire models participating in the Fire Model Intercomparison Project (FireMIP;
Rabin et al., 2017) to provide a complete dataset of biomass burning emissions from 1750 to
2014 for use in CMIP6. As was the case for anthropogenic emissions, emissions from the
years 2005-2014 are used here. For both anthropogenic and biomass burning, the emissions
were re-gridded from their native resolution to N96L85 while conserving global annual totals
and seasonal cycles. For VOCs,  emissions of all $C_2$ and $C_3$ VOCs are included as ethane and
propane, respectively.
For natural emissions which are not simulated, *offline* emissions are prescribed through the
provision of pre-computed fluxes. For example, oceanic emissions of CO, ethane (including
ethene ($C_2H_4$)), propane (including propene ($C_3H_6$)) are taken from the POET (Granier et al.,
2005) inventory for the year 1990 which contains one annual cycle with 12 monthly fluxes.
These fluxes are applied perpetually to all years of the time series. Biogenic emissions of



acetaldehyde (MeCHO) make use of combined emissions of MeCHO and other aldehydes
from the MACCity-MEGAN emissions inventory (Sindelarova et al., 2014); biogenic emissions
of CO, HCHO, MeOH, and propane (including $C_3H_6$) are also taken from this inventory. For
biogenic acetone emissions, emissions of acetone and *other* ketones from the MACCity-
MEGAN emissions inventory (Sindelarova et al., 2014) are combined. Based on the years
2001-2010, a monthly mean climatology is derived and applied to all years (see Section 3 for
the implementation of the emission in the model). Finally, soil emissions of $NO_x$ are distributed
according to Yienger and Levy (1995) and scaled to give a global annual total of 12.0 Tg NO/yr
and again perpetually applied to all years.
**2.6.2 Biogenic VOC emissions**
In the standard configuration of UKCA StratTrop in UKESM1, emissions of organic compounds
from the natural environment (BVOC) are added to UKCA interactively (Sellar et al., 2019a).
Specifically, emissions of isoprene ($C_5H_8$) and (mono-)terpenes are *online*, the latter
represented by a lumped compound in UKCA with the formula $C_{10}H_{16}$ and a corresponding
molecular weight of 136 g mol$^{-1}$, are calculated by the interactive biogenic VOC (iBVOC)
emission model (Pacifico et al., 2011). Emission fluxes are passed to UKCA at every model
time step.
In iBVOC the emissions of isoprene are coupled to the gross primary productivity of the
terrestrial vegetation (Arneth et al., 2007; Pacifico et al., 2011). The biogenic emission of all
other organic compounds included in the iBVOC model, i.e., (mono-)terpenes, methanol, and
acetone, follow the original model described in Guenther et al. (1995). Note that the current
configuration of UKCA used in UKESM1 does not make use of the interactive emissions of
methanol or acetone; these are *offline* as discussed in Section 2.6.1. To the best of our
knowledge, in the case of the non-isoprene biogenic VOCs there exists no equivalent process-
based formulation for an interactive BVOC emission model applicable to Earth System Models
(ESMs).
For present-day conditions total global annual emissions of isoprene amount to 495.9 (±13.6)
Tg(C) yr$^{-1}$. This number represents the 10-year average annual total emission strength and
the uncertainty quantified by the standard deviation over the 10-year period between 2005
and 2014 taken from a historic run with UKESM1 (Sellar et al., 2019a). This is in good
agreement with estimates reported for other emission models  (e.g. Arneth et al., 2008;
Guenther et al., 2012; Messina et al., 2016; Müller et al., 2008; Sindelarova et al., 2014;
Stavrakou et al., 2009; Young et al., 2009). However, it has been argued that wide model
agreement is achieved rather due to model tuning than due to a high level of process
understanding (c.f., Arneth et al., 2008). For the global annual total (mono-)terpene emissions,
iBVOC calculates 115.1 (±1.6) Tg(C) yr$^{-1}$ over the same period of model simulation. This model
estimate is in reasonable good agreement with the literature  (e.g., Folberth et al., 2006;
Lathière et al., 2006; Arneth et al., 2007, 2011; Acosta Navarro et al., 2014; Sindelarova et al.,
2014; Bauwens et al., 2016; Messina et al., 2016).
In the configuration of UKCA StratTrop used in UKESM1, isoprene is included in the gas-
phase chemistry but does not contribute to the formation of secondary organic aerosol (SOA).
Emissions of (mono-)terpenes are oxidised using a fixed yield approach (e.g. Kelly et al., 2018)
to form SOA in the GLOMAP-mode aerosol scheme - see Table 2 and Mulcahy et al., 2019
for a detailed description and evaluation.



### 2.6.3 Emissions of $NO_x$ from lightning

The lightning $NO_x$ emissions scheme in UKCA StratTrop is based on the cloud top parameterisation proposed by Price and Rind (1992). Based on satellite data and storm measurements, the lightning flash density is parameterised as:

$$F_l = 3.44 \times 10^{-5} H^{4.9} \qquad \text{Eq 1.}$$
$$F_o = 6.2 \times 10^{-4} H^{1.3} \qquad \text{Eq 2.}$$

where F is the flash density (flash $min^{-1}$), H is the cloud top height (km), and the l and o subscripts are used to represent the land and ocean, respectively, and distinguish between the updraft velocities experienced over the two surfaces. The scheme also differentiates between cloud-to-cloud and cloud-to-ground flashes based on the grid cell latitude (Price and Rind, 1993) and is resolution-independent by the implementation of a spatial calibration factor (Prince and Rind, 1994). A minimum cloud depth of 5 km is required for $NO_x$ emissions to be activated and is diagnosed on a timestep basis from the physical model's convection scheme. For $NO_x$ production, the parameterisation assumes that the production efficiency per unit of energy discharged is $25 \times 10^{16}$ molec (NO) $J^{-1}$, with the energy discharged from cloud-to-ground flashes being 3 times greater than that for cloud-to-cloud flashes.

This implementation is identical to that implemented in HadGEM2-ES (Collins et al., 2011) by O'Connor et al. (2014) except that $NO_x$ emissions are now distributed linearly in altitude in log(pressure) rather than linearly in pressure. Whereas global annual lightning emissions in HadGEM2-ES were inadvertently too low (O'Connor et al., 2014; Young et al., 2013), here, the emissions have been scaled to give an average global annual emission rate of 5.93 and 5.98 TgN $yr^{-1}$ over the period 2005 to 2014 in the free-running and nudged simulations, respectively. When compared with anthropogenic, biomass burning and natural emissions, lightning contributes approximately 10 % to the global annual $NO_x$ emission rate, consistent with estimates from Schumann and Huntrieser (2007).

Figure 1 shows tropical distributions of decadal mean annual flash density as observed by the Lightning Imaging Sensor (LIS) on board the Tropical Rainfall Measuring Mission (TRMM) satellite (Mack et al., 2007) in comparison with the free-running simulation being evaluated here (see section 3 for details). It demonstrates that UKCA is capable of capturing the broad features of the observed climatology, with peak densities over S. America, Africa, and East Asia; the spatial coefficient of determination ($R^2$) between the modelled and observed climatology is 0.65 and 0.69 in the free-running and nudged (not shown) simulations, respectively. However, the model tends to be biased low in regions of low flash density (e.g. over the oceans and towards the extra tropics) than compared to the observations (Figure 2), consistent with the assessment of Finney et al. (2014). In considering the variability, the spatial $R^2$ between the modelled and observed variability is 0.57 and 0.59 in the free-running and nudged simulations, respectively. The variability from UKCA is comparable in magnitude to that observed over Africa, albeit displaced geographically. Over the Maritime continent and S. America, for example, UKCA overestimates the variability relative to the LIS observations.





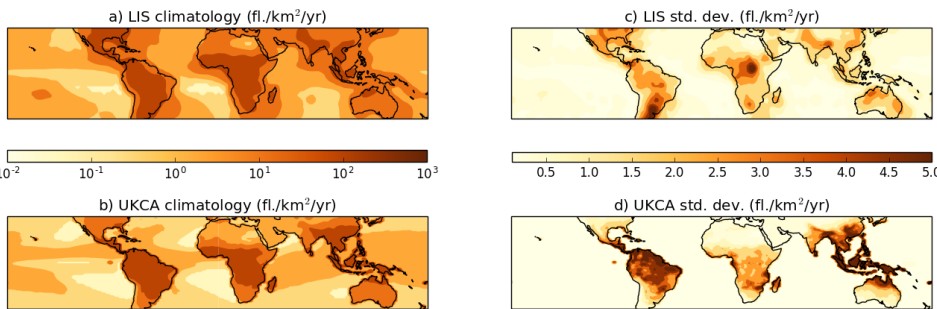

**Figure 1.** Tropical distribution of the LIS-observed climatological annual mean lightning flash density over the period 1999-2013 in a) in comparison with the modelled annual mean climatology from the period 2005-2014 in b). The corresponding standard deviation of the observed and modelled climatologies are shown in c) and d), respectively.

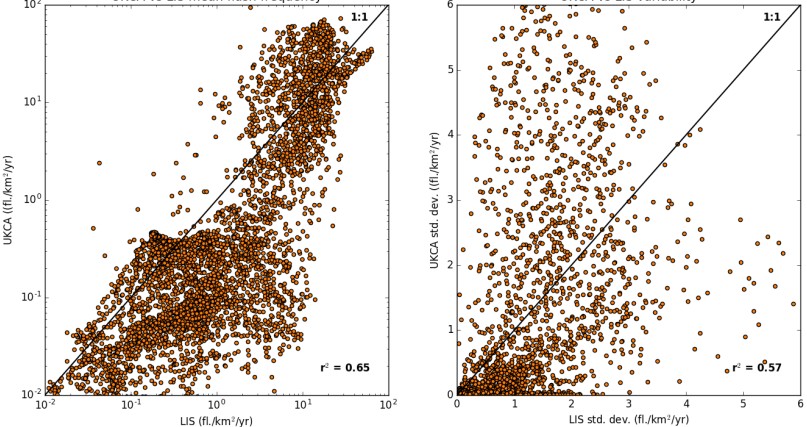

**Figure 2.** Scatter plot of the modelled versus the LIS-observed multi-annual annual mean lightning flash density (left) and the standard deviation (right).

Whilst the skill of the cloud top parameterisation is good relative to other parameterisations, (Finney et al., 2014) and the performance here in the free-running and nudged model simulations is consistent with that assessment, raising the diagnosed cloud top height over land to the power of 4.9 makes the cloud top parameterisation susceptible to model biases in cloud top height, as noted by Allen and Pickering (2002) and Tost et al. (2007). Lightning is potentially a key chemistry-climate interaction in Earth System Models but the sensitivity to how it is represented (i.e. using cloud top height (Banerjee et al., 2014) or ice-flux based parameterisations (Finney et al., 2018)) warrants further investigation. Indeed, Hakim et al. (2019) recently identified uncertainty in modelled lightning $NO_x$ in the Indian subcontinent as being an important source of uncertainty in model simulations of tropospheric ozone in that region.



### 2.6.4 Lower boundary conditions

Lower boundary conditions are provided at the surface for the chemical species $CH_4$, $N_2O$, CFC-11 ($CFCl_3$), CFC-12 ($CF_2Cl_2$), $CH_3Br$, $H_2$, and COS. Values for $H_2$ and COS are fixed at 500 ppb and 482.8 ppt respectively (invariant with time). Values for the remaining species are specified using time series data provided for the 5th Coupled Model Intercomparison Project (CMIP5) for the greenhouse gas concentrations (*see RCP webpage in references*). The values provided are valid on the 1st July for each year specified, and are linearly interpolated in time to give daily values if data for more than one time-point is defined. CFC-11, CFC-12, and $CH_3Br$ also contain contributions from other Cl and Br containing species to ensure that there is the correct stratospheric chlorine and bromine loading, with these contributing species given in Table 10. These values are converted into a two-dimensional "effective emission" field at each timestep that is used to fix the surface concentrations of these species.

**Table 10.** List of species contributing to the lower boundary conditions of CFC-11, CFC-12, and $CH_3Br$. Note that H-1211 contributes to both CFC-11 and $CH_3Br$ as it contains both Cl and Br. Contributions are included by moles of Cl or Br.

| CFC-11 | CFC-12 | $CH_3Br$ |
|---|---|---|
| $CCl_4$ | CFC-113 | H-1211 |
| $CH_3CCl_3$ | CFC-114 | H-1202 |
| HCFC-141b | CFC-115 | H-1301 |
| HCFC-142b | HCFC-22 | H-2402 |
| H-1211 | | |
| $CH_3Cl$ | | |

### 2.7 Coupling with other Earth System components

Secondary aerosol formation of sulphate and organic carbon in UKESM1 (Sellar et al., 2019a) is determined by oxidants (OH, $O_3$, $H_2O_2$, $NO_3$) modelled interactively by the UKCA StratTrop chemistry scheme. For further details on the oxidation of sulphate and SOA precursors, chemistry-aerosol coupling, and the scientific performance of the aerosol scheme (GLOMAP-mode; Mann et al., 2010) in UKCA and UKESM1, the reader is referred to Mulcahy et al. (2019).

In the HadGEM2-ES model (Collins et al., 2011) used for CMIP5, radiative feedbacks between UKCA modelled methane and tropospheric ozone concentrations were active (OC14); stratospheric ozone was prescribed and combined with the modelled interactive tropospheric concentrations. In UKESM1 (Sellar et al., 2019a), however, the coupling between the UKCA modelled radiatively active trace gases and the radiation scheme has been extended to include $N_2O$ and stratospheric ozone (in addition to methane and tropospheric ozone). Although chlorofluorocarbons (CFCs) and hydrochlorofluorocarbons (HCFCs) are modelled in UKCA StratTrop, the radiation scheme cannot handle the speciation. Therefore, separate lumped species (CFC12-eq and HFC134a-eq) are prescribed in the radiation scheme (see Section 2.6.4 on how the lumping/mapping is done).



**2.7.1 Heterogeneous chemistry couplings:**

In UKCA StratTrop as implemented in UKESM1, 5 different heterogeneous reactions are included (see Table 5). These reactions occur on the modelled soluble aerosol surface area, which in the troposphere is calculated interactively using GLOMAP-mode by summing over all soluble aerosol modes. In the stratosphere (defined here as being 12 km above the surface) the aerosol surface area comes from the stratospheric sulfate surface area density input climatology, discussed in Sellar et al. (2019b). The combining of the stratospheric aerosol surface area density from the climatology and the interactive components of GLOMAP-mode is calculated at each UKCA time step and only the soluble aerosol modes simulated by GLOMAP are included in the calculation.

Heterogeneous reactions are extremely important for simulating composition change in the stratosphere (Keeble et al., 2014) and there is increasing attention to the simulation of these processes in the troposphere too (e.g. Jacob et al., 2000; Lowe et al., 2015). One of the most important tropospheric heterogeneous reactions is that of $N_2O_5$ on aerosol surfaces (Jacob et al., 2000). This reaction is complicated because of the dependence of the uptake parameter ($\gamma$) on the composition of the aerosol as well as on temperature and relative humidity (Bertram and Thornton 2009). Macintyre and Evans (2010) suggest that models that use high values of $\gamma N_2O_5$ (~0.1) overestimate the impact of changing aerosol loadings on tropospheric composition through heterogeneous uptake. In UKCA StratTrop, $\gamma N_2O_5$ is set at this higher value, 0.1, throughout the atmosphere. In part this compensates for the fact that there is an important missing aerosol surface in UKESM1 in the troposphere, in the form of nitrate aerosol. The lack of nitrate aerosol is an issue for UKESM1 simulations of particulate matter, particularly in regions with high levels of ammonia emissions. A better understanding of $\gamma N_2O_5$ is needed to both understand current composition but also the combined impact of changing gas and aerosol-phase composition. Whilst more sophisticated treatments of $\gamma N_2O_5$ are available (e.g. Bertram and Thornton 2009) and have been included in versions of UKCA, further work is required to improve this aspect of the mechanism for UKCA in UKESM1.

**2.7.2 Chemical production of $H_2O$.**

There are many chemical reactions which consume or produce water vapour in the troposphere and stratosphere. For example, reactions between the hydroxyl radical (OH) and VOCs usually result in the production of a water molecule:

$$OH + VOC \rightarrow H_2O + \text{Organic Radical} \qquad \text{Eq 3.}$$

In the troposphere the chemical source of water vapour is negligible compared with that from the oceans and plant evapotranspiration, but given the low temperatures around the tropopause, chemically produced water is very important in the lower stratosphere. Furthermore, the main source of chemical water in the middle to upper stratosphere comes from the oxidation of $CH_4$. Complete oxidation of $CH_4$ to $CO_2$ can result in the net production of two water molecules.

In previous versions of UKCA, such as that used in HadGEM2-ES, the oxidation of $CH_4$ to produce chemical water was neglected. Instead stratospheric water vapour was simulated using the following simple relationship:

$$2*[CH_4] + [H_2O] = 3.75 \ (ppm) \qquad \text{Eq 4.}$$



where UKCA was used to calculate [$CH_4$]. In UKCA StratTrop as implemented in UKESM1 we
now include interactive $H_2O$ production from all chemical reactions in the mechanism. In this
way UKCA now passes the water vapour field after the chemistry step back to the main climate
model where it is used in other routines. The annual mean zonal mean chemical production
of $H_2O$ as simulated by UKESM1 is shown in Figure 3. There are two clear regions which
dominate where $H_2O$ chemical production takes place, in the tropical lower troposphere and
the tropical upper stratosphere. In both regions the primary source of chemical water is the
oxidation of $CH_4$. Figure 3 compares the absolute production of chemical water (panel a) and
the production of chemical water as expressed in mixing ratio units (panel b). In this sense,
panel (b) shows the relative production of chemical water is greatest in the upper stratosphere.
The contribution of this source of stratospheric $H_2O$ to the present day forcing of climate
relative to the pre-industrial period will be assessed in O'Connor et al. (2019).

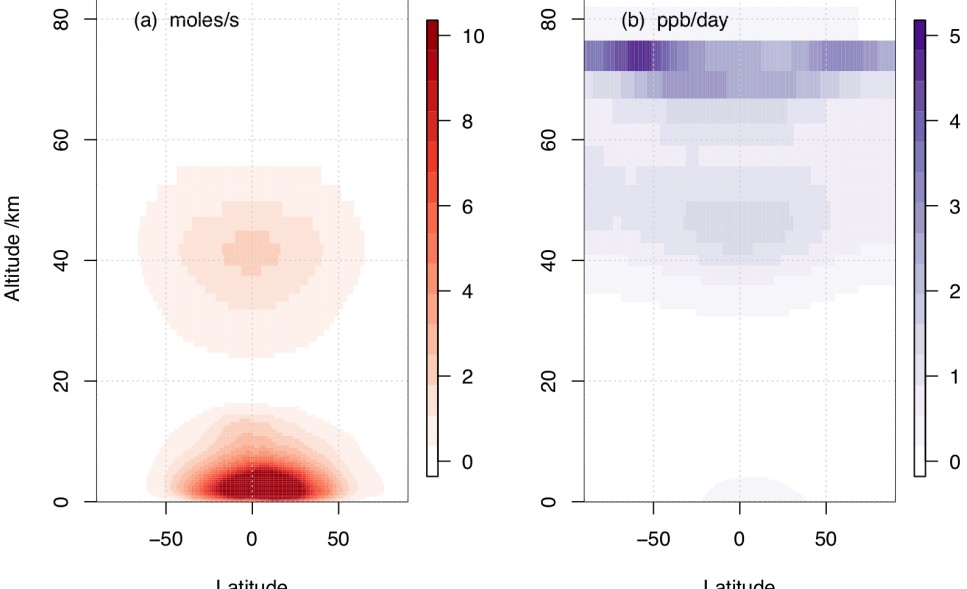

**Figure 3.** Multi annual mean zonal mean production of $H_2O$ from the UKCA StratTrop mechanism in
UKESM1. Panel (a) shows the production in moles/s and panel (b) in ppb/day, highlighting the larger
relative source of water from chemical processes in the upper atmosphere.
**2.7.3 Future couplings.**
Although UKESM1 (Sellar et al., 2019a) represents a significant enhancement in the
representation of atmospheric chemistry and ES interactions, a number of key interactions are
not included. For example, the coupling of aerosols with Fast-JX is omitted despite the impact
of aerosols on tropospheric photochemical production of ozone (e.g. Xing et al., 2017; Wang
et al., 2019). This development is currently underway and will be included in future versions
of UKCA and UKESM. Ozone damage to natural and managed ecosystems (e.g. Ashmore,
2005) has an important impact on the strength of carbon uptake by vegetation (Sitch et al.,
2007; Oliver et al., 2018) and has yet to be implemented. In addition, although the terrestrial
carbon cycle considers nitrogen availability/limitation, nitrogen deposition rates are prescribed
in UKESM1; future work will include implementing a nitrate aerosol scheme in GLOMAP-mode



and coupling deposition of both oxidised and reduced nitrogen from the atmosphere to the
terrestrial biosphere.
**2.8 Historic development of the chemistry scheme**
During the development of the StratTrop chemistry scheme, several simulations were run to
test the scheme and its sensitivity to different (a) rate coefficients (updating the JPL and IUPAC
recommendations), (b) reactions (by looking at the sensitivity to specific reactions associated
with isoprene oxidation (Archibald et al., 2011) and the reaction between $HO_2$ and NO
(Butkovzkaya et al., 2005; 2007; 2009), (c) treatment of photolysis, (d) emissions and (e)
deposition parameters. These one-at-a-time simulations are outlined in Table S1 in the
Supplement. It should be noted that these simulations provide an ensemble of opportunity;
they were not designed to probe model sensitivity in a targeted way. However, they result in
some useful information which helped the development of the StratTrop mechanism. These
simulations made use of an older version of the MetUM and earlier atmosphere only version
of UKCA, which is now deprecated. That version of UKCA ran at a lower resolution than the
version discussed in this paper and used in UKESM1 (about half the resolution). Results from
these simulations are shown in Figure 4 were they are compared against results from model
intercomparison studies (further analysis of the model sensitivity tests is presented in the
Supplement Figures S1 to S6). Figure 4 focuses on a subset of the full range of experiments
performed but contextualises these by comparing to results from the ACCENT simulations
discussed in Stevenson et al. (2006) (black dots) and the ACCMIP simulations discussed in
Young et al. (2013) (orange dots). In addition to the early sensitivity tests (the blue dots in
Figure 4), we also show the results from the simulations presented here, labelled UKESM1
(red triangle in Figure 4). The figure focuses on the relationship between methane lifetime and
ozone chemical loss, important metrics for representing key sources and sinks of tropospheric
OH (Wild 2007). Both metrics are calculated by masking out the stratosphere. The methane
lifetime is calculated by dividing the burden of methane in the model by the reaction flux
between methane and OH in the troposphere and so represents the lifetime with respect to
OH in the troposphere. The ozone loss is calculated by summing the reaction fluxes which are
key for $O_3$ loss in the troposphere (reactions of $O_3$ with $HO_x$ species and the reaction between
$O(^1D)$ and $H_2O$). The experiments outlined in Table S1 and shown in Figure 4 emphasise that
the range in $O_3$ loss and $CH_4$ lifetime spanned by changing aspects of the UKCA model span
a range as wide as that covered by the ACCMIP models (Young et al., 2013). In other words,
the *ensemble of opportunity* from the early tests of the UKCA StratTrop scheme span as wide
a range in the metrics presented as the structurally different ACCMIP and ACCENT models.
Interestingly, the UKESM1 simulations discussed in this paper in detail lie close to the
ACCENT ensemble (black dots), yet the early test simulations using the same chemical
mechanism but an earlier version of the MetUM model do not (the blue cluster of dots). This
highlights that structural changes in the underlying meteorological model can substantially
influence key metrics of atmospheric composition through changes in the distribution of
clouds, water vapour and other key variables.





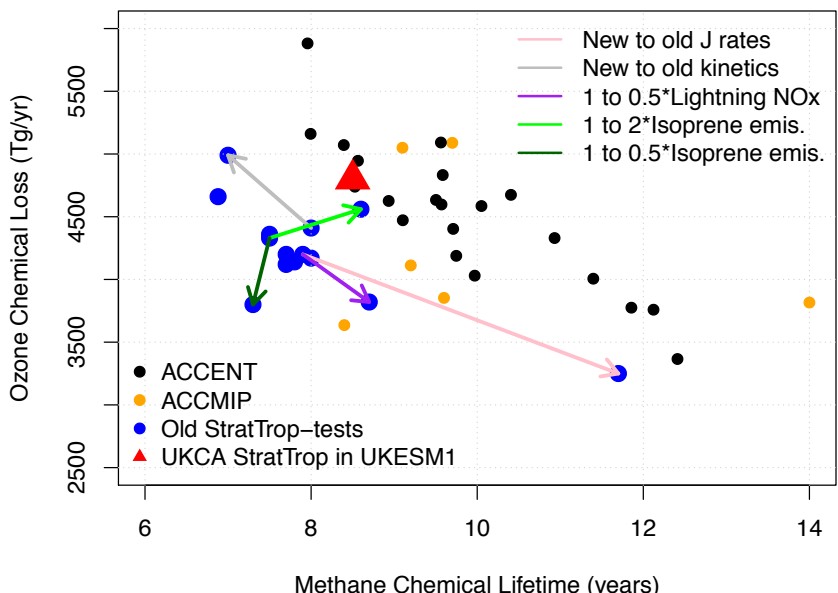

**Figure 4.** Comparison of early tests of the StratTrop scheme running in an older version of UKCA (blue dots) with the scheme applied in UKESM1 (red triangle) and other CCMs which took part in the ACCMIP intercomparison (orange dots) and CTMs which took part in the ACCENT intercomparison (black dots).

These sensitivity studies highlight some important points. Simulations using kinetic data recommendations from IUPAC and JPL updated from 2005 to 2011 led to a decrease in model methane lifetime and an increase in ozone chemical loss flux (grey arrow), indicating increased photochemically activity. The attribution of which rate coefficients were dominant in this behaviour is outside of the scope of this work. Similarly, we note that the metrics analysed are sensitive to lightning $NO_x$ (Banerjee et al., 2014); decreasing the lightning $NO_x$ emissions by 50 % (to ~ 3 Tg/yr) results in an increased methane lifetime of ~ 1 year (purple arrow). Figure 4 also highlights a non-linear response in the simulations to changes in isoprene emissions; scaling them by a factor of two (100 % increase and 50 % decrease, green arrows) leads to a highly non-linear response in the metrics analysed. Finally we note that the change which had the biggest impact on the metrics was switching to the FAST-JX photolysis scheme (Telford et al., 2013) from pre-calculated photolysis rates and a look-up table (pink arrow). The main reason for this is that the pre-calculated photolysis rates had underestimated rates for the photolysis of $O_3$ to $O(^1D)$. This behaviour has been documented previously (Voulgarakis et al., 2009; Telford et al., 2013).

In addition to the tests described above we found during the testing of the StratTrop scheme that inclusion of the termolecular reaction:

$$HO_2 + NO + M \rightarrow HONO_2 + M, \qquad\qquad\qquad\qquad Eq 5.$$

which has been shown to exhibit both pressure and water vapour dependence (Butkovzkaya et al., 2005; 2007; 2009), led to large changes in the metrics analysed in Figure 4 (see the Supplement for further details). Previous modelling work highlighted that this could have an





important impact on the simulation of ozone (Cariolle et al., 2008). However, owing to
uncertainty in its recommendation between the recent evaluations by JPL and IUPAC we have
omitted it from the StratTrop scheme used in UKESM1.
**3.0 Model simulations to evaluate UKCA StratTrop in UKESM1.**
In this section, we discuss a series of simulations that have been performed to evaluate the
performance of the UKCA StratTrop scheme in UKESM1. These simulations link closely to
the UKESM1 Historical and AMIP simulations by using similar inputs, e.g. emissions, and
crucially the version of UKCA StratTrop is identical to that used in UKESM1 (Sellar et al.,
2019a).
Simulations analysed in this paper have been carried out with an atmosphere-only
configuration of UKESM1 (Sellar et al., 2019a). The sea surface temperatures and sea ice
cover used to drive the model are those specified for the historical period by the 6th Coupled
Model Intercomparison Project (CMIP6 project; Durack et al., 2016). Land cover fraction,
vegetation canopy height and leaf area index (LAI) have been provided as multi-annual
monthly mean climatologies, derived from a historical simulation of UKESM1 which includes
the dynamic vegetation model TRIFFID (Cox, 2001). Anthropogenic and biomass burning
emissions of ozone precursors are prescribed on a monthly basis using a 2005-2014
timeseries from Hoesly et al. (2018) (see section 2.6) and van Marle et al. (2017), respectively.
Land-based biogenic emissions not simulated within the JULES model (e.g. CO) are provided
as monthly climatologies for the period 2001-2010 from the MEGAN-MACC dataset
(Sindelarova et al 2014), supplemented by soil $NO_x$ emissions based on Yienger and Levy
(1995) and oceanic emissions from POET. Greenhouse gas concentrations for CFC12, $CH_4$,
$CO_2$, HFC134 and $N_2O$ are derived from the dataset generated by Meinshausen et al (2017)
for CMIP6. Concentrations of other CFCs seen only by UKCA are derived from the same
dataset but described in more detail under Lower Boundary Conditions (Section 2.6.4). The
model is initialised using output after nearly 150 years of the UKESM1 coupled historical
simulation. The land surface setup used in this paper is based on a 27 sub-grid tile
configuration including 13 plant functional types (three broadleaf tree tiles, two needleleaf tree
tiles, three C3-grass tiles including crops, three C4-grass tiles including crops, and two tiles
representing shrubs), one water tile (to represent lakes), one tile for bare soil, one urban tile
and 11 land ice tiles.
Two simulations have been carried out using the Atmosphere-only configuration, covering
January 1999 to December 2014. The first is a *free-running* (FR) simulation where the
meteorology is allowed to evolve independently based on the influence of the aforementioned
forcing agents. The second is a *Nudged* (ND) simulation where the meteorology, though under
the same forcings as the FR simulation, is in addition relaxed toward the ECMWF's ERA-
Interim reanalysis (Dee et al., 2011) using the Nudging functionality in the MetUM (Telford et
al., 2008). Nudging is applied to model temperature and winds from about 1.2 km (to be
generally free of the boundary layer) to 65 km (maximum height of ERA data), using an e-
folding relaxation timescale of 6 hours.
For both simulations, output from the first 6 years is considered as spin-up and analysis from
the years 2005-2014 inclusive is presented in this paper. Model fields used in the analysis
have been output mainly as monthly means. In addition, some aerosol-related fields were





produced at daily and 6-hourly intervals, while ozone, nitric acid and nitrogen dioxide at the
surface were produced at hourly intervals.
Table 11 provides a summary of the sectors contributing to the emissions of the nine
tropospheric ozone precursor species treated in UKCA StratTrop and their corresponding
global annual totals, averaged (mean) over the 2005-2014 time period covered by the two
simulations. Figures 5 and 6 show the multi-annual global annual mean distributions and the
seasonal cycle for different emission sectors and regions for NO and CO, respectively. While
the figures illustrate that the main contribution to NO and CO emissions is of anthropogenic
origin, other sectors are relevant in shaping the yearly cycle. Examples include emissions of
NO from biomass burning in the tropics, soil NO emissions in the extratropics, land biogenic
CO emissions in the extratropics and ocean biogenic CO emissions in the southern
extratropics.
**Table 11.** List of emitted tropospheric ozone precursor species in UKCA StratTrop, including the
contributing sectors and the corresponding global annual totals, averaged over the time period of the
simulations i.e. 2005-2014 inclusive.

| Species | Sector | Total |
|---|---|---|
| $NO_x$ | Anthropogenic | 89.4 |
| | Biomass burning | 14.3 |
| | Soil | 11.8 |
| | Aircraft | 1.9 |
| | Lightning | 12.7 |
| | **Total (Tg(NO)/year)** | **130.1** |
| CO | Anthropogenic | 603.3 |
| | Biomass burning | 347.0 |
| | Land biogenic | 88.6 |
| | Ocean biogenic | 19.6 |
| | **Total (Tg(CO)/year)** | **1089.5** |
| HCHO | Anthropogenic | 2.4 |
| | Biomass burning | 4.8 |
| | Land biogenic | 4.6 |
| | **Total (Tg(HCHO)/year)** | **11.8** |
| $C_2H_6$ (including $C_2H_4$) | Anthropogenic | 16.3 |
| | Biomass burning | 9.3 |
| | Land biogenic | 31.1 |
| | Ocean biogenic | 2.4 |
| | **Total (Tg($C_2H_6$)/year)** | **59.1** |
| $C_3H_8$ (including $C_3H_6$) | Anthropogenic | 10.2 |
| | Biomass burning | 4.5 |
| | Land biogenic | 15.6 |
| | Ocean biogenic | 2.8 |
| | **Total (Tg($C_3H_8$)/year)** | **33.1** |
| MeCHO (including other aldehydes | Anthropogenic | 1.9 |
| | Biomass burning | 7.1 |



| but *not* HCHO) | Land biogenic | 21.5 |
| | **Total (Tg(MeCHO)/year)** | **30.5** |
| $Me_2CO$ | Anthropogenic | 2.8 |
| | Biomass burning | 3.0 |
| | Land biogenic | 37.4 |
| | **Total (Tg(Me$_2$CO)/year)** | **43.2** |
| MeOH | Anthropogenic | 3.8 |
| | Biomass burning | 8.1 |
| | Land biogenic | 129.1 |
| | **Total (Tg(MeOH)/year)** | **141.0** |
| $C_5H_8$ | Land biogenic | 495.9 |
| | **Total (Tg(C)/year)** | **495.9** |
| $C_{10}H_{16}$ | Land biogenic | 115.1 |
| | **Total (Tg(C)/year)** | **115.1** |

2



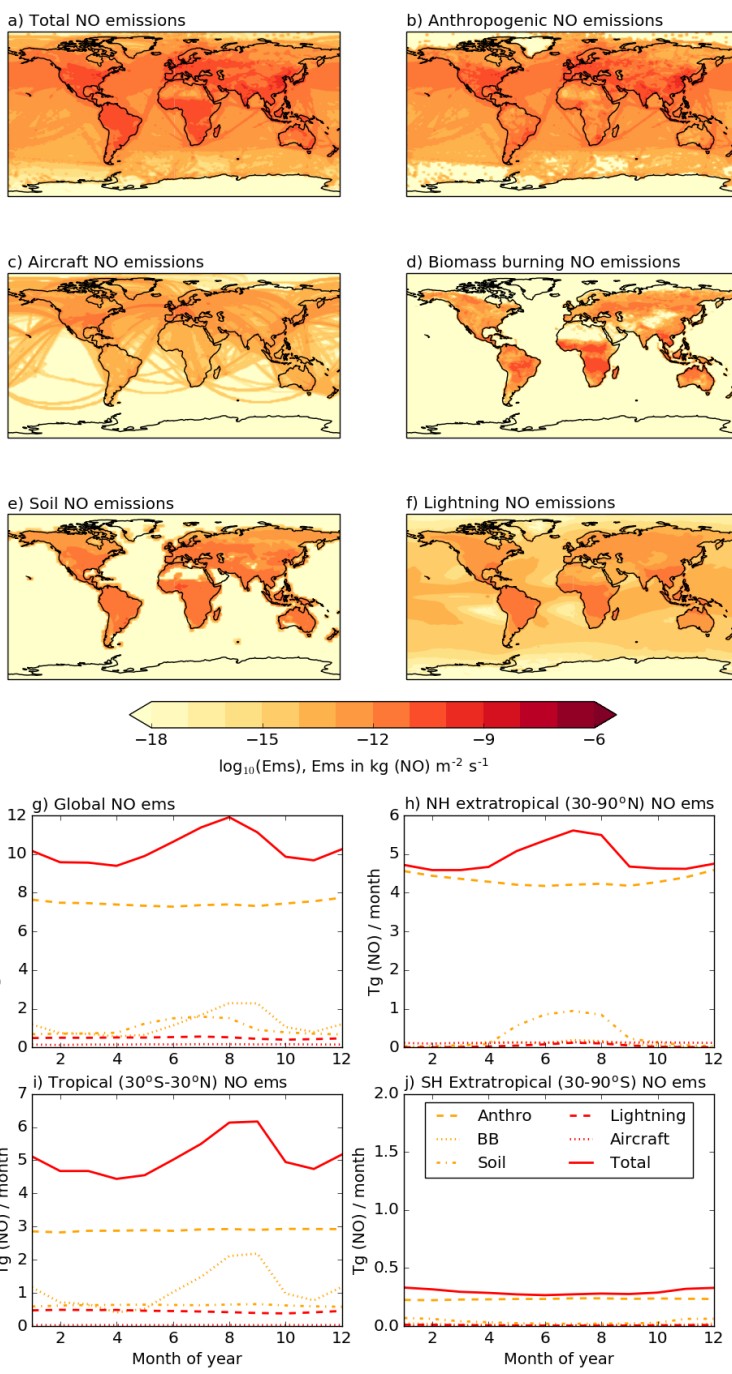

**Figure 5.** Multi-annual mean NO emissions distribution (plotted as log$_{10}$(Ems), with Ems in kg (NO) m$^{-2}$ s$^{-1}$) used in the simulations presented here. Panel (a) highlights the total NO emissions, while panels (b) to (f) show the contributions from anthropogenic, aircraft, biomass burning, soil, and lightning sources, respectively. Aircraft and lightning emissions have been integrated in the vertical. Panels (g)



to (j) show the multi-annual seasonal cycle in Tg(NO)/month over the whole globe, the northern
hemisphere (NH) extratropics (30-90°N), the tropics (30°S-30°N), and the southern hemisphere (SH)
extratropics (30-90°S), respectively.

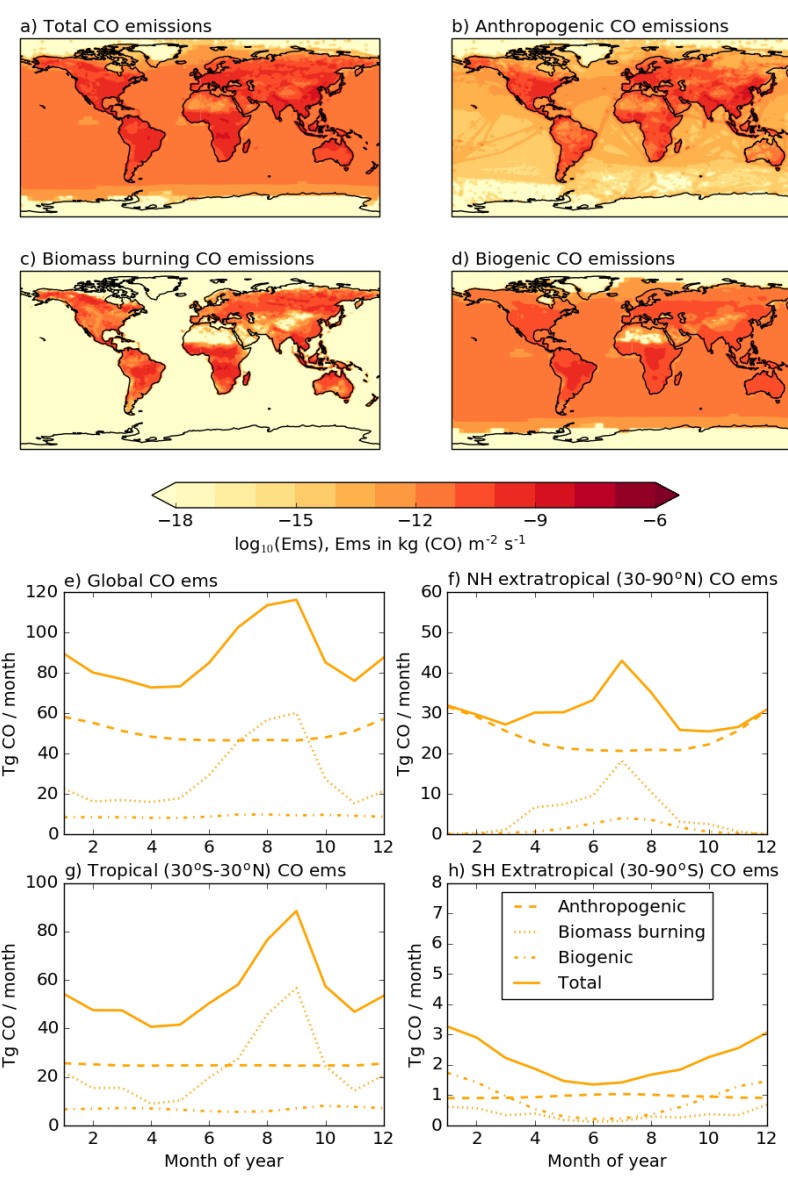

**Figure 6.** Multi-annual mean CO emissions distribution (plotted as $\log_{10}$(Ems), with Ems in kg (CO) m$^{-}$
$^2$ s$^{-1}$) used in the simulations presented here. Panel (a) highlights the total CO emissions, while panels
(b) to (d) show the contributions from anthropogenic, biomass burning and biogenic sources,
respectively. Panels (e) to (h) show the multi-annual seasonal cycle in Tg(CO)/month over the whole



globe, the northern hemisphere (NH) extratropics (30-90°N), the tropics (30°S-30°N), and the
southern hemisphere (SH) extratropics (30-90°S), respectively.
**4.0 Evaluation of model fields**
We start our evaluation of UKCA StratTrop in UKESM1 by assessing the performance of the
model in the troposphere, against surface observations, and build up the evaluation to focus
on tropospheric integrated quantities and stratospheric quantities before concluding with an
analysis of transport in the model.
**4.1 Evaluation of surface ozone against TOAR observations.**
The surface $O_3$ concentrations in the FR simulations with UKCA StratTrop in UKESM1 for
December-January-February (DJF) and June-July-August (JJA) (seasonal means over the
2005-2014 period) show elevated values across the tropics in both seasons as well as in the
northern mid-latitudes in JJA (Figure 7a and c). Maximum surface $O_3$ concentrations of more
than 60 ppb are simulated across the Middle East, Northern Africa and South Asia in JJA due
to large anthropogenic and biogenic sources of $O_3$ precursors. In DJF, surface $O_3$
concentrations are lower over the continental northern mid-latitudes due to slow $O_3$ production
and an enhanced $O_3$ removal from elevated $NO_x$ emissions. Meanwhile, surface $O_3$
concentrations are slightly higher over oceanic areas (North Atlantic and North West Pacific)
in DJF, probably due to transport from the stratosphere and a reduced chemical sink from
weaker photolysis of $O_3$ (Banerjee et al., 2016). Surface $O_3$ concentrations are slightly higher
over some oceanic areas in JJA, indicating long range transport from polluted continental
areas.
Surface $O_3$ concentrations simulated in the nudged configuration of UKESM1 have been
evaluated over the period 2005-2014 by comparing to the gridded rural observations in the
TOAR database (Schultz et al., 2017). These data provide a global perspective on surface $O_3$
and is by far the most comprehensive surface $O_3$ database for use in evaluation of global
models. However, the TOAR database does not provide globally uniform coverage and as
such the evaluation of the model performance for surface $O_3$ over key regions, such as South
Asia (Hakim et al., 2019), will be analysed in more specific follow up studies making use of
bespoke datasets. Figure 7b and d shows that the model underpredicts surface $O_3$
concentrations in DJF and overpredicts $O_3$ in JJA across the northern midlatitudes, in a similar
way to other global models (Young et al., 2018). Potential reasons for these discrepancies
could be the coarse model resolution, associated errors in the emissions inventories, errors in
the vertical injection of the emissions (for example we inject most of the $NO_x$ near the surface
which will titrate $O_3$), representation of VOCs in the chemistry scheme and uncertainties in $O_3$
loss processes (dry deposition).





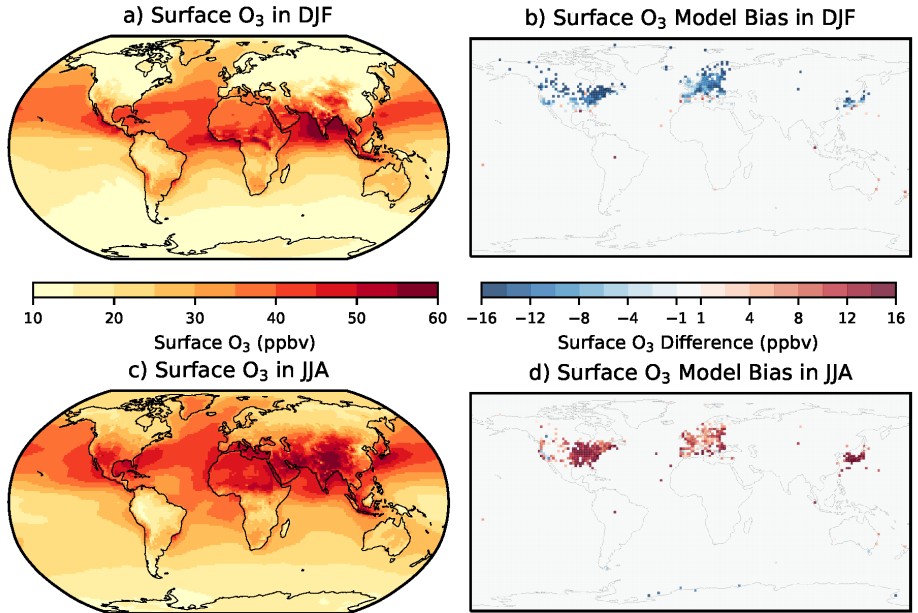

**Figure 7.** Simulated seasonal mean surface $O_3$ concentrations in a) December-January-February (DJF)
and c) June-July-August (JJA) over the 2005-2014 period. Difference between simulated and observed
surface $O_3$ from the gridded TOAR database in b) DJF and d) JJA.
Each grid point containing observations has been evaluated against the corresponding model
values by calculating a normalised mean bias factor (NMBF, Yu et al., 2006). Figure 8 shows
the distribution of NMBFs within a particular region for different seasons. Over northern
midlatitudes (Europe, North America and East Asia) the model clearly underrepresents
surface $O_3$ in DJF (by a factor of 1.5 to 2), suggesting excessive $O_3$ titration by $NO_x$. The model
agrees better with observations in other seasons across these regions, with a slight
overprediction in JJA. The limited available observations in other regions (<10 grid points)
makes it difficult to draw firm conclusions but suggests that UKCA StratTrop in UKESM1 tends
to overpredict surface $O_3$ across the oceanic and southern hemisphere sites. The model
consistently underpredicts observed surface $O_3$ at sites located in Antarctica, implying a lack
of transport and a too low modelled $O_3$ lifetime in this region, particularly in March-April-May
(MAM) and JJA.

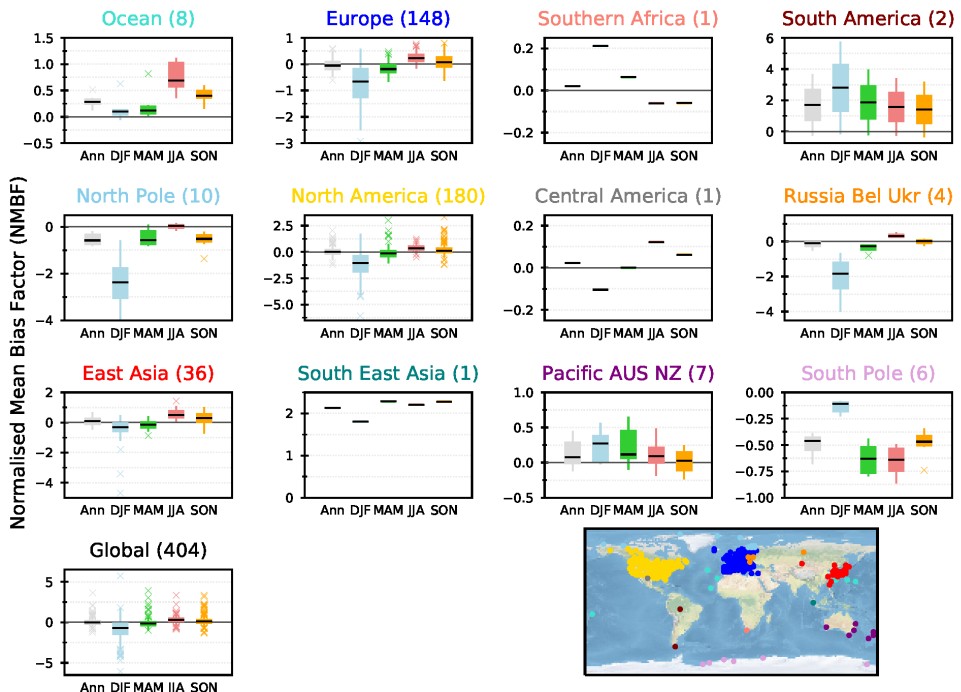

**Figure 8.** Normalised mean bias factors (NMBF) calculated for annual and seasonal means by comparing modelled concentrations of surface $O_3$ in UKESM1 to gridded observations from the TOAR database across each region over the 2005-2014 period. The solid line shows the median value for the region, the boxes show the 25th and 75th percentile values with the error bars showing the maximum and minimum values and the crosses representing outliers (values >1.5 x interquartile range). The total number of sites used for each region is shown in parenthesis. Comparisons on annual (grey), DJF (blue), MAM (green), JJA (red) and SON (orange) timescales are shown.

Simulated daily and monthly mean surface $O_3$ concentrations over the period 2005-2014 from UKESM1 have been interpolated and compared to four individual observation locations from the TOAR database (Figure 9). UKESM1 is able to reproduce the seasonal cycle of surface $O_3$ observed at Cape Grimm ($r^2$ = 0.74 NMBF = -0.08) and South Pole ($r^2$ = 0.79, NMBF = -0.81), although it underestimates the magnitude in JJA at Cape Grimm and in all seasons at the South Pole (Figure 9). There is reasonably good model observational agreement in JJA at the two northern hemisphere sites (Barrow and Mace Head) (albeit with some disagreement in the phase), although in DJF the model underpredicts surface $O_3$ at both sites. The surface model evaluation of UKESM1 at selected individual measurement locations exhibits a similar performance to that of HadGEM2-ES in O'Connor et al. (2014).

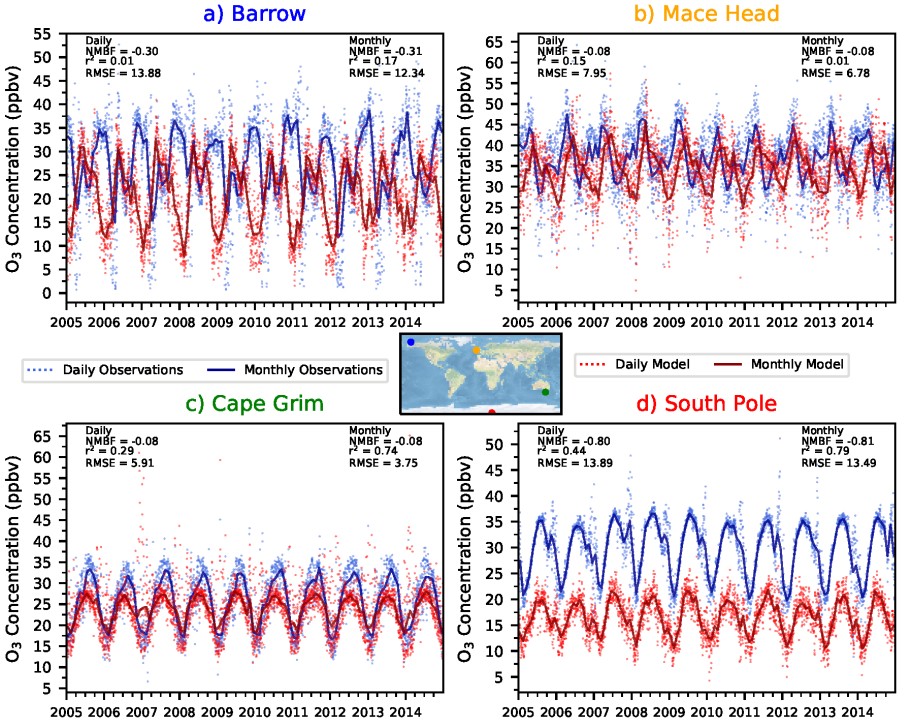

**Figure 9.** Simulated and observed daily and monthly mean surface O$_3$ over the period 2005-2014 at four individual monitoring locations of a) Barrow, b) Mace Head, c) Cape Grimm and d) South Pole

## 4.2 Dry deposition of ozone – comparison with HTAP models and observations

Roughly 1000 Tg(O$_3$), around 20 % to 25 % of the gross chemical ozone production in the troposphere, is removed from the atmosphere at present every year through dry deposition at the surface (Stevenson et al., 2006; Wild, 2007; Young et al., 2013; Hardacre et al., 2015). Uptake by terrestrial vegetation plays a crucial role, however, Hardacre et al. (2015) demonstrated that the oceans represent a very important sink, too. Much uncertainty still remains about the exact magnitude and many of the processes around ozone removal at the surface (e.g., Hardacre et al. 2015; Luhar et al., 2017). A thorough evaluation and, if necessary, re-calibration of ozone dry deposition models is, thus, critical in developing robust models of atmospheric composition.

### 4.2.1 Comparison with the HTAP multi-model ensemble ozone deposition fluxes

Figure 10 shows a comparison of multi-annual average monthly mean ozone deposition modelled by UKCA StratTrop in UKESM1 with a multi-model ensemble of 15 HTAP atmospheric composition models (Hardacre et al., 2015). The StratTrop model data here are taken from the FR simulation. Monthly mean ozone deposition is depicted for the entire global domain (Figure 10 a) and split into the northern extra-tropics, the tropics, and the southern extra-tropics, respectively, each representing a distinctly different deposition regime (Figure 10 b-d). The solid black line and filled circles represent ensemble average monthly mean ozone deposition with the error bars indicating ±1σ in the single model monthly mean ensemble; the solid grey lines represent single model monthly means from the HTAP models

indicating the spread in the multi-model ensemble. The multi-annual average (10 years)
monthly mean ozone dry deposition flux modeled by UKESM1-UKCA is shown as the red solid
line.
In general, ozone dry deposition from UKCA StratTrop in UKESM1 compares favorably with
the HTAP multi-model ensemble falling nearly always within the 1σ-range of the HTAP multi-
model average. UKCA StratTrop also correlates well with the multi-model average seasonal
cycle for each of the depicted regions; however, a systematic low-bias is evident, particularly
in the global and tropical domains (plots a and c in Figure 10). Most of the low bias occurs in
the tropical region. Since the tropics are dominated by both a large ocean surface area and
the most productive portion of the Earth's terrestrial vegetation in the form of the tropical rain
forests of South America, equatorial Africa and the maritime continent, the tropical low bias in
the model could be due to an underestimation of $O_3$ concentration, the stomatal ozone uptake
by tropical rain forests or a similar underestimation of $O_3$ removal at the ocean's air-sea
interface. The latter seems less likely in view of the relatively good performance in the southern
extratropics which are also dominated by a large ocean surface.

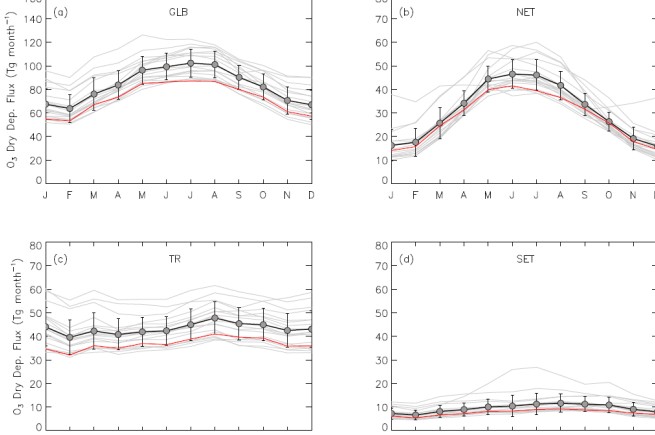

**Figure 10.** Multi-annual average monthly mean $O_3$ dry deposition for the global domain (a) and three
latitudinal sections (b-d): northern extra-tropics (NET; 90N-30N; b), tropics (TR; 30N-30S; c), and
southern extra-tropics (SET; 30S-90S; d) for 15 models participating in the HTAP model
intercomparison. Multi-model ensemble average (solid black line and filled circles) and single model
monthly means (grey solid lines) were provided by Hardacre et al. (2015). Error bars indicate ±1σ in the
single model monthly means. Solid red line shows UKCA StratTrop multi-annual average (2005-2014)
monthly mean $O_3$ dry deposition. (Figure based on Hardacre et al. (2015)).

### 4.2.2 Comparison with observations of ozone deposition fluxes.

Measurements of ozone dry deposition fluxes collected over extended periods of time are still
very sparse, however, a number of long-term datasets exist. Hardacre et al. (2015) compiled
a comprehensive dataset from available long-term and short-term observations. This
comprehensive dataset has been adopted for our evaluation of $O_3$ dry deposition in UKCA
StratTrop in UKESM1. Table 12 summarises the locations of all the measurement sites
included in this comparison. A comparison of the dry deposition fluxes of ozone with



observations at these 16 sites is presented in Figure 11. Some sites cover the seasonal cycle
over several years (e.g., Castel Porziano, Harvard Forest, or Ulborg) and others only offer
data spanning less than one month (e.g., Klippeneck, Le Dezert, or Viols en Levant).
**Table 12.** Ozone surface dry deposition measurement sites (reproduced from Hardacre et al., 2015).

| Site Name | Grid reference | Land cover | Sampling height (m) | Sampling Period | Ref. |
|---|---|---|---|---|---|
| Long-term sites | | | | | |
| Auchencorth Moss | 55°47'N 3°14'E | Moorland | 0.3-3.0 | Oct 1995–Dec 2000 | 1 |
| Blodgett Forest | 38°53'N 120°37'W | Pine plantation | 12.5 | Jan 2001–Dec 2007 | 2 |
| Citrus Orchard | 36°21'N 119°5'W | Citrus Orchard | 1.0–9.2 | Oct 2009–Nov 2010 | 4 |
| Castel Porziano | 41°44'N 12°24'E | Holm Oak | 35 | Jan–Dec 2013 | 5 |
| Harvard Forest | 42°32'N 72°11'W | Mixed deciduous forest | 30 | Jan 1992–Dec 2001 | 7 |
| Hyytiala | 61°51'N 24°17'E | Scots Pine forest | 23 | Jan 2002–Dec 2003 | 8 |
| Ulborg | 56°17'N 8°25'E | Mixed coniferous | 18, 36 | Oct 1995–Dec 2000 | 10 |
| Short-term sites | | | | | |
| Borneo OP3 | 45°8'N 117°51'E | Tropical forest | 75 | Apr, Jul 2008 | 1 |
| Burriana | 39°55'N 0°03'W | Citrus Orchard | 10 | 16–29 Jul 1995 28 Apr–3 May 1996 | 3 |
| La Cape Sud | 44°24'N 0°38'E | Maize Crop | 3.4, 3.7, 6.4 | Jul–Oct 2007 | 6 |
| Klippeneck | 48°10'N 8°45'E | Grass | 2, 8 | 1–22 Sep 1992 | 3 |
| Le Dezert | 44°05'N 0°43'E | Pine forest | 37 | 16–18 Apr 1997 | 3 |
| San Pietro Capofiume | 44°39'N 11°37'E | Beet crop | 8 | 15–22 Jun 1993 | 3 |
| South-western Amazon | 3°00'S 60°00'W | Tropical forest | 53 | May 1999 Sep–Oct 1999 | 9 |
| Viols en Levant | 43°41N 3°47'E | Mediterranean shrub | 37 | 16–24 Jul 1998 | 3 |
| Voghera | 45°01'N 9°00'E | Onion field | 2.5 | May–Jul 2003 | 11 |

References: 1) Fowler et al. (2001); 2) Fares et al. (2010); 3) Cieslik (2004); 4) Fares et al. (2012);
5) Fares et al. (2014); 6) Stella et al. (2011); 7) Munger et al. (1996); 8) Rannik et al. (2012);
9) Fan et al. (1990); 10) Mikkelsen et al. (2004, 2000); 11) Gerosa et al. (2007)
Due to its removal via stomatal exchange and relative insolubility in water, $O_3$ dry deposition
depends strongly on the underlying land surface type. Therefore, a reliable representation of
ozone dry deposition in models requires not only the composition model to perform well. A
robust model of the land surface including dynamic vegetation is also indispensable. The land
surface representation in UKCA StratTrop in UKESM1 relies on JULES (Best et al., 2011;
Clark et al., 2011). Thus, a comparison of ozone dry deposition (or any dry deposition process



for that matter) reflects on the broader Earth system framework than just the atmospheric
composition component alone.

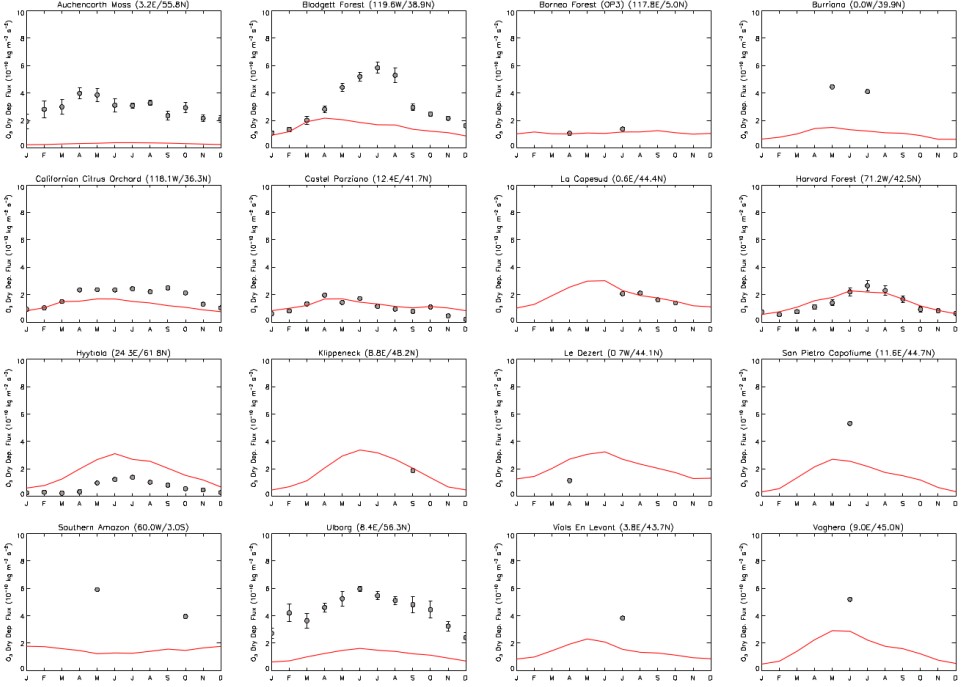

**Figure 11.** Comparison of observed and modelled monthly mean ozone dry deposition fluxes. Grey
circles indicate monthly mean ozone deposition fluxes at measurement sites (c.f. Table 12 for site
details); error bars denote standard errors. Solid red lines represent modelled multi-annual average
monthly mean $O_3$ deposition fluxes extracted from UKCA StratTrop in UKESM1 at the site locations by
interpolation of the nearest grid boxes. Ozone dry deposition fluxes are given in $10^{-10}$ kg m$^{-2}$ s$^{-1}$,
measurement data from Hardacre et al. (2015) and references therein.
Overall, Figure 11 shows that the UKCA(StratTrop)/JULES/UKESM1 framework shows a
reasonably good performance, albeit with some substantial model-to-obs deviations evident
from Figure 11. At the Castel Porziano, La Cape Sud, and Harvard Forest sites the model
reproduces well both magnitude and seasonal cycle of ozone dry deposition. To a somewhat
lesser degree model performance is also good at the California Citrus Orchard and Hyytiala
sites. At both locations the model captures most of the seasonal cycle well but fails to
reproduce the magnitude of the flux fully. Interestingly, there is no systematic bias in the
model-to-obs deviations with respect to magnitude and land cover type.
Further locations with good model-to-obs agreement include the densely forested OP3 site in
Borneo and Klippeneck site in Germany. However, these sites only provide campaign data for
a limited period of time. The model shows very low skill in reproducing either magnitude or
seasonal cycle at three sites with long-term observational records; namely Auchencorth Moss
(Scotland, UK), Blodgett Forest (California, U.S.A), and Ulborg (Denmark). In all three cases



the model severely underestimates $O_3$ dry deposition fluxes. The model also shows a fairly
low skill in reproducing the seasonal cycle at these three sites. Potential reasons for the low
model skill at these long-term observation sites include modelled surface ozone levels,
deposition velocities and the appropriateness of the vegetation type, but more detailed
analysis is required to explore these further. However, by and large, the model performance
appears reasonable when compared to both observations and other models, although with an
overall negative bias.
**4.3 Model simulated methane and OH**
Here we discuss the performance of UKCA StratTrop modelled methane and OH distributions
in the troposphere. OH is the primary oxidising agent in the troposphere and is the key
determinant on the burden of methane in the troposphere (Monks et al., 2015).
A commonly cited indicator of tropospheric oxidising capacity, the tropospheric lifetime of
methane with respect to OH has been calculated for the FR simulation, averaged over the
entire length of the run. The modelled average tropospheric mean methane lifetime with
respect to OH oxidation is calculated to be 8.5 years ( with a standard deviation of 0.1 years).
This value is in good agreement with the ACCMIP ensemble average of 9.7 ± 1.5 years (Naik
et al., 2013) (i.e. falling within one standard deviation of the ACCMIP ensemble). We note that
the methane lifetime for UKESM1 is much shorter than the methane lifetime for HadGEM2ES
and highlight that Figure 4 shows this is largely down to improvement in the treatment of
photolysis since HadGEM2ES (Telford et al., 2013).
We further focus our analysis on comparing the climatological distribution of OH as a function
of latitude and altitude (Figure 12).

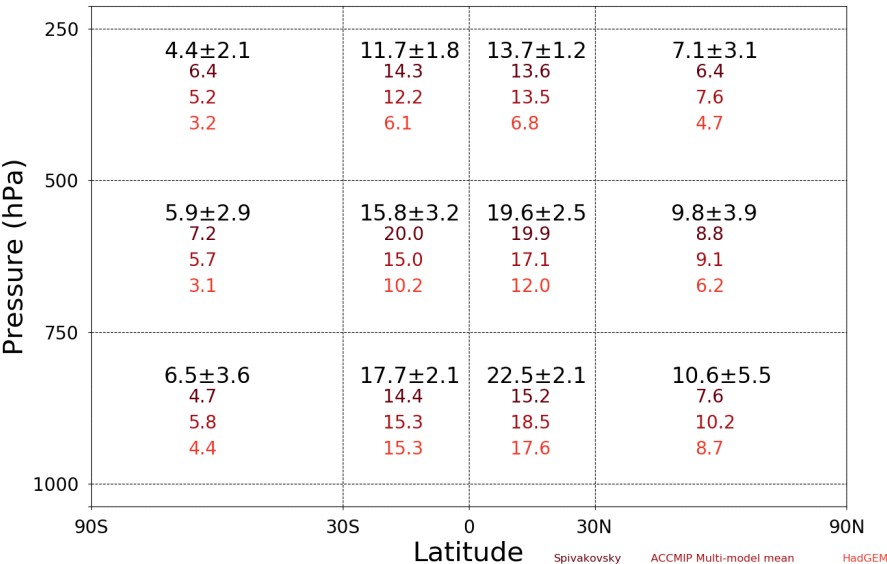



**Figure 12.** Evaluation of the UKCA StratTrop zonal distribution of tropospheric [OH] ( x $10^5$ molecules $cm^{-3}$). Values plotted in black refer to the UKCA StratTrop multi-annual mean [OH] in each region of latitude and pressure range, with the values following being ± 1 standard deviation around the mean.

The UKCA StratTrop simulations result in a global mean tropospheric [OH] of 1.22 x $10^6$ molecules $cm^{-3}$, averaged over the period 2005-2014 in the FR simulation. As with the methane lifetime, this value is slightly higher than the ACCMIP ensemble mean (11.1 ± 1.6 x $10^5$ molecules $cm^{-3}$) but sits within the standard deviation of the ACCMIP ensemble mean (Naik et al., 2013). Figure 12 shows how the distribution of [OH] varies throughout the troposphere relative to the ACCMIP multi model mean, the HadGEM2-ES model and the data from Spivakovsky et al. (2000) who pioneered the development of [OH] climatologies in the troposphere. Compared against these data, UKCA StratTrop in UKESM1 performs well: The global tropospheric mean [OH] is within 10 % of the ACCMIP ensemble mean. The model captures the latitudinal and vertical profiles found in the other data sets and agrees on the magnitude of [OH] in 10 of the 12 regions analysed (when considering the model uncertainty).

The [OH] is higher in UKCA StratTrop than in HadGEM2-ES, partly because of different emissions used in the HadGEM2-ES study, but also in part owing to the change in photolysis scheme (as discussed previously). UKCA StratTrop agrees better with the ACCMIP multi model mean than Spivakovsky or HadGEM2-ES, but the tropics from 1000-750 hPa are regions where the model consistently disagrees with the other datasets, simulating higher levels of OH in these regions. These regions of the troposphere are the regions where most $CH_4$ is oxidised and so high biases in the model here will tend to lead to lower $CH_4$ lifetimes in the model than in observation-derived estimates.

In the previous configurations of UKCA (MO09 and OC14), methane concentrations fell off too quickly with height above the tropopause; this was attributed to the stratospheric transport timescale being too long in the respective physical model. Comparisons of methane columns from the HadGEM2-UKCA coupled model with SCIAMACHY, for example, were too low and required modelled methane above 300 hPa to be overwritten with Halogen Occultation Experiment (HALOE, Russell et al., 1993) and Atmospheric Chemistry Experiment (ACE, Bernath et al., 2005) assimilated output from TOMCAT (Hayman et al., 2014). Figure 13 shows that the fall-off of methane with height in UKESM1 is less rapid than in HadGEM2 and is consistent with the age of air in the model being comparable to that inferred from observations (Section 4.6.2). As comparisons with surface observations and SCIAMACHY (with its strong sensitivity to surface concentrations) are not appropriate here, because surface methane concentrations are relaxed to LBCs (Section 2.6), only comparisons with stratospheric observations are shown here.



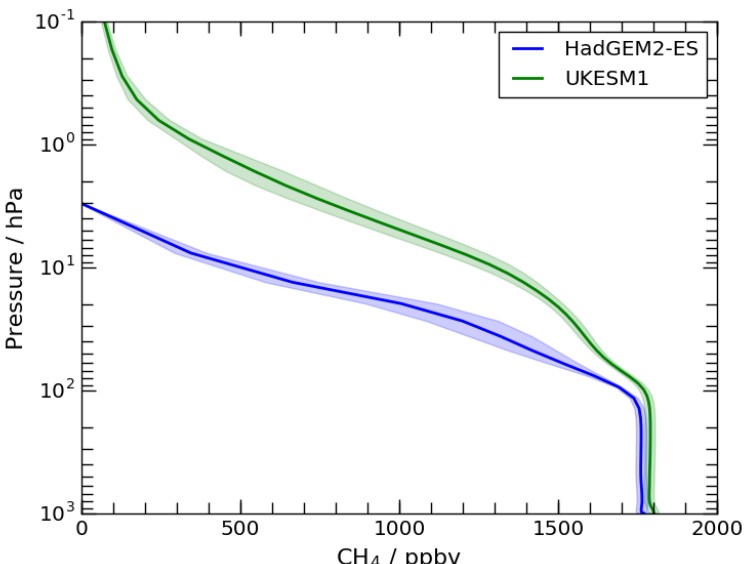

**Figure 13.** Vertical profiles of the mean tropical (±10 °N) modelled methane from multi-annual annual
mean output from atmosphere-only simulations of HadGEM2-ES (OC14) and UKCA StratTrop in
UKESM1 (this study).
Figure 14 shows multi-annual zonal mean comparisons for January and July of modelled
methane against the HALOE/Cyrogenic Limb Array Etalon Spectrometer (CLAES) climatology
(Kumer et al., 1993). It indicates that UKCA StratTrop in UKESM1 is capable of simulating the
absolute concentrations as well as the morphology of the observed distribution. The only
exception to this is the tongue of methane-depleted air descending from the mesosphere over
the SH high latitudes in July, which was also evident in MO09. Nevertheless, UKESM1 is able
to capture the observed vertical fall-off with height. There is an excellent 1:1 correspondence
between the model and observations: the slope of the least squares fits for January and July
are within 0.05 of unity, the correlation coefficients are greater than 0.98 and the root mean
square errors between UKESM1 and the HALOE/CLAES climatology are less than 0.1 ppm.



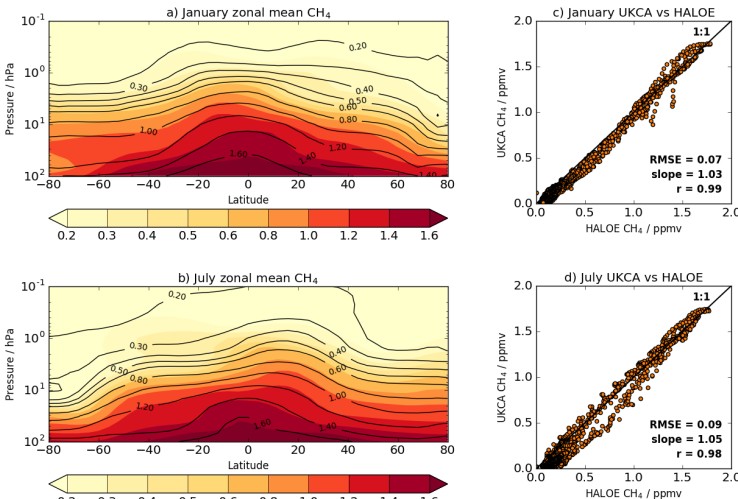

**Figure 14.** Multi-annual monthly zonal mean methane in ppm in a) January and b) July, with scatter plots of modelled versus observed concentrations for January and July in panels c) and d), respectively. The coloured contours in a) and b) are from UKCA and the black contours are the HALOE/CLAES climatology. The scatter plots also include a 1:1 line and metrics such as root mean square error (RMSE), the slope of a least squares linear fit, and the correlation coefficient (r).

## 4.4 Comparison of model simulated and observed distribution of $H_2O_2$.

Whilst not as important an oxidant as OH or $O_3$ in the gas phase, $H_2O_2$ is important for the oxidation of $SO_2$ in the aqueous phase. Indeed, in our UKCA StratTrop simulations roughly 30% of $SO_2$ is oxidised by $H_2O_2$ in the aqueous phase and as such it is important to evaluate the performance of our simulation of $H_2O_2$. Unlike $O_3$, there are no satellite products available to give global coverage, or indirect measures of its abundance as in the case for OH with regards to the lifetime of $CH_4$. As such we rely on in situ observations to make an assessment of model performance.

Figure 15 compares the vertical profiles of the decadal average $H_2O_2$ mixing ratios from the FR simulation described above in red and in blue one of the older UKCA StratTrop simulations described in Section 2.8 (i.e. Figure 4 blue dots) with observations from a range of aircraft measurement campaigns in grey. This analysis builds on the work of Emmons et al. (2000) and the secondary y axis on each plot (right hand side) indicates the number of observations that make up the observed mean and standard deviation. Each panel in Figure 15 shows the vertical profile of $H_2O_2$ sampled at different times (monthly averages for the model and periods indicated on the top of the panels for the observations).

The general feature of Figure 15 is that the UKCA StratTrop in UKESM1 simulations tend to result in higher levels of $H_2O_2$ than in the previous version of UKCA. There is generally good agreement with the observed vertical profile in most locations. The model simulations tend to underestimate the variability shown in the observations. Nevertheless, caution should be applied when assessing the spread of the observations as many of the campaigns had specific



foci to target chemical or meteorological events while here they are compared to monthly
mean $H_2O_2$ in the model.

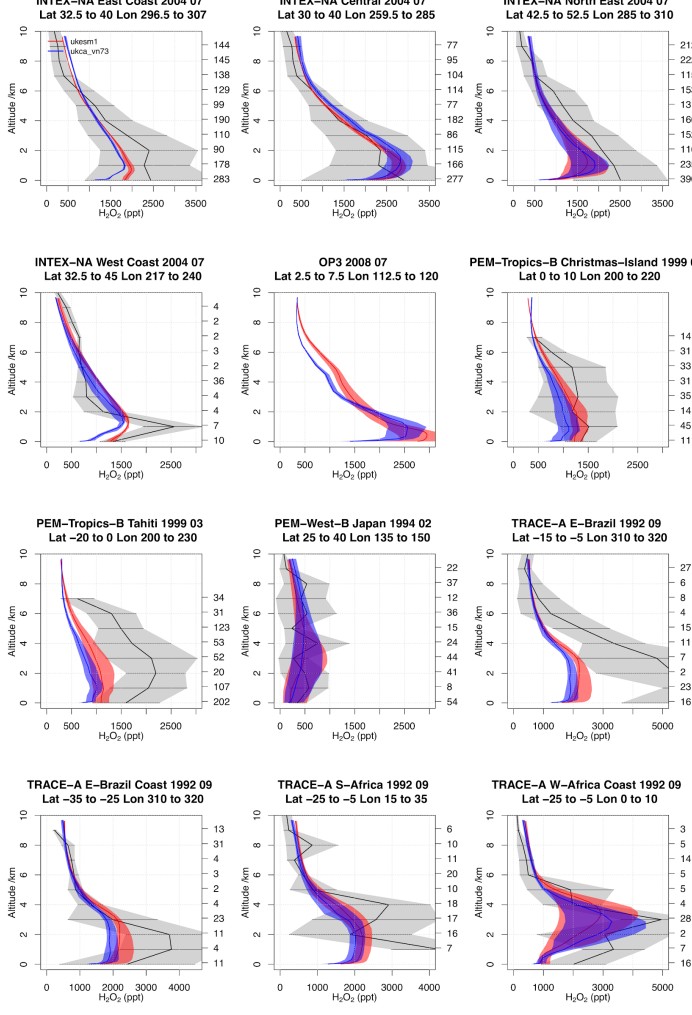

**Figure 15.** Comparison of modelled (red and blue) $H_2O_2$ vertical profiles against observations from
aircraft campaigns (grey). The red profiles indicate the results from the free running UKCA StratTrop in
UKESM1 simulation described here and blue for results from an older UKCA StratTrop simulation
running in MetUM vn7.3, (Experiment G, see Supplement for details). All plots show the mean as the
solid line and the envelope being ±1 standard deviation.

### 4.5 Comparisons with satellite retrievals of tropospheric columns of $O_3$, CO and $NO_2$

Here we compare the results from the UKCA StratTrop runs against satellite data with a focus
on assessing performance in the troposphere. In all cases, the runs analysed are the nudged





dynamics runs (ND) discussed in Section 3. Nudging enables a more robust comparison against the satellite observations as it reduces biases caused by circulation errors in the free running model, although we note that it does not completely remove these biases (Orbe et al., 2018). As well as nudging, the model output is sampled instantaneously every three hours to allow for time and space sampling to the satellite data locations. The comparison between the model and the observations is made using OMI-MLS for the tropospheric column of $O_3$, MOPITT for the tropospheric column of CO and OMI for tropospheric column of $NO_2$.

In the following analysis, the stratosphere has been removed by screening out regions where the monthly mean ozone exceeded 125 ppb, the ozonopause; columns are calculated by summing variables from the surface to the height at which the ozonopause starts. The model ozone data presented here has not been corrected to account for optically thick clouds in the troposphere which may affect retrieved ozone profiles (Ziemke et al. 2006) since averaging kernel (AK) information is not available for the OMI-MLS dataset. Since satellite measurement errors were not available, we have used $2 \times$ standard deviation of the retrievals to estimate when the differences between modelled and observed ozone are significant. This implies that the stippling area in the plots, corresponding to grid cells where |model bias| > satellite error, could be reduced (i.e. better agreement with the observations) if the satellite error is added to the $2 \times$stdev. The plots therefore show a 'worst case scenario'.

The model fields have been co-located in time and space with the observations to reduce representation errors. For each satellite retrieval, the nearest model grid box is sub-sampled within 3 hours of the observation and the model profile interpolated onto the satellite pressure grid. The satellite AKs (where available) are then applied to the model profile, to account for the vertical sensitivity of the instrument. Then the model sub-columns are calculated and summed between the surface and the tropopause to determine the co-located model tropospheric column. The equations used to apply the OMI $NO_2$ and MOPITT CO AKs to the model profiles are:

$y = A.x$ and $y = 10^{\wedge}(A(\log10(x)-\log10(xa))+\log10(xa))$

where $x$ is the co-located model profile interpolated onto the satellite pressure grid, $A$ is the satellite averaging kernel, $xa$ is the satellite apriori and $y$ is the modified model profile. Here $x$ for $NO_2$ is in sub-columns with units of ($10^{15}$ molecules cm$^{-2}$), while $x$ for CO has units of vmr before conversion into sub-columns/tropospheric column. Tropopause height information was provided by the OMI $NO_2$ files, but for MOPITT derived tropospheric column CO we use the climatological tropopause, described by Monks et al. (2017).

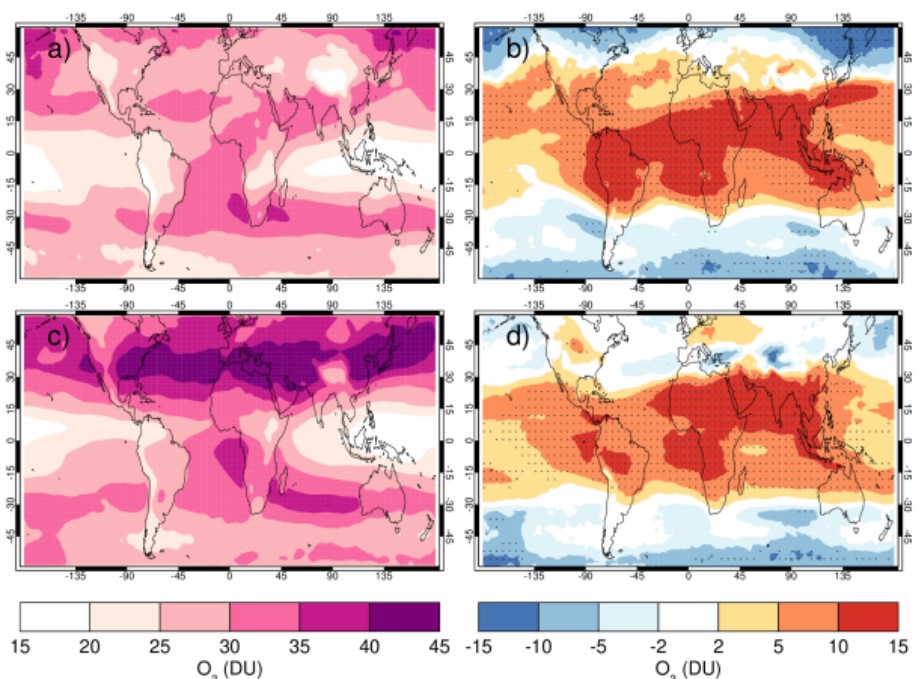

**Figure 16.** Comparison of observed and modelled tropospheric ozone columns (DU) from the ND
simulations. Plots show seasonal means and differences for the period 2005-2014.
a) OMI-MLS tropospheric column (DJF); b) difference between the model *nudged* simulations and
OMI-MLS tropospheric column (DJF); c) OMI-MLS (JJA) tropospheric column; d) difference between
model and OMI-MLS tropospheric column (JJA). Stippling indicates gridpoints where |bias| > 2×stdev
of obs.

The modelled tropospheric ozone column (TC_O3) is evaluated against the OMI-MLS
tropospheric ozone column (Ziemke et al., 2006). The general agreement between UKCA
StratTrop and OMI-MLS is good and in line with many other CCMs (Young et al., 2013). A
general feature of the model is a small underestimation in the tropospheric ozone column in
the southern hemisphere extratropics, generally good agreement in the northern hemisphere
extratropics but significant positive biases in the tropics. The underestimation in tropospheric
ozone in the Southern mid-latitudes is worse in the late summer and early autumn when OMI-
MLS shows a seasonal maximum in the Southern Hemisphere that the model fails to
reproduce (Figure 17 bottom panel).

For the Northern mid latitudes, Figure 16 panel b, shows that in DJF the model overestimates
tropospheric ozone over large parts of the North Atlantic Ocean while underestimating it over
Northern Russia and large parts of the North Pacific Ocean. These two biases counteract each
other in the timeseries plot (Figure 17, top panel) to give good net agreement overall. It is
worth noting that the timeseries plots show that there are very small, if any, trends in
tropospheric column ozone when averaging across these large domains. Fig 16 panel d and
Figure 17 top panel show that in JJA the model biases in the Northern mid latitudes are
generally very small and the amplitude and phase of the modelled seasonal cycle is in good



agreement with the OMI-MLS data. In the Tropics the differences shown in Fig 16 panel b and
d are around 25-50%. There are potentially several causes for this including (a) the
representation of chemistry in this region, (b) the underlying emission inventories (c) the
deposition rates (which are on the low end of compared with other models) and (d) the
emissions of ozone precursors. The pattern of the bias strongly resembles patterns in the
emissions of $NO_x$ from lightning. It has been noted before that the modelled tropospheric
ozone is extremely sensitive to the average global $NO_x$ emitted by lightning, which is mainly
centred around the tropics. The model bias in the tropics might be a result of the simplified
parameterisation of lightning $NO_x$ emissions and further work will focus on reducing this bias.

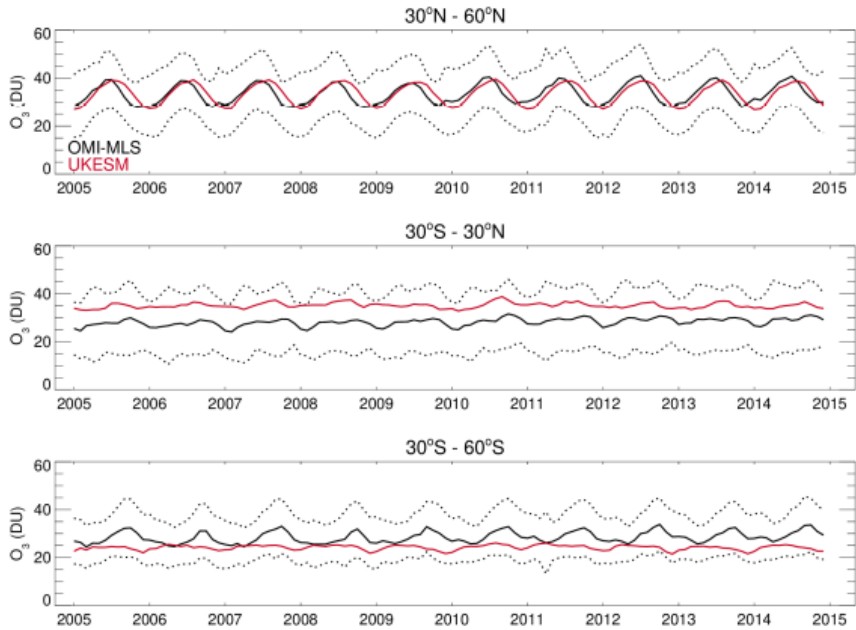

**Figure 17.** Tropospheric column $O_3$ (DU) zonal time-series (30-60°N - top panel, 30°N-30°S - middle
panel, 30-60°S - bottom panel) for model (red) and OMI-MLS (black). Dashed lines represent the
satellite uncertainty range (+/- 2×stdev).



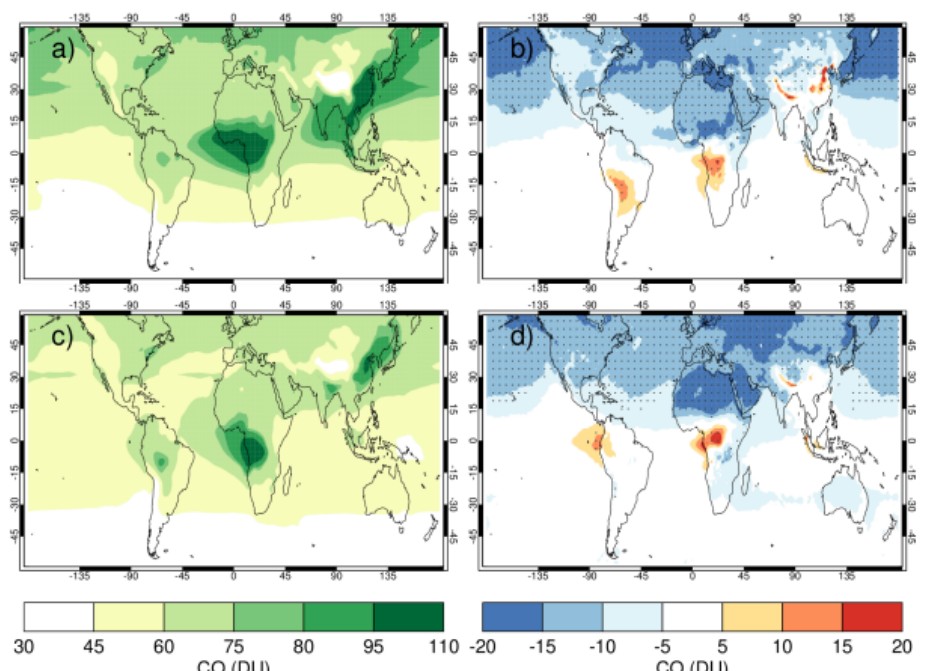

**Figure 18.** Comparison of observed and modelled tropospheric CO columns (DU) from the ND
simulations. Plots show seasonal means and differences for the period 2005-2014.
a) MOPITT tropospheric column (DJF); b) difference between model and MOPITT tropospheric
column (DJF); c) MOPITT (JJA) tropospheric column; d) difference between model and MOPITT
tropospheric column (JJA). Stippling indicates gridpoints where |bias| > satellite error.
Figure 18 shows a comparison of tropospheric column of CO in the UKCA StratTrop nudged
dynamics runs with retrievals from the MOPITT instrument on board (Terra, Emmons et al.,
2004). The MOPITT data reveal that the tropospheric column CO (TC_CO) is highest over
anthropogenic and biomass burning emission regions, and lowest over the remote oceans.
There is a strong north south gradient which is set up from the short lifetime of CO (~ 30 days)
and the time scales for interhemispheric mixing. (NB Figure 18 c highlights strong emissions
of CO in DJF in the Northern midlatitudes). The general feature evident from Figure 18 is that
the model significantly underestimates TC_CO in the northern hemisphere (NH), in both winter
and summer seasons. The negative bias in TC_CO is especially large at high northern
latitudes, consistent with surface CO biases in this region (e.g. Shindell et al., 2006). Whilst
the NH shows a negative bias, there is a strong positive bias in CO in regions associated with
agricultural (Indo gangetic plains) and forest burning (central Africa and northern South
America).
There are a number of reasons for the model-satellite biases in TC_CO, including 1)
insufficient secondary production of CO from non-methane VOC oxidation (e.g. Grant et al.,
2010), 2) excess biomass burning emissions in the southern hemisphere (SH) during DJF
(potentially the same cause in central Africa in JJA), 3) strong loss through OH in the NH in
both seasons. We note that these types of biases are not unique to UKCA StratTrop and that
further work is required to ameliorate them (Shindell et al., 2006).

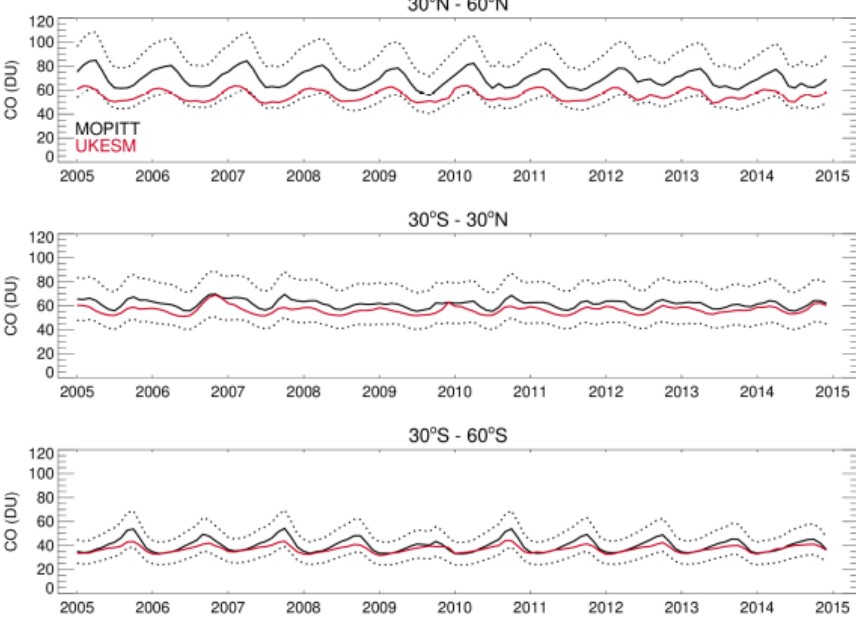

**Figure 19.** Tropospheric column CO (DU) zonal time-series (30-60°N - top panel, 30°N-30°S - middle
panel, 30-60°S - bottom panel) for model (red) and MOPITT (black). Dashed lines represent the
satellite uncertainty range.
As shown in Figure 19, there is no clear trend in modelled and observed TC_CO over time.
However, both data sets show seasonal cycles in TC_CO in the NH and SH with a very muted
seasonal cycle in the tropics. The model simulations again underestimate (~10-20 DU)
TC_CO in the NH mid-latitudes but successfully capture the amplitude and phase of the
seasonal cycle (albeit with a slightly smaller amplitude) and the magnitude of interannual
variability well. In the Southern Hemisphere, the model is doing very well capturing absolute
concentration, seasonal cycle and interannual variability, although it underestimates the peaks
during the Austral winter. There is also an underestimation of CO in the tropics despite the
positive bias over biomass burning areas.

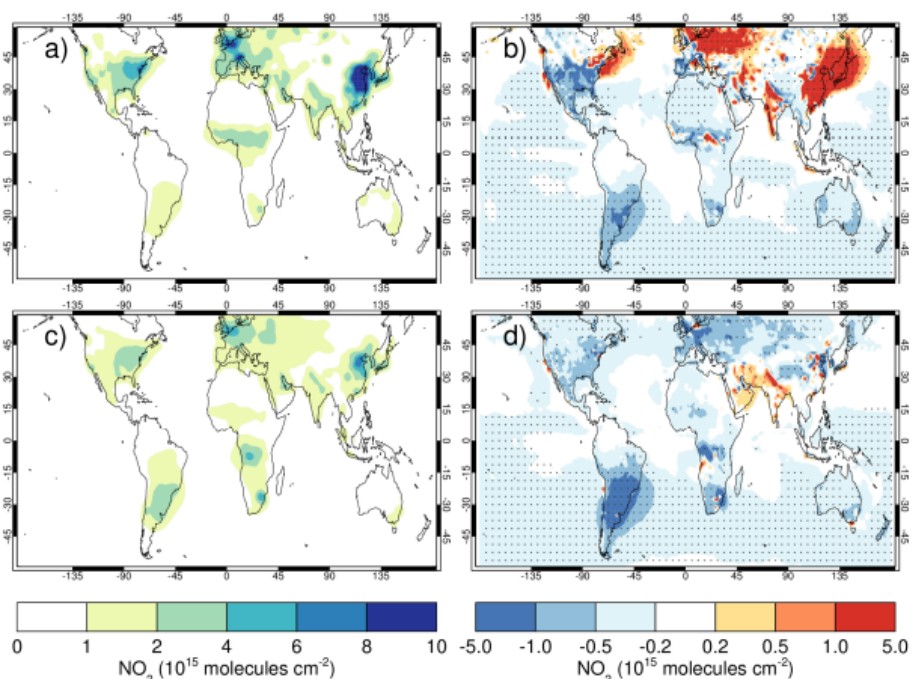

**Figure 20.** Comparison of observed and modelled UKCA StratTrop in UKESM1 tropospheric NO$_2$
columns (molecules cm$^{-2}$). Plots show seasonal means and differences for the period 2005-2014. a)
OMI tropospheric column (DJF); b) difference between model and OMI tropospheric column (DJF); c)
OMI (JJA) tropospheric column; d) difference between model and OMI tropospheric column (JJA).
Stippling indicates gridpoints where |bias| > satellite error.
Finally we focus on the comparison of modelled and observed tropospheric NO$_2$ columns. The
observed tropospheric NO$_2$ column (TC_NO2) data come from the OMI instrument on board
AURA (Boersma et al., 2007).
The observed NO$_2$ column is highly heterogeneous and localised to the major industrialised
regions, where anthropogenic emissions are highest, and major biomass burning zones
(Figure 20). The figure highlights strong seasonal differences in the observations, with
TC_NO2 being larger in winter (panel a) than in summer, most likely as a result of higher
emissions and longer NO2 lifetime than in the summer. Averaged across the whole
troposphere, the model compares well with OMI TC_NO2 spatially (Figure 20). However, there
are very significant positive biases over the main anthropogenic emission regions (i.e. South
Asia, Eastern Europe, East Asia and outflow from the US eastern seaboard), particularly in
local winter. These biases in TC_NO2 are only weakly correlated with the biases in TC_O3 in
these regions, suggesting different causes and they are dominant in different regions of the
atmosphere (boundary layer vs free troposphere). A high bias in TC_NO2 extends out from
the North China plains region, across the sea of Japan and into the Pacific ocean suggesting
either errors in the underlying emission inventory or in the modelled NO$_2$ lifetime.
Over biomass burning regions, there is evidence for low biases over central Africa and South
America (mainly in JJA). This may well be a vertical sensitivity issue in the comparison of the
data sets. As OMI has peak sensitivity in the mid-upper troposphere, OMI detects enhanced
$NO_2$ values over biomass burning regions due to the buoyant fire plumes. In UKEMS, the
anthropogenic emissions are injected on the surface level, so most of the $NO_x$ will be trapped
in the boundary layer where OMI is less sensitive. Therefore, the satellite AKs will give this
sub-column less weighting and a negative bias occurs.

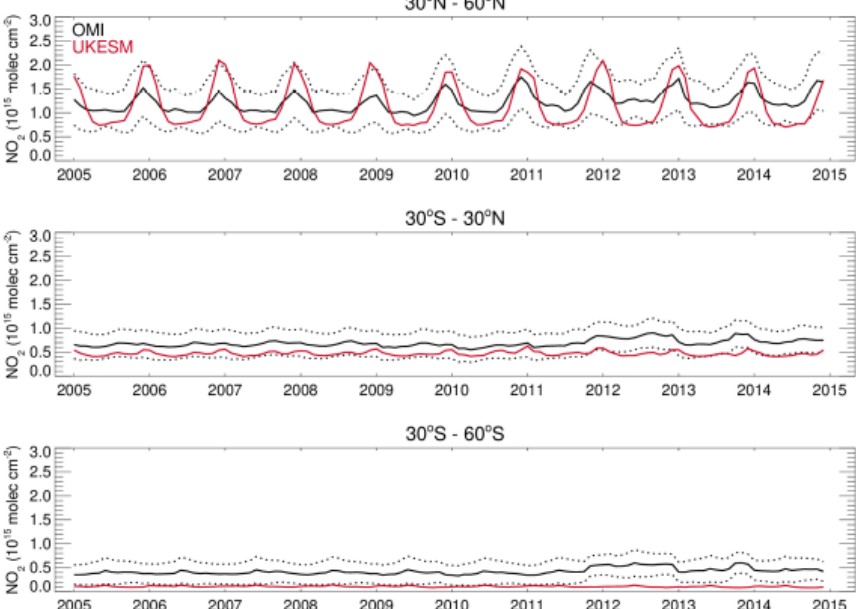

**Figure 21.** Tropospheric column $NO_2$ ($10^{15}$ molecules cm$^{-2}$) zonal time-series (30-60°N - top panel,
30°N-30°S - middle panel, 30-60°S - bottom panel) for model (red) and MOPITT (black). Dashed lines
represent the satellite uncertainty range.
Figure 21 highlights that in both the model simulations and satellite data, the average Southern
Hemisphere extra tropical TC_NO2 is lower than in the Northern Hemisphere, due to fewer
emission sources. However, in the model there is a significant low bias in this region, ~ 50%.
This bias is largest over the oceans and may be connected with biases in the representation
of $NO_y$ species (i.e. PAN) which are large contributors to NOx in this region.
In the Northern extratropics, the model simulated TC_NO2 is within the observational
uncertainty but with too large a seasonal cycle, the simulated mean annual minima/maxima
being much lower/higher than the observed mean annual minima/maxima.
**4.6 Evaluation of zonal mean stratospheric composition**
Sellar et al. (2019a) provide an overview of the simulation of total-column ozone. They show
that UKESM1 produces relatively realistic ozone fields albeit with some remaining issues.
Amongst these is a tendency for the Antarctic ozone hole to be too persistent, insufficiently



variable, and on average too deep. This is linked to a stratospheric cold bias noted before
(Dennison et al., 2019).
In the analyses below, UKCA StratTrop seasonal- and zonal-mean composition fields from
the FR simulation are compared to the Atmospheric Chemistry Experiment -- Fourier
Transform Spectrometer (ACE-FTS) climatology. ACE-FTS is a recent satellite mission
sponsored by the Canadian Space Agency; it covers a substantial number of species with a
coverage extending in some cases into the mesosphere. The climatology covers the period of
February 2004 to February 2013 (http://www.ace.uwaterloo.ca/climatology_3.5.php.) Here we
focus on NO, NOy (defined here as $NO + NO_2 + HNO_3$), CO, $H_2O$, and $O_3$.

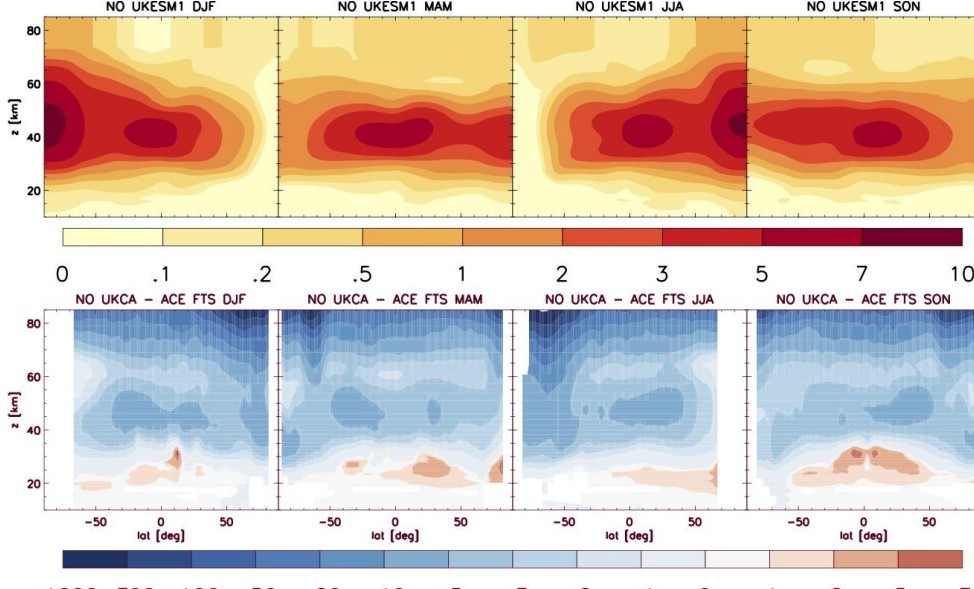

**Figure 22.** Zonal-, seasonal-, and multiannual mean nitrogen oxide (NO) volume mixing ratio, in ppb.
Top: UKCA StratTrop, February 2004 to February 2013. Bottom: Bias versus the ACE-FTS
climatology, same units and period. The climatology represents the average of AM (sunrise) and PM
(sunset) measurements.

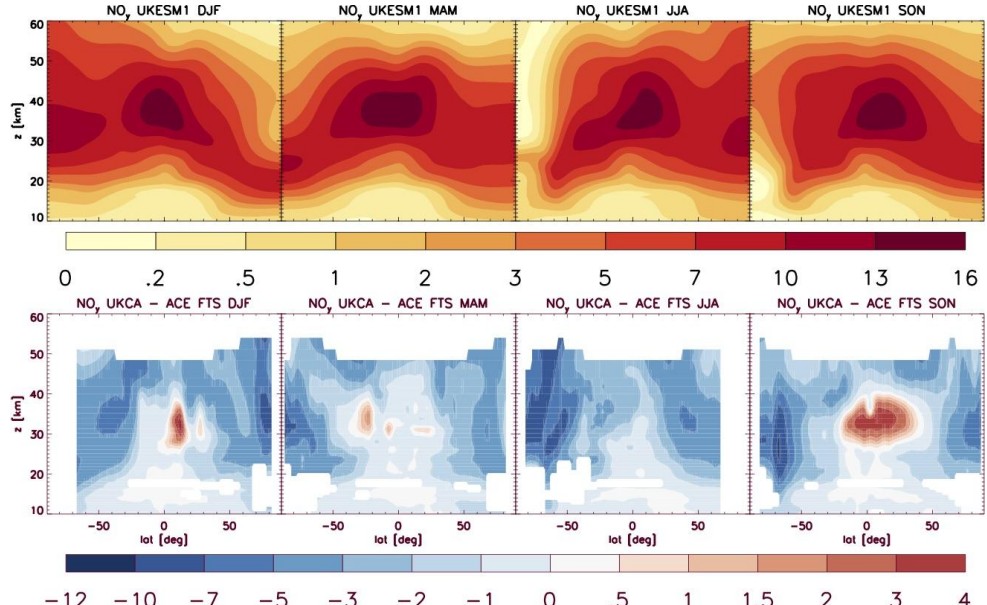

**Figure 23.** Same as Figure 22 but for odd nitrogen (NOy), defined here as $NO_y = NO + NO_2 + HNO_3$, in ppb. For ACE-FTS, NO and $NO_2$ fields are the average of AM and PM measurements.

NO is underestimated throughout the model domain (Figure 22). Away from the polar vortices, this underestimation might be associated with a sampling bias, but in the polar vortices of both hemispheres, the disagreement is so large that a sampling problem alone cannot explain the discrepancy, particularly near the model top where NO is relatively long-lived and the bias reaches 1 ppm. To overcome potential issues with sampling, we compare NOy (Figure 23). This diagnostic reveals tongues of nitrogen-depleted air descending in the polar vortices of both hemispheres which in the ACE-FTS measurements are however relatively nitrogen-rich. This discrepancy lasts into southern spring when NOy is underestimated by up to 12 ppb at around 70S. The depletion of $HNO_3$ due to denitrification in the lower Antarctic polar vortex appears to be well reproduced in winter but is perhaps overestimated in spring, in line with the generally excessively long lifetime of the polar vortex in the model (Sellar et al., 2019a; not shown).



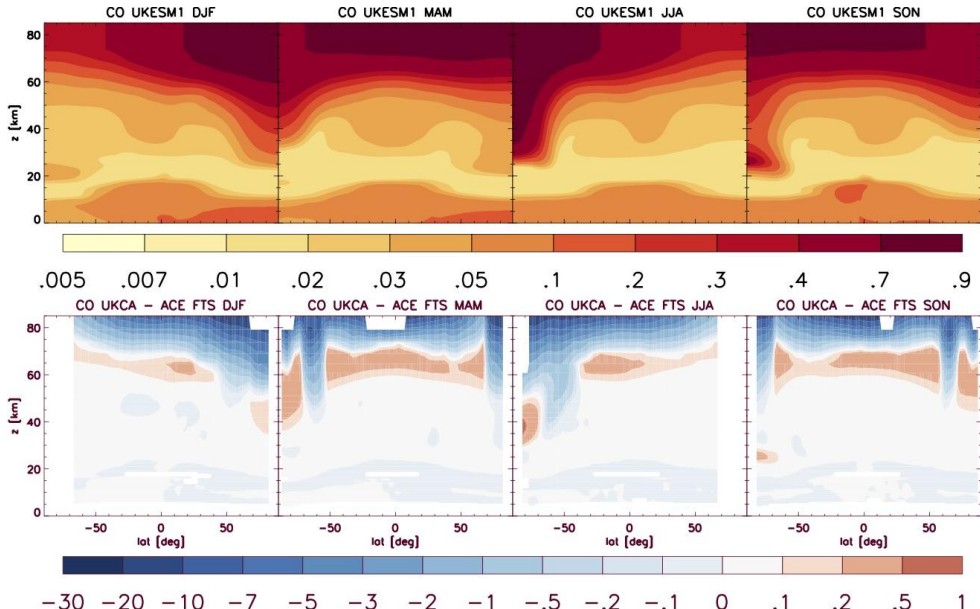

**Figure 24.** Same as Figure 22 but for carbon monoxide (CO), in ppm.

As for non-nitrogen compounds, the model gets the shape of the distribution of CO about right, but substantially underestimates the amount of CO in the mesosphere (Figure 24). A variant simulation with a modified top-boundary condition (TBC), whereby the top two levels are not overwritten with the third-highest level, reveals that with this variant TBC CO would now be overestimated. Essentially, CO production is due to $CO_2$ photolysis which is extremely height sensitive. The simulation shows that mesospheric air reaches the lower polar vortex in Antarctic spring; this process is relatively well simulated in the model.

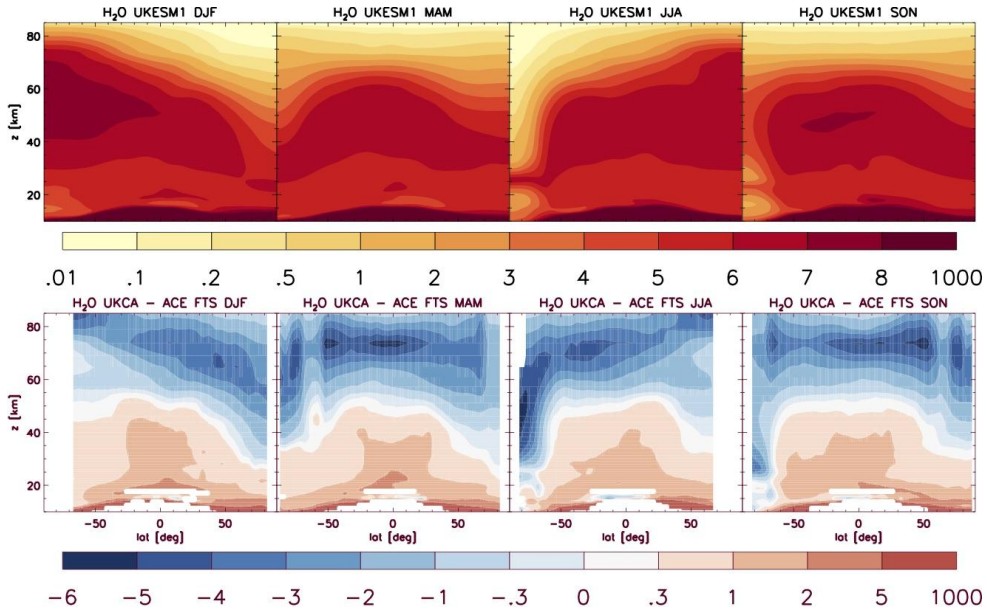

**Figure 25.** Same as Figure 22 but for water vapour ($H_2O$).

In much of the stratosphere, $H_2O$ is overestimated by 0.3 to 2 ppm, suggesting that perhaps the tropical-tropopause cold point is still slightly too warm (Figure 25). This has been a persistent problem in the MetUM coupled to UKCA (Morgenstern et al., 2009) and a significant amount of work identified remedies to this issue in earlier versions of UKCA StratTrop (Hardiman et al., 2015). One cause highlighted by Hardiman et al. (2015) was the role of ozone in the upper troposphere / lower stratosphere (UTLS) region region. Biases in ozone here are important to this issue of stratospheric moistening. In addition a new development in UKCA StratTrop has been the interactive simulation of $H_2O$ from $CH_4$ oxidation in the stratosphere and so biases in $CH_4$ or the transport of $CH_4$ into the stratosphere may also play a role. Further work will focus on understanding the causes of this $H_2O$ bias. In the mesosphere and in the polar vortices, however, $H_2O$ is underestimated by several ppm in many locations. Unlike all other gas phase chemical species, $H_2O$ is not subject to the overwriting of the top two levels. It photolyses at similarly short wavelengths as $CO_2$ (see above); an overestimation of its photolysis explains the mesospheric bias.





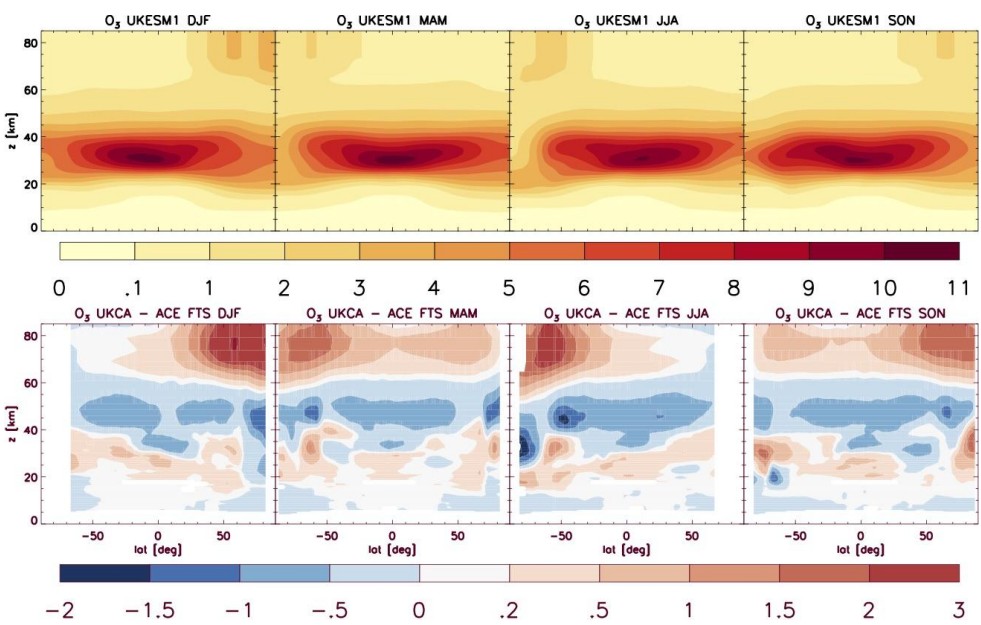

**Figure 26.** Same as Figure 22 but for ozone (O₃), in ppm.

Figure 26 highlights a good simulation of stratospheric ozone in UKCA StratTrop. In the lower stratosphere, ozone is mostly overestimated (by around 0.2 to 1 ppm), whereas in the upper stratosphere it is underestimated by similar amounts. Larger underestimations exist in Antarctic winter. In the mesosphere, ozone is generally overestimated.

Taken together, these disagreements indicate some progress with the simulation of odd nitrogen compounds albeit with substantial remaining problems. $HNO_3$ is now in better agreement with observations than documented by Morgenstern et al. (2009); however this appears to be mostly the case because ACE-FTS finds considerably more $HNO_3$ in the stratosphere than the older data used there. The substantial deficit of NO in the mesosphere is the result of missing model physics: Energetic particle precipitation (EPP) is well documented to cause the break-up of nitrogen molecules and the formation of $NO_x$ (for a review see e.g. Sinnhuber et al., 2012), but this process is not represented in UKCA StratTrop. This model deficiency results in a misrepresentation of odd nitrogen descending in the polar vortices towards the ozone layer. This might explain the $NO_y$ deficit in winter/spring over both poles, although further studies are needed to confirm this. This problem is receiving much more attention here than e.g. in the earlier investigation by Morgenstern et al. (2009) because the newer ACE-FTS satellite data offer much better coverage of high latitudes and altitudes than the observational references used by Morgenstern et al. (2009).

Morgenstern et al. (2009) had to artificially reduce water vapour at the tropical tropopause; the reasonable agreement found here is achieved without such an intervention. $H_2O$ loss and CO production are both the result of photolysis of molecules ($CO_2$, $H_2O$) that photolyse in the mesosphere where the photolysis rate increases sharply with height and may be sensitive to assumptions about the residual ozone column above the model top. In combination, these findings suggest that this residual ozone column (which is a parameter in the photolysis

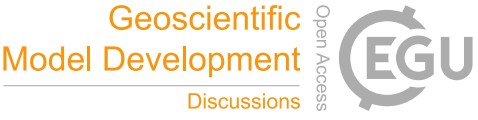

scheme) may be too small, or that making this a simple universal constant in the model may
be inadequate.
**4.7 Analysis of zonal asymmetry of ozone**
Stratospheric ozone is often validated against zonal-mean satellite data (e.g. see above). As
the simulation of ozone improves in models, attention turns to higher-order diagnostics. A
recent analysis by Dennison et al. (2017) revealed that zonal asymmetries of the stratospheric
polar vortex, in simulations by a model closely related to UKESM1, were very underestimated;
the vortex was generally too circular and its centre too close to the South Pole, when in reality
the Southern polar vortex is often distorted and displaced towards the Indian-Ocean sector.
Dennison et al. found a westward progression of this displacement, which their model failed
to reproduce. The climate impacts of ozone depletion are also often thought of in zonal-mean
terms (e.g. Kang et al., 2011); any effort to attribute regional climate change beyond the zonal-
mean to ozone depletion might well be impeded by such model behaviour. Hence here we
briefly assess how UKCA StratTrop handles zonal asymmetries of the Antarctic polar vortex.
Here we focus on the Historic UKESM1 simulations (Sellar et al., 2019a), which use the same
version of UKCA StratTrop documented here, rather than the experiments discussed in
Section 3.
The analysis consists of expanding TCO in a Fourier series:
$O_3 = ZMO3 + A \cos(\lambda + b) +$ higher order terms (ignored here)              Eq 6.
Here $O_3$ is monthly-mean total-column ozone, meridionally averaged over 60S to 70S, ZMO3
is its zonal mean, $A >= 0$ is the amplitude of the zonal asymmetry, $\lambda$ is longitude, and $b$ is the
phase shift. $b=0$ would correspond to an ozone maximum occurring at the Greenwich Meridian
and a minimum occurring at the Date Line. Positive values for b correspond to a westward
displacement of these features.

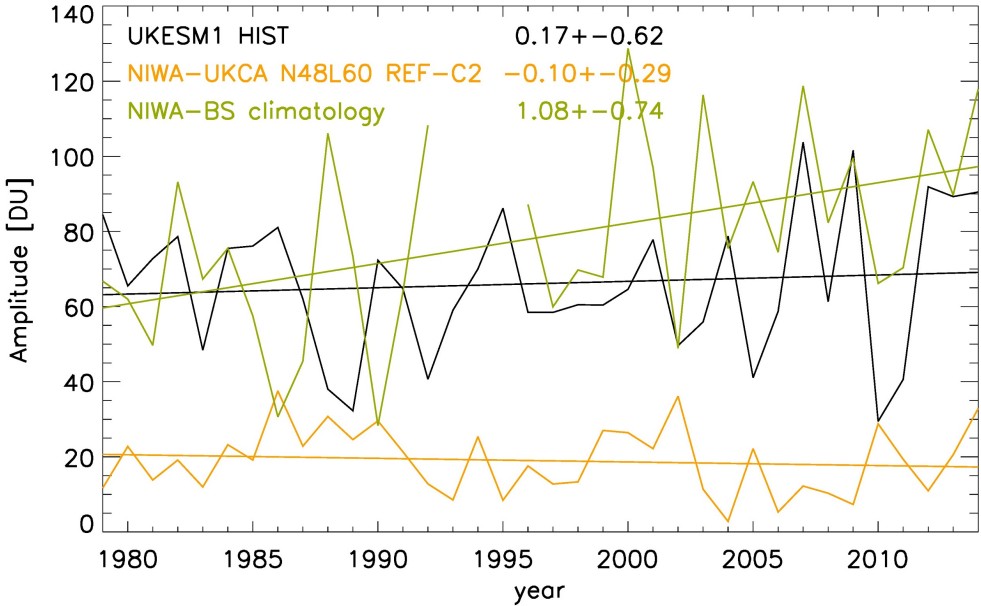

**Figure 27.** Amplitude *A* [DU] of the zonal asymmetry in total-column ozone at 60-70S in October.
Green: NIWA-Bodeker Scientific total-column ozone climatology, vn. 3.4. Orange: The model used by
Dennison et al. (2017), NIWA-UKCA. The data represent the average of 5 CCMI REF-C2 simulations
by their model. Black: UKESM1. The data represent the average of two CMIP6 "historical" simulations
(Sellar et al., 2019a). Straight lines are linear regression fits. The numbers represent mean trends and
associated 95% confidence intervals in DU/year.

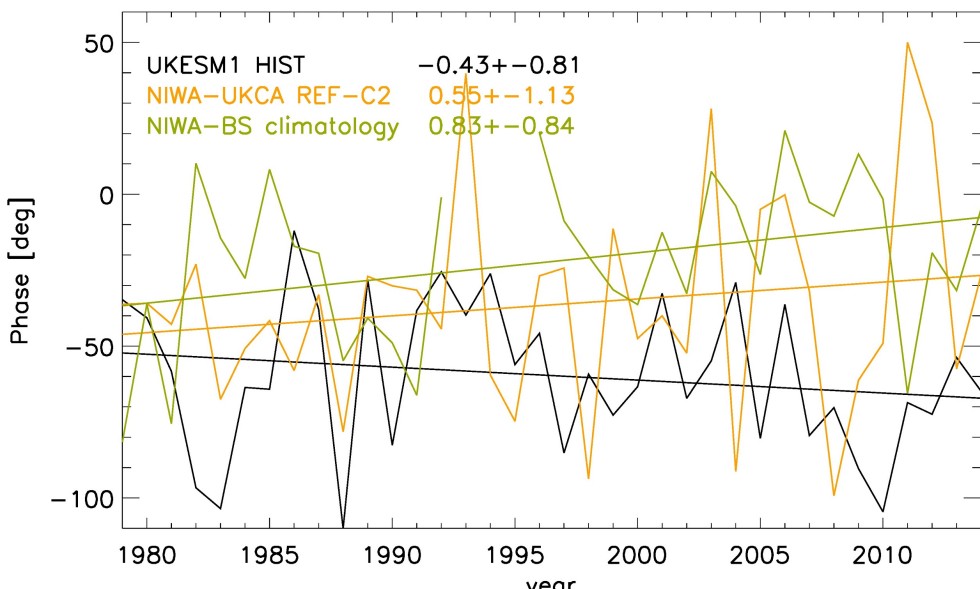

**Figure 28.** Same as Figure 27 but for the phase b, in degrees.
Figure 27 displays *A* for the months of October (when the ozone hole typically is deepest).
The    NIWA-Bodeker    Scientific    total-column    ozone    climatology
(http://www.bodekerscientific.com/data/total-column-ozone, green colour) indicates that the
zonal asymmetry is typically about 40 to 120 DU in size, and on average there is a positive
trend, with the ozone asymmetry increasing significantly by nearly 40 DU between 1979 and
2014. UKESM1 (black) reproduces the magnitude and variability of the ozone asymmetry, a
big advance over the model used by Dennison et al. (2017) (orange). The difference in the
trend is not statistically significant at the 95% confidence level. For the phase b (Figure 28) we
find that the model produces an ozone peak usually around 60-70E (i.e. in the Indian Ocean
sector) whereas in the NIWA-Bodeker Scientific climatology this maximum occurs further west,
on average around 20-30E. The mean eastward trend simulated by UKESM1 is outside the
range of possibilities for the observations (which indicate a westward trend), but the
uncertainty intervals overlap.
**4.8 Evaluation of transport and long-lived tracer-tracer correlation**
Our final aspect of model evaluation focuses on the comparison of the large-scale transport
in the modelled middle atmosphere, analysed through comparison of the modelled age of air
profiles against age of air determined using observations of $SF_6$ made by the MIPAS





instrument (Stiller et al., 2008) and through comparison of observed (ACE-FTS) and modelled
tracer-tracer correlations. The model data analysed here are from the FR simulation.
A simple but powerful way to test the representation of stratospheric chemistry in a model is
to analyse the correlations between long-lived trace gases (e.g. Chapter 6, SPARC2006).
Long-lived tracers are known to exhibit compact correlations with each other (Plumb and Ko,
1992) and comparison of modelled and observed correlations can test aspects of the model
chemistry independent of dynamics. This is particularly useful when comparing complex 3-D
climate models such as UKESM1 with observations made by a range of platforms at different
spatial resolution and coverage, and under different meteorological conditions.
Figure 29 shows the correlations of $CH_4$ vs $N_2O$, $CH_4$ vs $H_2O$ and $NO_y$ vs $N_2O$ from a present
day UKESM1 simulation (2005-2010) as well as from ACE and MIPAS satellite data. The ACE
V4 (2004-2018) data was obtained from http://www.ace.uwaterloo.ca/data.php and monthly
mean zonal mean values at $5^o$ latitude bins were created by averaging all profiles with retrieval
errors less than 100%. The Michelson Interferometer for Passive Atmospheric Sounding
(MIPAS) V1.4 data used here is an update of that used in CCMVal-2010 report (SPARC2010)
(see http://eodg.atm.ox.ac.uk/MIPAS/). Co-located profiles of $H_2O$, $CH_4$, $N_2O$, $NO_2$, and $HNO_3$
are retrieved simultaneously for both day and night time profiles and are available for the
mission period (2002-2012). MIPAS data was obtained via ftp://ftp.ceda.ac.uk/neodc/mipas-
oxford/data/.
$CH_4$ and $N_2O$ are two chemically independent, but long-lived, tracers with significant
stratospheric sinks. Accordingly, they are expected to show compact correlations in the
stratosphere (Plumb and Ko, 1992). Overall, UKESM1 seems to show very good agreement
with the recent satellite-observed relationships, suggesting that the relative loss of $CH_4$ and
$N_2O$ in the stratosphere is well represented. However, the model and the satellite observations
differ slightly from the older ER-2 in-situ lower stratospheric observations, possibly due to
different relative changes in $CH_4$ and $N_2O$ in recent years. Note also that the model simulation
covers the period 2000-2004 while ACE data covers 2004-2018, hence even after applying
the quality flag ACE $CH_4$ and $N_2O$ values in the troposphere are larger than model values.
More noticeable model-observation differences are found the $CH_4$:$H_2O$ correlation. These two
long-lived tracers are chemically linked in the stratosphere: $CH_4$ oxidation leads to the
production of nearly 2 molecules of $H_2O$ (with a small yield of $H_2$). As the maximum observed
upper stratosphere $H_2O$ mixing ratio is typically around 7 ppm, and $CH_4$ is the primary source
of stratospheric $H_2O$, the $H_2O$ vs $CH_4$ relationship is expected to be close to $H_2O + 2 \times CH_4 = 7$
ppmv, which is included in the plots as a reference. The ACE observations show a slightly
weaker relationship ($H_2O + 1.75 \times CH_4 = 6.8$) while MIPAS data shows a stronger slope, which
is larger than 2 ($H_2O + 2.4 \times CH_4 = 8.0$). There will be some uncertainty in the satellite data but
it is clear that UKESM1 has a significantly different relationship. The upper stratospheric $H_2O$
values are reasonable but the lower stratosphere seems to be much wetter compared to
observations. For example, near 90 hPa most of the ACE profiles show $H_2O$ values close to 3
ppm, whereas modelled values hardly go below 5 ppm, suggesting water vapour entry mixing
ratios near the tropical tropopause layer are not well constrained in the model. However, in
UKESM1 $CH_4$ oxidation appears to yield only 1 $H_2O$ per $CH_4$ oxidised, which allows the model
to achieve realistic upper stratospheric $H_2O$ values. Further detailed studies are required to
verify the cause of this model discrepancy. We have noted that there is a missing $H_2O$ product
in the reaction $HO_2 + MeOO$ (listed in Table 2). However, we calculate that this reaction only
accounts for 2.3% of the fate of MeOO in the stratosphere (which is dominated by reaction
with NO), so it appears unlikely that this is the source of the bias.



Finally, we compare the the NOy vs N$_2$O tracers, which are also chemically linked. N$_2$O is main
source of stratospheric NOy with a yield of about 6% via reaction of O($^1$D) (see equation 6.2b
in SPARC, 2010). ACE NOy values are calculated simply by adding the observations of HNO$_3$,
NO, NO$_2$, 2N$_2$O$_5$ and ClONO$_2$. For MIPAS, zonal mean (5$^\circ$ latitude bin) monthly mean profiles
were calculated by averaging all the measurements with standard errors less than 100%. For
NOy:N$_2$O plots, only nighttime profiles are selected (SZA >95) and NOy is calculated as
HNO$_3$+ NO$_2$+ 2N$_2$O$_5$ + ClONO$_2$. For large values of N$_2$O, the UKESM1 correlation is less
compact than the observations, although the modelled slope indicates a realistic 6.7% yield of
NOy. The model also produces a reasonable peak NOy mixing ratio of around 17 ppbv,
although this is slightly smaller than observations, in particular from ACE. The model also
tends to simulate larger occurrences of low NOy values for a given N$_2$O, which may be an
indication of strong polar denitrification.

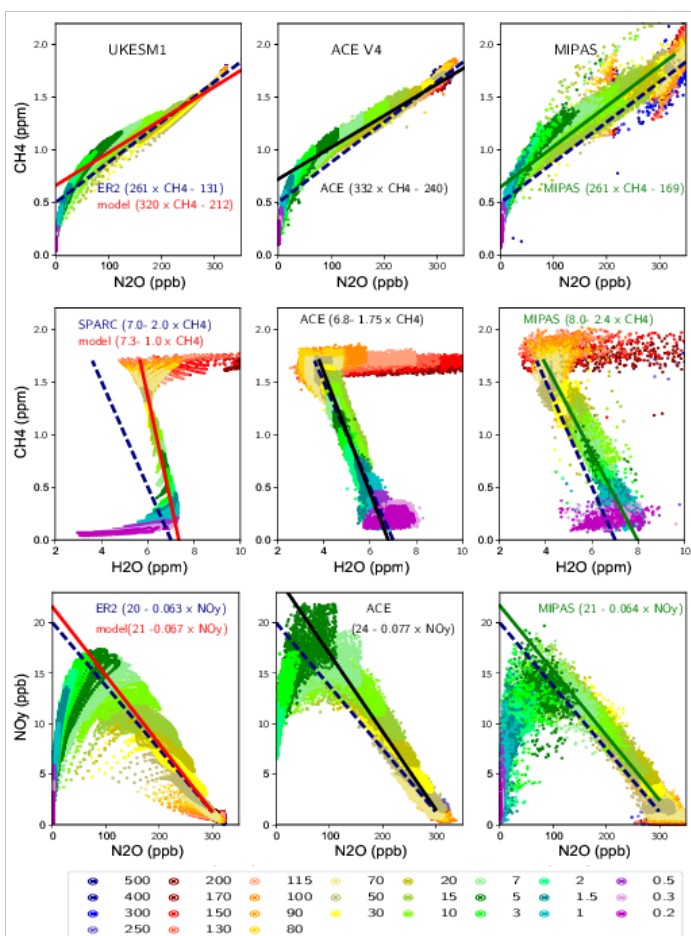

***Figure 29***: *Correlations between selected long-lived chemical species (monthly mean zonal mean*
*values for 60$^o$S-60$^o$N) from UKESM1 (left column), ACE V4 data (centre) and MIPAS data (right). The*
*coloured legend shows the corresponding pressure level of the data points. The linear regression fits*
*to the model, ACE and MIPAS data are shown in the respective panels along with the equations of*
*the lines. The MIPAS data is the same as that used in Figures 6.12, 6.13 and 6.14 in the CCMVal-2*

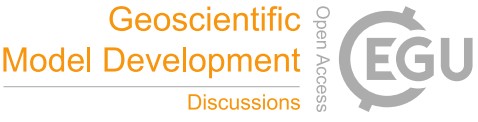

*report (SPARC 2010). ACE NOy values are calculated as NOy = NO + NO$_2$ + HNO$_3$ + 2N$_2$O$_5$ +*
*ClONO$_2$. (Top row): CH$_4$ vs N$_2$O. The linear fit is calculated for N$_2$O values ranging from 100 to 300*
*ppb. The dashed line shows estimated fit from ER-2 data (N$_2$O (ppb) = 261.8×CH$_4$ (ppm) -131, see*
*Kawa et al., 1993). (Middle row): CH$_4$ vs H$_2$O. The linear fit is calculated for CH$_4$ values ranging from*
*0.5 to 1.5 ppm. The dashed line represents H$_2$O + 2CH$_4$ = 7 ppm. (Bottom row): NOy vs N$_2$O. The*
*linear fit is calculated for N$_2$O values ranging from 100 to 300 ppb and the dashed line shows the*
*equation NOy (ppb) = 20.0 - 0.0625×N$_2$O (ppb), based on mid-latitude balloon profiles and ER-2 data*
*(see Kondo et al 1996).*

Figure 30 compares the modelled multi annual mean age of air profile in the stratosphere against observations of SF$_6$ from 2002-2010 used to calculate the age of air from the MIPAS instrument (Stiller et al., 2008). The model includes a diagnostic to quantify the age of air. This is a effectively a "species" in the model that is emitted at the model surface continually and undergoes full tracer advection and diffusion. Whilst below the modelled tropopause (based on a merger of the 380 K and 2 PVU surfaces) the tracer is set to have an age of zero; above the tropopause the tracer has its age increased every model time step that it stays above the tropopause.

Figure 30 panel (a) shows the modelled mean tropical (±10˚) age profile as a function of altitude and that there is very good agreement between the model and the observations, with an increase in the age of air as both profiles increase in altitude and a maximum age of around 5 years. The modelled northern hemisphere midlatitude (35˚ - 45˚N) age profile (panel b) agrees very well with the observations from 16 km to about 24 km, but the model tends to simulate an age of air which is younger than the observations above 24 km (up to a year difference younger). Panel (c) shows the difference between the tropical and mid-latitude profiles and further emphasises good agreement of the model with the observations below 23 km but divergence above this altitude. However, the zonal cross section at 23 km (~ 50 hPa) (panel c) shows that the model generally falls within the observational uncertainty (1 standard deviation of the multi annual observations) at all latitudes.





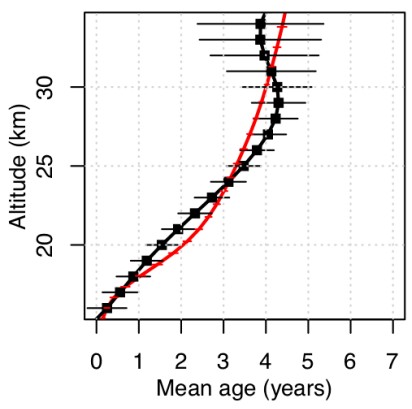
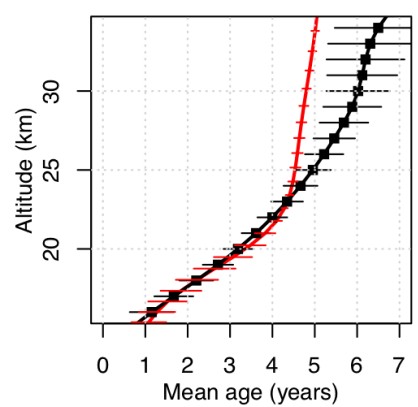
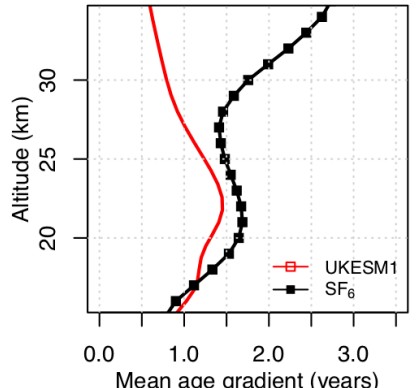
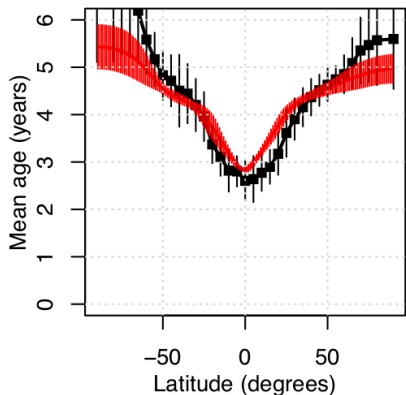

**Figure 30.** Comparison of multi-annual mean modelled (red) age of air with observations (black) from the MIPAS $SF_6$ record (Stiller et al., 2008).

**5.0 Discussion and Conclusions**

In this paper we have documented the species and reactions that make up the UKCA StratTrop mechanism for the first time and performed an evaluation of the model output for the recent past. UKCA is the key module for simulating chemical and aerosol processes in the UKESM1 Earth System model (Sellar et al., 2019a) and UKCA StratTrop enables a holistic representation of gas phase chemistry in the troposphere and stratosphere; important for understanding short lived climate forcers.

Our focus here has been to document the performance of the chemical fields simulated by UKCA StratTrop as it is implemented in UKESM1; the aerosol schemes, processes and performance are discussed in detail in Mulchay et al. (2019). Further studies are planned



which will assess the role of composition-climate Earth system couplings in the UKESM1
framework. Hence, we present simulations which have enabled a more focused assessment
of key performance indicators of the UCKA StratTrop scheme. We have analysed data from
two model runs; the first was a *free-running* (FR) simulation where the meteorology was
allowed to evolve independently based on the influence of the prescribed forcing agents
(SSTs, GHGs and sea ice) and the second was a *Nudged* (ND) simulation where the
meteorology was relaxed toward ERA-Interim reanalysis.
In general, and focusing on the gas phase as we have here, we find that the performance of
UKCA StratTrop in UKESM1 is in line with the range of models that are applied to simulating
the coupled chemistry-climate system (Young et al., 2013; 2018).
Our key performance indicators have included:
● An assessment of the magnitude and spatial distribution of lightning $NO_x$:
We note here that whilst the model simulates a global annual total lightning $NO_x$ emissions
magnitude that is in the middle of the range quoted in the literature based on observational
constraints (~ 6 Tg/yr), and the spatial distribution in lightning flash frequency matches well
with observations from satellites, the variability in lightning flash frequency is not in good
agreement with the observations (Figure 2). The UKESM1 model predicts too much lightning
activity in the tropics at an expense of the extra tropics, something which could be resolved
by moving to an ice-flux based scheme (Finney et al., 2018). Moreover, the vertical profile of
lightning $NO_x$ may have a significant impact on modelled $O_3$. Hakim et al. (2019) have shown
that across India the vertical profile in lightning $NO_x$ is very model dependent. We suggest
further work is performed to better understand the impacts of both the spatial distribution of
lightning $NO_x$ and the impacts of lightning $NO_x$ on the tropospheric column biases in $O_3$ in the
model.
● Surface ozone correlations and mean bias against TOAR observations:
TOAR (Shultz et al., 2018) provides the chemistry modelling community with an
unprecedented dataset to evaluate surface $O_3$. In our analysis of the FR and ND runs
presented here, we show that the annual mean bias is very low, but this masks out biases in
summer and wintertime (Young et al., 2018). However, we suggest that further work be
performed to understand the cause of the low and high biases in surface $O_3$, especially with
regards to how these may impact studies that use UKESM1 surface $O_3$ in health assessment
studies.
● The tropospheric oxidising capacity:
A key component to determine the lifetime of emitted reactive gases in the troposphere is the
oxidising capacity. Whilst this has to be inferred from observations (i.e. through the inferred
lifetime of methane) it is an important metric to evaluate the model against. In this study we
found that the methane lifetime in the troposphere with respect to OH was 8.5  years, within
the ACCMIP multi model range but slightly low compared to observational analyses (Naik et
al., 2013). When compared against other model estimates of the zonal distribution of OH, the
model performs well in 10 out of 12 regions analysed; with a significantly high bias in the
tropical boundary layer. This is a region where the majority of methane oxidation takes place
and may explain the slightly low modelled methane lifetime. With the recent development of
aircraft OH datasets appropriate for global model evaluation (Prather et al., 2017) we hope to



extend this analysis further and interrogate the model with these data to confirm if the bias is indeed large compared with direct observations.

• Tropospheric columns of reactive gases (CO, $NO_2$ and $O_3$):

The analysis of the model ND runs highlighted some success and failure in the models representation of tropospheric columns of CO, $NO_2$ and $O_3$. The best performance was found for $O_3$ (Figures 16-17), although we note that there is a significant bias in the tropics (which has been shown to have an effect on modelled tropospheric photolysis rates (Hall et al., 2018)). In part we believe this bias is connected with the vertical profile and magnitude of lightning NOx and further work will focus specifically on this area. The modelled tropospheric column of CO shows significant biases in the northern hemisphere (Figure 18). In part this is believed to relate to biases in the representation of higher hydrocarbons that could contribute significantly to secondary CO production (Grant et al., 2010) but high OH could also be a contributing factor. The performance of modelled $NO_2$ tropospheric columns was found to be generally acceptable in Northern midlatitudes (Figure 21) but there are large biases in regions of high emissions (such as the North China plains (Figure 20)). One hypothesis is that the model simulates too little OH in the regions of high $NO_2$ emissions, owing to lack of reactive VOC emissions and titration of $O_3$, which extends the lifetime of $NO_2$ in these regions. Further studies are required to evaluate the modelled $NO_2$ lifetime and its response to changes in emissions of $NO_x$.

• Biases in stratospheric composition

By examining extensive climatologies of observations from satellite (Figures 22-26) we've been able to show here that the simulation of stratospheric composition has improved significantly in StratTrop compared with the older "stratosphere" focused scheme of MO09. In part this is largely thanks to improvements in the dynamical model (MetUM) and improvements in biases in modelled water vapour (Hardiman et al., 2017). Key questions remain about the fidelity of the upper stratospheric/mesospheric photolysis rates and the upper boundary conditions. Given the generally poorer performance of NO and $NO_y$ it would be useful to investigate the implementation of parametrised EPP to see if this ameliorates the problems. Further work is also required to understand the cause of the disagreement between the $CH_4:H_2O$ correlation in the stratosphere, which suggests that too little $H_2O$ is produced from methane oxidation in the model.

• Middle atmosphere age of air:

The modelled middle atmosphere circulation has been evaluated against observations of $SF_6$ and through the use of tracer tracer-correlations. The tracer-tracer correlations further motivate the need for a more detailed investigation of the modelled stratospheric NOy and its budget (production and loss). The comparison of the age of air in the model generally looks acceptable in the middle stratosphere but tends to deviate at higher altitudes. In part there is more uncertainty in observations at higher altitudes (owing to loss process of $SF_6$) but further studies are required to understand if these biases are dependent on the resolution of the model. To understand this a high top, > 120 km, version of the model is in preparation as are simulations of UKESM1 at much higher horizontal resolution (~ 25 km).

In summary, UKCA StratTrop represents a substantial step forward compared to previous versions of UKCA. We have shown here that it is fully suited to the challenges of representing interactions in a coupled Earth System Model (key for CMIP6 and beyond) and we have



identified key areas and components for future development that will make it even better in
the future.
**Acknowledgements**
The authors would like to acknowledge the international community of UKCA users for all their
efforts in developing and applying the model. In particular we would like to acknowledge Prof.
John A. Pyle who pioneered the development of the UKCA project. We would especially like
to thank the atmospheric chemistry observational community who have developed numerous
datasets used in this paper to help evaluate the model. This work used JASMIN, the UK
collaborative data analysis facility.
**Data and Model Code Availability**
Due to intellectual property rights restrictions, we cannot provide either the source code or
documentation papers for the UM (including UKCA) or JULES.
*Obtaining the UM (including UKCA).* The Met Office Unified Model (MetUM) is available for
use under licence. A number of research organisations and national meteorological services
use the UM in collaboration with the Met Office to undertake basic atmospheric process
research, produce forecasts, develop the UM code, and build and evaluate Earth system
models. For further information on how to apply for a licence, see
http://www.metoffice.gov.uk/research/modelling-systems/unified-model (last access: 14
August 2019).
*Obtaining JULES.* JULES is available under licence, free of charge. For further information on
how to gain permission to use JULES for research purposes see http://jules-
lsm.github.io/access_req/JULES_access.html (last access: 14 August 2019).
*Details of the simulations performed.* UM and JULES simulations are compiled and run in
suites developed using the Rose suite engine
(http://metomi.github.io/rose/doc/html/index.html, last access: 14 August 2019) and scheduled
using the Cylc workflow engine (https://cylc.github.io/cylc/, last access: 14 August 2019). Both
Rose and Cylc are available under version 3 of the GNU General Public License (GPL). In this
framework, the suite contains the information required to extract and build the code as well as
configure and run the simulations. Each suite is labelled with a unique identifier and is held in
the same revision-controlled repository service in which we hold and develop the model's
code. This means that these suites are available to any licensed user of both the UM and
JULES.
All code related to the offline emissions is freely available on Github: https://github.com/acsis-
project/emissions and the data for biogenic emissions are available for free download
from http://eccad.sedoo.fr/. The model-satellite evaluation codes are available on request.
We acknowledge the use of the TEMIS OMI NO2 (DOMINO vn 2.0;
http://www.temis.nl/airpollution/no2.html) data and NASA's MOPITT CO (vn7.0;
https://search.earthdata.nasa.gov/) data. The observations used to evaluate age of air were
the IMK/IAA generated MIPAS-ENVISAT datasets developed at KIT and available from:
http://www.imk-asf.kit.edu/english/308.php.
**Author contributions.** ATA and NLA lead the initial development of StratTrop and ATA and
FOC lead the writing of the manuscript. NLA, FO'C, JPM, AJH, GAF, MD, ST, contributed
during model development, data analysis and paper preparation. Simulation design, setup,
and execution was performed by FO'C and MD. CH and OW provided the ozone data for the





O₃ dry deposition evaluation. All co-authors contributed to writing sections of the manuscript,
performing evaluation and reviewing drafts of the manuscript.
**Competing interests.** The authors declare that they have no conflict of interest.
**Financial support.**
MD, GAF, AJH, JPM, and FO'C were supported by the Joint UK BEIS/Defra Met Office Hadley
Centre Climate Programme (GA01101). GF and FO'C also acknowledge additional funding
received from the Horizon 2020 European Union's Framework Programme for Research and
Innovation "Coordinated Research in Earth Systems and Climate: Experiments, Knowledge,
Dissemination and Outreach (CRESCENDO)" project under grant agreement no. 641816. ST
was supported by the UK-China Research and Innovation Partnership Fund through the Met
Office Climate Science for Service Partnership (CSSP) China as part of the Newton Fund.
JMK received funding from the European Community's Seventh Frame-work Programme
(FP7/2007-2013) under grant agreement no. 603557 (StratoClim). The development of UKCA
for inclusion in UKESM1 has also been facilitated by the use of the Monsoon2/NEXCS system,
a collaborative facility supplied under the Joint Weather and Climate Research Programme, a
strategic partnership between the Met Office and the Natural Environment Research Council.
OW and CH thank the Natural Environment Research Council for support under grant
NE/K001272/1. RJP was supported by the UK Natural Environment Research Council (NERC)
by providing funding for the National Centre for Earth Observation (NCEO). CO acknowledges
funding from grant RYC-2014-15036. GZ and OM acknowledge support by the New Zealand
Government under its Strategic Science Investment Fund, and under the Deep South National
Science Challenge.

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
