# Peer review of "Description and evaluation of the UKCA stratosphere-troposphere chemistry scheme (StratTrop vn 1.0) implemented in UKESM1."

_Geoscientific Model Development, 2019_

## Referee Comment (RC1) · Anonymous Referee #1 · 23 Oct 2019

The manuscript by Archibald et al. presents an exhaustive description of the latest version of the gas-phase chemistry scheme implemented in the UK Earth System Model. The documentation includes a description of the gas-phase chemical mechanism that is applicable to the troposphere and middle atmosphere, as well as a description of associated processes such as photolysis, wet and dry deposition and interactive emissions such as lightning NOx. The discussion then moves on to presenting diagnostics of the behaviour of the model in both the troposphere and stratosphere.

The manuscript is well laid out and the presentation of the model components and diagnostics of the model behaviour are clearly discussed. My only significant criticism

is the length of the paper. I understand the need to document the many different components of the model and the authors do a good job of referring the reader to other papers for a more in-depth discussion of various model components. But the inclusion of large tables providing the list of all chemical species and reactions in the main body of the manuscript forces the reader to push through many pages before the discussion continues. I strong urge the authors to move many of these tables (Tables 1, 2, 3, 4, 5, 6, 7, 8 and 9) into the supplementary material and provide the reader with a much more compact description of the important facets of these model components. In addition, in the detailed comments for Pages 31 – 35 there is additional material on emissions that I would suggest the authors re-consider in the interest of making the paper easier to get through.

After having said all that, a few suggestions to make the paper longer... Could the authors include a very short description of how polar stratospheric clouds are treated in the model? I could not find any information on how these are calculated in the model. Note that there is a reference to Denison et al. (2018) in the caption for Table 5, but the reference is missing. And while a significant part of the paper is dedicated to assessing the stratosphere in the model there is no presentation of the total column ozone climatology that is produced by the model. Such a basic and well-observed quantity should really appear in a documenting paper.

Page 3, Line 25 – The reference for Quiquet et al. (2015) seems to be missing.

Page 21, Lines 26 – 28 - The discussion of 'Boreal and temperate forest and deforestation emissions..' is obviously referring to biomass burning but the passage could use a more explicit reference to burning.

Page 21, Line 32 – The word 'that' in 'The CEDS emissions are generally greater that those of...'

Page 23, Line 18 – The NO production from lightning is given as molecules/J, but the assumed energy of a cloud-to-cloud or cloud-to-ground flash is not given so it is

not possible to deduce the NO production per flash. Could the authors include either enough information to calculate the production per flash or the production per flash directly.

Page 29, Line 1 – In Figure 4 is it possible to add the letters of each run that appears in Table S1 to the description of each change? Something like 'New to old J-rates (B to D)'?

Page 31, Line 4 – Page 35, Line 2 – I would suggest either cutting completely or moving to the supplementary material the summary of the sectors, geographic distribution and seasonal cycle of emissions presented here. The figures and tables are based on prescribed emissions that come from independent sources and present information that is relatively well known to anyone with a background in global tropospheric chemistry – the seasonal importance of biomass burning emissions in different regions, for example.

Page 41, Line 13 – Regarding the comparison of dry deposition against field observations (Figure 11), do you compare the ozone flux from the model for a particular vegetation type that is most like the one that dominates the observation site or is the comparison using the flux averaged over the model grid square?

Page 49, Lines 10 - 14 – Looking at Figure 17b, the uncertainties on the tropical tropospheric ozone column defined as two times the variance span quite a large range and easily encompass the model results. But in Figure 16, showing the geographic distribution of the average DJF and JJA columns, large sections of the tropics show the differences as being significant. I know there are different quantities being shown, but the difference between the model and the 2xstdev is quite large. I am also a bit concerned because the upper and lower bounds on the 2xstdev do not seem symmetric around the mean. Particularly the lower bound on the 2xstdev seems to have a variation that is independent of the mean.

Page 50, Line 9 – misplaced bracket in 'MOPITT instrument on board (Terra, Emmons

et al. 2004).'

Page 50, Line 22 – Page 51, Line 2 – I believe problems with the anthropogenic emission estimates of CO has been flagged as a possible source of the general low bias of CO in the northern hemisphere – Miyazaki et al. Atmos. Chem. Phys., 15, doi:10.5194/acp-15-8315-2015, 2015, for example. But that would have been for older emissions inventories and I am not sure how that would apply to the CMIP6 inventory.

Page 53, Lines 1 – 6 - 'Over biomass burning regions... In UKEMS, the anthropogenic emissions are injected on the surface level, so most of the NOx will be trapped in the boundary layer where OMI is less sensitive.' From Page 21 the authors state the biomass burning emissions are injected over the lowest 3 km of the model, which does not seem to agree with what is stated here.

Page 62, Figure 29 – the linear regression from the model is shown in the left-most panel for each of the comparisons, but it would help greatly in the comparison with ACE and MIPAS if the regression for the model was reproduced in the other two panels.

Page 63, Lines 27-28 – It should be '(panel d)' in 'However, the zonal cross section at 23 km ($\sim$ 50 hPa) (panel c) shows...'. More fundamentally, there is no explanation for the red region in panel d.

---

## Referee Comment (RC2) · Anonymous Referee #2 · 24 Oct 2019

In this manuscript, Archibald et al aim at giving a comprehensive description and evaluation of the gas phase chemistry scheme used in the Earth System Model UKESM1. A first part is dedicated to the description of the implementation of the physical and chemical processes in the model, a second part to a very detailed evaluation of the model, from deposition on the ground to the upper atmosphere.

In general, this manuscript is very well written and pleasant to read, due to its very clear organization. Many details are given, and they all are usefull to get a precise description of the model, but it gives a very long manuscript. Maybe some tables (in particular tables 2 to 6) could be moved to the supplement. It would not change the

essence of the manuscript and make it lighter.

Beyond the form of the manuscript, the content is very interesting and well addressed. In my opinion, it corresponds to what is expected from a paper on description/evaluation of a model, that can help the other developers. I would then recommend its publication, after the authors have answered or addressed the following questions or remarks :

1. General remark The authors mention that 2 simulations have been run with the model, one in free run (FR),and the other in nudged mode (ND). That is of course common and very interesting. But all along the manuscript, the evaluation is made sometimes on the FR simulation, sometimes on the ND one without any justification (except in the confrontation to satellite measurements section). Strikingly, for instance, the 4.1 section begins with FR and continues with ND.. Could the authors explain what has driven their choices? Moreover, I could not find a comment on the eventual difference of behavior of the two modes (FR and ND). I think that the manuscript would benefit from being completed by this analysis.

2 Specific comments/remarks

Section 2.2, page 6 line 8-10 : could the author give more details about the solver used in the model? The solver is known to be a crucial point and would deserve a few lines to describe its characteristics (accuracy, conservatism etc..)

Section 2.6.1 page 21 line 41 : 'For VOCs, emissions of all C2 and C3 VOCs are included as ethane and propane, respectively' . Could the author precise if there is any aggregation coefficient linked to the reactivity to the species?

Section 2.6.2 page 22 line 36-38 : 'However, it has been argued that wide model agreement is achieved rather due to model tuning than due to a high level of process understanding'. First I don't understand what this sentence has to do with here. Second, assuming it has its place, what should be inferred from it? It sounds a little too

elliptic for such a precise paper.

Section 3; table 11 : the amount of NOx emissions is set as 130.1 Tg(NO)/year, that corresponds to roughly 60.7 Tg of N. This is considerably higher than the average of previous studies. In Young et al 2013 for instance, ACCMIP models had in average rather 47Tg of N / year, and the variability was linked to the LiNOx emissions, that have been calibrated here. I think it is indispensable that the author give a comment on that point, before comparing to other studies. Maybe this higher emissions could explain some following discrepancies.

Section 4.1 : could the author specify how the average is made (what output frequency is used, and what is the temporal resolution of the TOAR database?)

Section 4.2 : I did not see any global budget of the total dry deposition of ozone in the model (despite a reference to the 1000Tg generally thought to be a good estimation)

Section 4.4 : Comparison to observed H2O2. I find this subsection questionable :1) monthly means are compared to instantaneous measurements, 2) some observations are from the beginning of the 1990's - can we really be sure that nothing has changed ? As the authors mention, caution has to be applied and I think nothing can be really deduced from this comparison. I would simply skip it.

Section 4.5 page 50, line 22-25 : could another explanation simply be that emissions are underestimated in the Northern Hemisphere? Section 4.5 page 51 line 12-14 : the authors mention that the model captures the inter annual variabllity. Could they precise here (or in the emissions section) if an inter annual variability takes place in the emissions? Depending on this, the comment could be different.

Section 4.5 page 52, line 15-16 : the authors mention that higher tropospheric columns of NO2 in summer could be attributed to higher emissions. Figure 5 seems to show that it is not the case. Section 4.5 page 52, line 20 -22 : this sentenceÂă'These biases . . . troposphere)' is unclear to me. Section 4.5 page 53, line 1-7 : this paragraph is not

in agreement with the injections of biomass burning emissions in altitude as described in section 2.6.1

Section 4.6 page 57 line 9 :typo (two times 'region')

---

## Author Response (AR1)

Responses to Reviewer Comments on "Description and evaluation of the UKCA stratosphere-troposphere chemistry scheme (StratTrop vn 1.0) implemented in UKESM1" by Archibald et al.

We would like to thank both referees and the editor for a timely turnaround of the paper (especially given its size!) and for their comments which have helped improve the revised version. We deal with each referee's point in turn below in blue text.

**Anonymous Referee #1**
*The manuscript by Archibald et al. presents an exhaustive description of the latest version of the gas-phase chemistry scheme implemented in the UK Earth System Model. The documentation includes a description of the gas-phase chemical mechanism that is applicable to the troposphere and middle atmosphere, as well as a description of associated processes such as photolysis, wet and dry deposition and interactive emissions such as lightning NOx. The discussion then moves on to presenting diagnostics of the behaviour of the model in both the troposphere and stratosphere.*

*The manuscript is well laid out and the presentation of the model components and diagnostics of the model behaviour are clearly discussed. My only significant criticism is the length of the paper. I understand the need to document the many different components of the model and the authors do a good job of referring the reader to other papers for a more in-depth discussion of various model components. But the inclusion of large tables providing the list of all chemical species and reactions in the main body of the manuscript forces the reader to push through many pages before the discussion continues. I strong urge the authors to move many of these tables (Tables 1, 2, 3, 4, 5, 6, 7, 8 and 9) into the supplementary material and provide the reader with a much more compact description of the important facets of these model components. In addition, in the detailed comments for Pages 31 – 35 there is additional material on emissions that I would suggest the authors re-consider in the interest of making the paper easier to get through.*

Response: We thank the reviewer for their comments on the manuscript. We agree that the paper is very long and we have tried to address this issue in our revision adopting the referees specific points. We have moved Tables 2-9 from the discussion paper into the Supplement but kept Table 1. We have moved the material the referee highlights on the emission set-ups for the model simulations into the Supplement too.

*After having said all that, a few suggestions to make the paper longer... Could the authors include a very short description of how polar stratospheric clouds are treated in the model? I could not find any information on how these are calculated in the model. Note that there is a reference to Denison et al. (2018) in the caption for Table 5, but the reference is missing. And while a significant part of the paper is dedicated to assessing the stratosphere in the model there is no presentation of the total column ozone climatology that is produced by the model. Such a basic and well-observed quantity should really appear in a documenting paper.*

Response: We thank the referee for their time and for their valuable comments. We have added the following text to the manuscript: "The treatment of Polar Stratospheric Cloud (PSC) has been recently expanded in UKCA (Dennison et al. 2019), but these improvements did not make it into the UKESM1 version of UKCA discussed here, which remains unmodified from the original Morgenstern et al. (2009) scheme. The abundance of nitric acid trihydrate (NAT) and mixed NAT/ice polar stratospheric clouds is calculated following Chipperfield (1999)

assuming thermodynamic equilibrium with gas-phase HNO₃ and water vapour; the treatment of reactions on liquid sulfate aerosol also follows Chipperfield (1999). Sedimentation of PSCs is included in the model whilst dehydration is handled as part of the model's hydrological cycle. Denitrification is prescribed in the same way as in Chipperfield (1999) with two different sedimentation velocities. We refer the reader to Morgenstern et al. (2009) and Dennison et al. (2019) for further details. "

and have added in new reference:

Chipperfield, M. P.: Multiannual simulations with a three dimensional chemical transport model, J. Geophys. Res., 104(D1), 1781–1805, 1999.

We have also added a new evaluation of the total column ozone climatology (§4.4). The new figure (R1 ) show that the model generally captures the main features of the total ozone climatology well but with some biases. In particular the tropical column is biased high when compared to the Boedecker climatology (2005-2010) and the ozone hole extends for too long in the model, leading to low biases in the austral summer. High biases are also found in the boreal extratropics in regions where transport of lower stratospheric ozone dominates suggesting that the cause of this could be related to the bias in the tropics. Comparison of the OMI-MLS tropospheric column bias suggests that a large fraction of the total column ozone bias is attributed to a high bias in the troposphere.

[Figure]

**Figure R1.** Evaluation of the FR UKCA total ozone climatology. Panel (a) shows the FR simulation multi annual mean total column ozone climatology (DU). Panel (b) shows the difference (DU) between the FR ozone climatology and the Boedeker ozone climatology v2.8.

We have added the following text:
"**4.4 Comparison of total ozone column**
Here we discuss the modelled total ozone column through analysis of the data from the FR simulation averaged over the 2005-2014 period. We note here that there is little difference between the ND and FR total column so for simplicity we focus on the FR data.

Figure 13 (panel a) shows the multi annual average total ozone column in Dobson Units as a function of latitude and time. As with most chemistry-climate models (Dhomse et al., 2018), UKCA simulates the main features of the total column well, with a minimum in the tropics and maxima at high latitudes during the hemispheric spring seasons. When compared with the total column ozone in older versions of UKCA (M09 Figure 9) the current model configuration simulates similar biases at high latitudes but a pronounced positive bias in the tropics. Figure 13 (panel b) highlights that the tropical column is biased high by 30-40 DU when compared to the Bodeker climatology (Hassler et al., 2008) and the Antarctic ozone hole extends for too long in the model, leading to low biases in the austral summer. The high biases in total column

ozone in the tropics are very likely in part to be driven by high biases of around 15 DU in the tropical tropospheric ozone column (see Section 4.5 below). The extra tropical biases may well be related to this bias through the transport of ozone-rich air in the UTLS into this region but further work is needed to resolve the causes of the bias in the total ozone column."

and have added in an acknowledgment to Boedker Scientific for use of their data and the following reference:

Hassler, B., G. E. Bodeker, and M. Dameris (2008), Technical Note: A new global database of trace gases and aerosols from multiple sources of high vertical resolution measurements, Atmospheric Chemistry and Physics, 8, 5403-5421.

*Page 3, Line 25 – The reference for Quiquet et al. (2015) seems to be missing.*
Response: Thanks. We have fixed this error.

*Page 21, Lines 26 – 28 - The discussion of 'Boreal and temperate forest and deforestation emissions..' is obviously referring to biomass burning but the passage could use a more explicit reference to burning.*
Response: We thank the reviewer. We have now made an explicit reference to biomass burning emissions in the revised manuscript in Section 2.6.1.  We also note, there was an error in the discussion manuscript in the way the emissions were described on Page 21, Lines 26-28 i.e. "Boreal and temperate forest and deforestation emissions are considered 'high-level' and are spread uniformly up to level 20 (~3 km in L85)".

This has been re-written as follows: "*While boreal and temperate forest and deforestation emissions of black carbon (BC) and organic carbon (OC) are considered 'high-level' (Mulcahy et al., 2019) and are spread uniformly up to level 20 (~3 km in L85), all gas-phase biomass burning emissions are added to the surface layer.*"

*Page 21, Line 32 – The word 'that' in 'The CEDS emissions are generally greater that those of...'*
Response: Thanks for spotting this error. Now corrected as suggested.

*Page 23, Line 18 – The NO production from lightning is given as molecules/J, but the assumed energy of a cloud-to-cloud or cloud-to-ground flash is not given so it is not possible to deduce the NO production per flash. Could the authors include either enough information to calculate the production per flash or the production per flash Directly.*
Response: Thank you for pointing out this omission. We have now amended Page 23, Lines 17-19 as follows: "For $NO_x$ production, the parameterisation assumes that the production efficiency per unit of energy discharged is $25 \times 10^{16}$ molec (NO) $J^{-1}$, with the energy discharged from cloud-to-ground flashes ($3.0 \times 10^9$ J $flash^{-1}$) being approximately 3 times greater than that for cloud-to-cloud ($0.9 \times 10^9$ J $flash^{-1}$) flashes (Schumann and Huntrieser, 2007)."

In adding the energy discharged per flash, we also noticed that the Schumann and Huntrieser (2007) reference was missing from the reference list in the submitted version. This has now been added to the revised manuscript.

*Page 29, Line 1 – In Figure 4 is it possible to add the letters of each run that appears in Table S1 to the description of each change? Something like 'New to old J-rates (B to D)'?*
Response: We thank the reviewer for this comment and we have modified accordingly.

*Page 31, Line 4 – Page 35, Line 2 – I would suggest either cutting completely or moving to the supplementary material the summary of the sectors, geographic distribution and seasonal cycle of emissions presented here. The figures and tables are based on prescribed emissions that come from independent sources and present information that is relatively well known to anyone with a background in global tropospheric chemistry – the seasonal importance of biomass burning emissions in different regions, for example.*
Response: We thank the reviewer for their suggestion. On this basis, we have replaced all the text, Table 11, and Figures 5 and 6 from Page 31, Line 4 to Page 35, Line 2 with the following text: "Table S10 provides a summary of the sectors contributing to the emissions of the nine tropospheric ozone precursor species treated in UKCA StratTrop and their corresponding global annual totals, averaged over the 2005-2014 time period covered by the two simulations. Figures S7 and S8 show the multi-annual global annual mean distributions and the seasonal cycle for different emission sectors and regions for NO and CO, respectively."

And moved all the pre-existing text, table, and figures into the Supplementary Material. Relevant figure numbers all updated.

*Page 41, Line 13 – Regarding the comparison of dry deposition against field observations (Figure 11), do you compare the ozone flux from the model for a particular vegetation type that is most like the one that dominates the observation site or is the comparison using the flux averaged over the model grid square?*
Response: The ozone surface deposition flux in the model is diagnosed (currently) as the surface tile area weighted gridbox mean. We did not make an attempt to diagnose the dry deposition flux of ozone from the land surface type closest to the properties of the observation site since such a diagnostic is currently lacking from the model. We are considering, however, to implement such a diagnostic capability in a future model version. Corresponding figure caption updated for clarity.

*Page 49, Lines 10 - 14 – Looking at Figure 17b, the uncertainties on the tropical tropospheric ozone column defined as two times the variance span quite a large range and easily encompass the model results. But in Figure 16, showing the geographic distribution of the average DJF and JJA columns, large sections of the tropics show the differences as being significant. I know there are different quantities being shown, but the difference between the model and the 2xstdev is quite large. I am also a bit concerned because the upper and lower bounds on the 2xstdev do not seem symmetric around the mean. Particularly the lower bound on the 2xstdev seems to have a variation that is independent of the mean.*
Response: We thank the reviewer for their comments. We have thoroughly checked the scripts and data used to create the ozone timeseries figures the referee refers to. After careful analysis we can confirm that the plots do show that the 2xstdev is symmetric around the mean ozone. On inspection of the figure we note that it is the red line (the model data) which "trick" the eye into thinking that the observational error bounds are unsymmetric.

With regards to discrepancy between Fig 16 and 17, these show indeed different things. In Figure 16, the uncertainty used to calculate significance is across times for a single geographical location. In Fig 17 the uncertainty is across different geographical locations (within the specified latitude band) for a single time. Ozone geographical variability within the Tropics is on average ~10 DU (used in fig 17b), which is larger than the time (seasonal/interannual) variability in the Tropics, which has a value of ~3-5DU (this is used in Fig 16). The latter is calculated using monthly means, probably different results would be obtained if daily or hourly data is used.

*Page 50, Line 9 – misplaced bracket in 'MOPITT instrument on board (Terra, Emmons et al. 2004).'*
Response: Thanks, we have fixed this in the revised version.

*Page 50, Line 22 – Page 51, Line 2 – I believe problems with the anthropogenic emission estimates of CO has been flagged as a possible source of the general low bias of CO in the northern hemisphere – Miyazaki et al. Atmos. Chem. Phys., 15, doi:10.5194/acp-15-8315-2015, 2015, for example. But that would have been for older emissions inventories and I am not sure how that would apply to the CMIP6 inventory.*
Response: We thank the reviewer for their comment. We have added in a new point based on this: *"There are a number of reasons for the model-satellite biases in TC_CO, including 1) CO emissions in the NH are underestimated (Miyazaki et al., 2015), 2) insufficient secondary production of CO from non-methane VOC oxidation (e.g. Grant et al., 2010), 3) excess biomass burning emissions in the southern hemisphere (SH) during DJF (potentially the same cause in central Africa in JJA), 4) strong loss through OH in the NH in both seasons. We note that these types of biases are not unique to UKCA StratTrop and that further work is required to ameliorate them (Shindell et al., 2006)."* and added in a new reference (Miyazaki et al., 2015).

*Page 53, Lines 1 – 6 - 'Over biomass burning regions... In UKEMS, the anthropogenic emissions are injected on the surface level, so most of the NOx will be trapped in the boundary layer where OMI is less sensitive.' From Page 21 the authors state the biomass burning emissions are injected over the lowest 3 km of the model, which does not seem to agree with what is stated here.*
Response: We thank the reviewer for this point. We have corrected this mistake in the description of the vertical profiling of emissions in the revised manuscript.

*Page 62, Figure 29 – the linear regression from the model is shown in the left-most panel for each of the comparisons, but it would help greatly in the comparison with ACE and MIPAS if the regression for the model was reproduced in the other two panels.*
Response: We thank the reviewer for their comment. We have modified the figure to include the model regression line in all panels.

*Page 63, Lines 27-28 – It should be '(panel d)' in 'However, the zonal cross section at 23 km (~ 50 hPa) (panel c) shows...'. More fundamentally, there is no explanation for the red region in panel d.*
Response: Thanks. We have fixed this.

**Anonymous Referee #2**

*In this manuscript, Archibald et al aim at giving a comprehensive description and evaluation of the gas phase chemistry scheme used in the Earth System Model UKESM1. A first part is dedicated to the description of the implementation of the physical and chemical processes in the model, a second part to a very detailed evaluation of the model, from deposition on the ground to the upper atmosphere.*

*In general, this manuscript is very well written and pleasant to read, due to its very clear organization. Many details are given, and they all are usefull to get a precise description of the model, but it gives a very long manuscript. Maybe some tables (in particular tables 2 to 6) could be moved to the supplement. It would not change the essence of the manuscript and make it lighter.*

Response: Again we would like to thank the referee for their time and comments. As this comment is similar to that from referee #1 we feel moving Tables 2-9 to the SI addresses this point.

*Beyond the form of the manuscript, the content is very interesting and well addressed. In my opinion, it corresponds to what is expected from a paper on description/evaluation of a model, that can help the other developers. I would then recommend its publication, after the authors have answered or addressed the following questions or remarks :*

*1. General remark The authors mention that 2 simulations have been run with the model, one in free run (FR),and the other in nudged mode (ND). That is of course common and very interesting. But all along the manuscript, the evaluation is made sometimes on the FR simulation, sometimes on the ND one without any justification (except in the confrontation to satellite measurements section). Strikingly, for instance, the 4.1 section begins with FR and continues with ND.. Could the authors explain what has driven their choices? Moreover, I could not find a comment on the eventual difference of behavior of the two modes (FR and ND). I think that the manuscript would benefit from being completed by this analysis.*

Response: We greatly thank the reviewer for their comment. To address the point on the underlying philosophy first: we intended to make two sets of simulations to evaluate the performance of UKCA in UKESM1, with one set of simulations enabling a less biased evaluation against observations. The first set of simulations using prescribed sea surface temperatures (FR) were aimed at evaluating the model in atmosphere only mode (note Sellar et al. (2019) and many future studies will show results from UKCA in the full Earth System model configuration, warts and all). The second, using atmospheric nudging (ND) to meteorological reanalysis, was designed for high time frequency output and comparison with surface and satellite data as the main objective. Whilst we appreciate that nudging may not result in "perfect" transport (Orbe et al., 2019), we feel that the ND simulations should enable comparisons with observations that are less biased. However, in the end (partly owing to errors in outputting fields of model data in both runs) we analysed output from both sets of simulations against observations (i.e. we included a comparison to stratospheric tracers focusing only on the FR simulations in Section 4.8).

In reviewing the manuscript we noted that there was confusing use of ND and FR runs and even noticed that we had got some of the text mixed up. What we had not shown in the manuscript, as the referee points out, was if there was any difference between the ND runs

and FR runs. We show here that for some species there is a difference and propose to modify the manuscript to make account of this. We also propose to clearly state in the figure captions which model run the data presented come from (FR or ND). Finally, where possible we also have made changes to the manuscript to present a more unified approach to the presentation of the results. For example, Section 4.2.1 and Section 4.5 which make most evaluation against ozone observations make use of consistent model simulations (ND).

With regards to Section 4.1 (the comparison to surface observations), the first reference to FR is erroneous and should have referred to the ND simulations instead as that is what the comparisons were based on. We have now changed this in the revised manuscript. We have compared the seasonal surface $O_3$ in both the FR and ND simulations, as well as recreated the figures both with the FR and ND simulations. Figure R2 shows here the seasonal difference in surface $O_3$ between the ND and FR simulations (averaged over 2005-14 period). The largest differences seem to occur over the Tropical ocean regions and south Asia. However, there are few (if any) surface observation locations in these areas of largest discrepancy. Therefore the evaluation of surface $O_3$ is very similar in both the ND (Figure R3) and FR (Figure R4) model simulations. Comparing Figure R3 and R4 highlights that over the regions of surface observations from the TOAR database there are very small differences between the FR and ND runs and so we propose to keep the figures as they are in the manuscript but clarify that the analysis is of the ND simulations throughout this section and highlight that whilst there are some significant differences between the ND and FR runs that these occur in regions where there are few observations to constrain the differences.

[Figure]

**Figure R2.** Seasonal difference in surface O$_3$ between the ND and FR model simulations over the period (2005-14)

[Figure]

**Figure R3.** NMBF of surface O$_3$ between the Observations and FR simulations over the period 2005-14

[Figure]

**Figure R4** (Figure 6 in revised paper). NMBF of surface $O_3$ between the Observations and ND simulations over the period 2005-14

Section 4.3 included data solely focussed on the FR simulation in the submitted manuscript. We show here (Figure R5) the differences between the FR and ND simulations for the vertical methane profile. This highlights that there are differences in the vertical methane profile and we have included this new figure as a replacement to Figure 13 in the revised manuscript.

[Figure]

**Figure R5** (Figure 11 in revised manuscript). Vertical profiles of the mean tropical (±10 °N) modelled methane from multi-annual mean output from atmosphere-only simulations of HadGEM2-ES (blue; OC14) and UKCA StratTrop in the free-running (FR; green) and nudged (ND; red) simulations of UKESM1 (this study). The shading represents ±1 standard deviation about the multi-annual mean.

Figure R5 highlights the largest differences in the global mean vertical profile of methane between the ND and FR simulations were above 1 hPa. However, the two simulations are very similar below 1 hPa and the difference between the ND or FR UKESM1 simulations and HadGEM2-ES is much larger. We also show below that the comparison between methane in the ND and FR simulations is fairly small when looking at satellite data too.

[Figure]

**Figure R6.** Left hand side (a-d) FR data (as submitted in paper - originally Fig 14, now 12). Right hand side (a-d) ND data.

Figure R6 highlights that the comparisons to HALOE/CLAES are extremely similar between the FR and ND runs, except that the RMSEs are slightly lower in the ND than in the FR one.

Given that the ND simulations are to some extent constrained by the re-analysis we felt it was more important to evaluate the climatological age of air distribution in the free running simulation. In Section 4.8 We have added the following text: "Figure 28 compares data from the FR simulation and observations. The FR run is shown here as this allows for a more robust comparison of the model data where it is not constrained by the re-analysis meteorology. Figure 28 shows the..."

*2 Specific comments/remarks*
*Section 2.2, page 6 line 8-10 : could the author give more details about the solver used in the model? The solver is known to be a crucial point and would deserve a few lines to describe its characteristics (accuracy, conservatism etc..)*
Response: We have expanded on the description of the solver with the text below:
"The time-dependent chemical reactions are integrated forward in time using an implicit backward Euler solver with Newton Raphson iteration (Wild and Prather, 2000). This solver has a relative convergence criterion of $10^{-4}$ with a time step of 60 minutes throughout the atmosphere. An extensive discussion of the solver used here is presented in Esentürk et al. (2018)." and added new references:

Esentürk, E., Abraham, N. L., Archer-Nicholls, S., Mitsakou, C., Griffiths, P., Archibald, A., and Pyle, J.: Quasi-Newton methods for atmospheric chemistry simulations: implementation in UKCA UM vn10.8, Geosci. Model Dev., 11, 3089–3108, https://doi.org/10.5194/gmd-11-3089-2018, 2018.

Wild, O. and Prather, M. J.: Excitation of the primary tropospheric chemical mode in a global three dimensional model, J. Geophys. Res.-Atmos., 105, 24647–24660, 2000.

*Section 2.6.1 page 21 line 41 : 'For VOCs, emissions of all C2 and C3 VOCs are included as ethane and propane, respectively' . Could the author precise if there is any aggregation coefficient linked to the reactivity to the species?*
Response: We can confirm there is no modification to the lumping coefficient to account for differences in OH reactivity. As such the emissions are purely lumped by mass.

*Section 2.6.2 page 22 line 36-38 : 'However, it has been argued that wide model agreement is achieved rather due to model tuning than due to a high level of process understanding'. First I don't understand what this sentence has to do with here. Second, assuming it has its place, what should be inferred from it? It sounds a little too elliptic for such a precise paper.*
Response: We thank the reviewer for their remark. In our comment in the manuscript we wanted to draw attention to a paper published some time ago by Arneth et al. (2008) in which the authors argued that good agreement on the global isoprene flux between BVOC emission

models is potentially due to the fact that modellers tend to tune emission factors in order to reach the "magical number" of 500 Tg(C)/yr at the present day. Thus, good agreement does not necessarily reflect a high degree of process understanding. We concede that this may not be the best place to discuss philosophical issues in BVOC model development. The remark has been removed.

*Section 3; table 11 : the amount of NOx emissions is set as 130.1 Tg(NO)/year, that corresponds to roughly 60.7 Tg of N. This is considerably higher than the average of previous studies. In Young et al 2013 for instance, ACCMIP models had in average rather 47Tg of N / year, and the variability was linked to the LiNOx emissions, that have been calibrated here. I think it is indispensable that the author give a comment on that point, before comparing to other studies. Maybe this higher emissions could explain some following discrepancies.*

Response: Thank you for pointing this out. There are two reasons for the difference in global annual total NOx emissions between this study and the ACCMIP studies. ACCMIP used decadal mean emissions, centred on the year 2000 for the present day whereas CMIP6 has annual emissions (including a seasonal cycle) up to and including the year 2014. In the simulations evaluated here, we used the transient CMIP6 emissions although in the figures and the corresponding table (now moved into the Supplementary Material), we used a time-mean of the emissions over the evaluation period (i.e. 2005-2014). In Hoesly et al. (2018), the CMIP6 emissions show a significant positive trend in the anthropogenic NOx emissions between 2000 and 2014, so the difference in time period is one contributing factor. However, comparing CMIP5 anthropogenic emissions with CMIP6 (excluding aircraft and agricultural waste burning) for the same time period also shows an appreciable difference (See Figure 2 in Hoesly et al., 2018), with CMIP6 emissions being approximately 10 % higher than CMIP5.

Some text has now been added on the differences between the CMIP5 and CMIP6 emissions to the Supplementary Material as follows:

*"It is worth noting that the global annual total for anthropogenic NOx emissions (Hoesly et al., 2018), for example, is higher than that used in the 5th Coupled Model Intercomparison Project (Lamarque et al., 2010) and the Atmospheric Composition and Climate Model Intercomparison Project (ACCMIP; Lamarque et al., 2013). Part of this difference is due to the time period. ACCMIP used decadal mean emissions, centred on the year 2000 for the present-day whereas here, transient emissions with an annual frequency up to and including the year 2014 are used. With a strong positive trend in anthropogenic emissions shown in Hoesly et al. (2018) since 2000, this difference in time period will be a contributing factor. However, even comparing CMIP5 and CMIP6 anthropogenic emissions (excluding aircraft and agricultural waste burning) for the same time period shows an appreciable difference (Figure 2 in Hoesly et al., 2018), with CMIP6 emissions being approximately 10 % higher than those used in CMIP5. This could have implications for model performance relative to ACCMIP studies. "*

*Section 4.1 : could the author specify how the average is made (what output frequency is used, and what is the temporal resolution of the TOAR database?)*

Response: As far as we understand the averaging is consistent across time for the model and observations. Monthly mean observations at rural sites have been obtained from the TOAR database for comparison to the monthly mean output (sampled over all model timesteps) from the model simulations. We have added text to section 4.1 in the revised manuscript to reflect this.

*Section 4.2 : I did not see any global budget of the total dry deposition of ozone in the model (despite a reference to the 1000Tg generally thought to be a good estimation)*

Response: We thank the reviewer for pointing out this gross omission. We have updated the text to be more specific as follows (the new text below): *"1030 Tg ($O_3$), around 20 % to 25 % of the gross chemical ozone production in the troposphere, is removed from the atmosphere in the ND simulation through dry deposition at the surface (Stevenson et al., 2006; Wild, 2007; Young et al., 2013; Hardacre et al., 2015)."*

We note that while uptake by terrestrial vegetation plays a crucial role, Hardacre et al. (2015) demonstrated that the oceans represent a very important sink, too. Much uncertainty still remains about the exact magnitude and many of the processes around ozone removal at the surface (e.g., Hardacre et al. 2015; Luhar et al., 2017). A thorough evaluation and, if necessary, re-calibration of ozone dry deposition models is, thus, crucial in developing robust models of atmospheric composition.

*Section 4.4 : Comparison to observed H2O2. I find this subsection questionable :1) monthly means are compared to instantaneous measurements, 2) some observations are from the beginning of the 1990's - can we really be sure that nothing has changed ? As the authors mention, caution has to be applied and I think nothing can be really deduced from this comparison. I would simply skip it.*

Response: We are happy to remove this and move to the SI but would like to keep it there. We note that there are few evaluations of global vertical and zonal distributions of $H_2O_2$ in spite of the important role it plays in the Sulfur cycle. However we agree that these comparisons do not enable a rigorous assessment that we can benchmark our model against and so it belongs not in the main discussion.

*Section 4.5 page 50, line 22-25 : could another explanation simply be that emissions are underestimated in the Northern Hemisphere? Section 4.5 page 51 line 12-14 : the authors mention that the model captures the inter annual variabllity. Could they precise here (or in the emissions section) if an inter annual variability takes place in the emissions? Depending on this, the comment could be different.*

Response: We thank the reviewer for their point which we have addressed in the manuscript in Section 2.6.

*Section 4.5 page 52, line 15-16 : the authors mention that higher tropospheric columns of NO2 in summer could be attributed to higher emissions. Figure 5 seems to show that it is not the case. Section 4.5 page 52, line 20 -22 : this sentence 'These biases ˇ. . . troposphere)' is unclear to me. Section 4.5 page 53, line 1-7 : this paragraph is not in agreement with the injections of biomass burning emissions in altitude as described in section 2.6.1*

Response: We apologise for the confusion here and have rectified the text to reflect that all gas-phase emissions are injected into the model surface layer. Only OC and BC emissions from biomass burning have multi level emissions.

*Section 4.6 page 57 line 9 :typo (two times 'region')*

Response: Thanks for pointing this out. Now removed.

**References:**

Orbe, C., Plummer, D. A., Waugh, D. W., Yang, H., Jöckel, P., Kinnison, D. E., Josse, B., Marecal, V., Deushi, M., Abraham, N. L., Archibald, A. T., Chipperfield, M. P., Dhomse, S., Feng, W., and Bekki, S.: Description and Evaluation of the Specified-Dynamics Experiment in the Chemistry-Climate Model Initiative (CCMI), Atmos. Chem. Phys. Discuss., https://doi.org/10.5194/acp-2019-625, in review, 2019.

Sellar, A. A., Jones, C. G., Mulcahy, J., Tang, Y., Yool, A., Wiltshire, A., et al. UKESM1: Description and evaluation of the UK Earth System Model. *Journal of Advances in Modeling Earth Systems*, 11. 2019.